# Memory Savings at What Cost?
# A Study of Alternatives to Backpropagation

## Abstract

Forward-mode automatic differentiation (FMAD) and zero-order (ZO) optimization have been proposed as memory-efficient alternatives to backpropagation (BP) for gradient computation, especially in low-resource settings. However, their practical benefits remain unclear due to two key gaps: a lack of comparison against memory-efficient BP variants, such as activation checkpointing, and a lack of systematic characterization of tradeoffs between accuracy, memory, and computation efficiency among these methods. This work presents a comprehensive comparison of BP, FMAD, and ZO methods. Through theoretical analysis under a common framework, we present intuition that, while FMAD and ZO can reduce memory usage, they incur significant costs in accuracy, convergence speed, and computation compared to BP with checkpointing. These drawbacks worsen with larger models or constrained perturbation budgets. Through empirical experiments on large language and vision-language models, we show that BP with checkpointing outperforms FMAD and ZO variants, including those enhanced with variance reduction, achieving up to 31.1% higher accuracy, 34.8% faster convergence, and 3.8× fewer computations at comparable memory usage. We also investigate specific failure modes in FMAD and ZO, including instabilities in Jacobian-vector products that can destabilize training. Our results highlight fundamental limitations of FMAD and ZO, and the effectiveness of BP with checkpointing for model training under memory-constrained settings.

## 1 Introduction

Backpropagation (BP) (Rumelhart et al., 1986) is the standard algorithm for gradient computation in deep learning due to its convergence efficiency and widespread support in automatic differentiation frameworks such as PyTorch (Paszke et al., 2019) and JAX (Bradbury et al., 2018). However, BP incurs high memory overhead in training large models, as it must store intermediate activations for the backward pass. To address this limitation, recent research has explored alternative gradient estimation methods such as forward-mode automatic differentiation (FMAD) (Baydin et al., 2017; 2022; Panchal et al., 2024) and zero-order (ZO) optimization (Richardson, 1955; Malladi et al., 2023), which approximate gradients (using directional derivatives or two forward pass evaluations) based on randomly perturbed weights. These methods are often promoted as memory-efficient or hardware-friendly alternatives to BP, especially in resource-constrained or non-differentiable settings (Panchal et al., 2024; Malladi et al., 2023; Xu et al., 2024).

Despite growing interest, prior work on FMAD and ZO suffers from two critical limitations that leave their practical value inadequately understood. First, the existing comparisons (Gautam et al., 2024; Zhang et al., 2024) often overlook activation checkpointing (Chen et al., 2016), a widely supported and effective BP variant that substantially reduces memory usage by recomputing rather than storing intermediate activations. Second, as shown in Table 1, key considerations such as computational cost and wall-clock time to convergence are often omitted, leaving even the comparisons against vanilla BP incomplete. This one-sided narrative of ZO and FMAD as superior to BP motivates our study: we aim to provide a comprehensive account of these trade-offs, encompassing not only memory usage but also convergence speed and overall computational efficiency of the gradient estimation methods.

This paper addresses the above-mentioned limitations through a comprehensive study of BP, FMAD, and ZO approaches. We first outline the expected trade-offs among convergence behavior, memory

Table 1: While some existing research empirically compares vanilla backpropagation (BP-Vanilla) across multiple metrics including memory usage, convergence time, and computational cost, they examine only a subset of these criteria, and notably, none include comparisons with backpropagation using checkpointing (BP-Checkpointing). We omit accuracy as it is evaluated in all the studies.

| METHODS | CONV. TIME | MEMORY | COMP. COST | CONTRIBUTION |
|---|---|---|---|---|
| MEZO (Malladi et al., 2023) | ✗ | ✓ | ✗ | ZO uses 12× less memory than Vanilla BP while achieving accuracy within 5%. |
| MEZO-SVRG (Gautam et al., 2024) | ✗ | ✓ | ✗ | Enhances the convergence accuracy of MEZO through variance reduction, improving accuracy by up to 20%. |
| Revisiting ZO (Zhang et al., 2024) | ✗ | ✓ | ✗ | Benchmarks ZO optimization in LLM fine-tuning, along with proposing novel techniques that enhance accuracy over MeZO by up to 3%. |
| ZOSPARSE (Guo et al., 2025) | ✓ | ✗ | ✗ | ZO fine-tuning achieves full ZO accuracy by updating just 0.1% of sensitive parameters, with up to 2.5× speedup. |
| SPRY (Panchal et al., 2024) | ✓ | ✓ | ✗ | Distributes trainable parameters across federated clients, improving FMAD's convergence speed by up to 20× and final accuracy by up to 13% compared to ZO. |
| FOMOH (Cobb et al., 2024) | ✗ | ✗ | ✗ | Introduces forward-mode second-order optimization; improves accuracy by 1–3% compared to first-order FMAD on logistic regression and CNN tasks. |
| This paper | ✓ | ✓ | ✓ | First to evaluate how BP with checkpointing fares in the three-way tradeoff vs. variance-reduced ZO and FMAD. |

consumption, and computational cost as functions of model dimensionality $d$ and the number of perturbations per iteration $n$. These theoretical results suggest that, while FMAD and ZO may reduce memory under certain regimes (e.g., when perturbations are evaluated sequentially), they face scalability challenges: higher per-iteration computational cost, $\mathcal{O}(nd)$, and slower convergence in high dimensions or with limited perturbation budgets. In contrast, BP with activation checkpointing is expected to achieve favorable convergence with comparable memory usage.

We then conduct extensive empirical evaluations on large language and vision-language models across tasks including text classification, text generation, and visual question answering. We compare BP with checkpointing against a wide range of FMAD and ZO variants (including SVRG (Liu et al., 2018), multiple perturbations per iteration (Feng et al., 2024)), and our own enhanced versions with variance reduction: gradient accumulation and adaptive perturbation sampling. As illustrated in Figure 1, BP with checkpointing consistently achieves higher accuracy and faster convergence, while using memory on par with FMAD and ZO variants.

Beyond standard performance metrics (accuracy, memory, and convergence time), we also perform a dedicated study of specific failure modes in FMAD and ZO, focusing on instabilities in Jacobian-vector products (`jvps`) that can arise under adaptive optimizers and hinder conver-

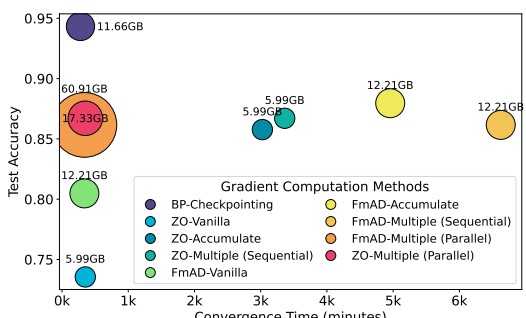

Figure 1: The three-way trade-off between accuracy, convergence time, and memory consumption during training of LLAMA 3.1 (8B) on the AG-News dataset. The circle radii are proportional to the memory consumption. BP-CHECKPOINTING achieves highest accuracy with lowest convergence time using comparable memory to FMAD and ZO variants. §4 describes these methods in detail.

gence. This analysis provides insight into why these gradient estimation methods behave unpredictably in practice and complements our broader evaluation of their scalability and reliability.

These findings lead to a critical insight: despite recent enthusiasm for forward-mode and zero-order methods (Panchal et al., 2024; Malladi et al., 2023; Gautam et al., 2024; Liu et al., 2018), they remain fundamentally constrained by their inability to efficiently scale to large models. Rather than serving

as alternatives to backpropagation, they operate as inefficient approximations that trade off accuracy or convergence speed for marginal memory reductions.

This paper's main contributions are:

- A theoretical analysis of the convergence rates, memory cost, and compute complexity of BP, FMAD, and ZO under *a common theoretical framework*, highlighting their three-way trade-offs.
- A comprehensive empirical study of BP, FMAD, and ZO on large-scale models across diverse tasks. We show that BP with checkpointing consistently achieves 4.5–31.1% higher accuracy, 21.4–34.8% faster convergence, and 3.2–3.8× lower computation cost than FMAD and ZO variants, while using comparable memory.
- The design and benchmarking of two new variance reduction methods for FMAD and ZO. These methods improve accuracy by 7.5–14.0%, but still fall short of BP's overall efficiency, and introduce overheads in either convergence time or memory.
- An analysis of FMAD's and ZO's failure modes, including high-dimensional perturbations, noisy Jacobian-vector products, and optimizer-dependent instabilities (e.g., abrupt `jvp` spikes under adaptive optimizers like AdamW) that destabilize training and degrade convergence.

## 2 BACKGROUND

This section reviews three gradient computation techniques central to our study: (a) reverse-mode automatic differentiation (RMAD, of which backpropagation is a special case), (b) forward-mode automatic differentiation (FMAD), and (c) zero-order (ZO) finite-difference methods. For an in-depth survey of these approaches, we refer readers to Baydin et al. (2017). Appendix A reviews related work in detail. Details on signal propagation mechanism of these methods are in Appendix G.

The three methods described below operate on a function $f$, which in deep learning corresponds to a neural network and can be non-convex. This function $f$ is composed of nested functions $f_i$, $i \in [p]$, where $p$ is the number of layers given a neural network. Each nested function produces intermediate activations $y_i = f_i(w_i, y_{i-1})$, given weights $w_i$ and previous activations $y_{i-1}$, where $y_0 = x$ is the input. The weights are represented by the vector $\boldsymbol{w} = w_1, w_2, \ldots, w_p$, where each $w_{[1,\ldots,p]} \in \mathbb{R}^{[m_1,\ldots,m_p]}$. The total number of trainable parameter is $d = \sum_{i=1}^{p} m_i$. The intermediate activations are $\boldsymbol{y} = y_1, \ldots, y_p$. The final output is $y = y_p = f(\boldsymbol{w}, x) \in \mathbb{R}^q$, where typically $q \ll m_i, \forall i \in [p]$. The loss function $\mathcal{L}(y, \hat{y}) \in \mathbb{R}$ measures the difference between the predicted output $y$ and the true target values $\hat{y}$.

**Reverse-mode Auto Differentiation (RMAD).** RMAD computes gradients by propagating sensitivities (which is the rate at which the output of a function changes with respect to a given intermediate value) backward through the neural network. RMAD relies on vector-Jacobian product (`vjp`), where the *Jacobian* represents partial derivatives of an intermediate activation $y_i$ with respect to weights $w_{i-1}$, denoted $\frac{\partial y_i}{\partial w_{i-1}}$, and the *vector* is the activation gradient $\frac{\partial f}{\partial y_i}$. RMAD starts by setting $\frac{\partial f}{\partial y_p}$ to 1, and propagating $\frac{\partial f}{\partial w_{i-1}} = \frac{\partial f}{\partial y_i} \frac{\partial y_i}{\partial w_{i-1}}$ and $\frac{\partial f}{\partial y_{i-1}} = \frac{\partial f}{\partial y_i} \frac{\partial y_i}{\partial y_{i-1}}$, for $i \in [2, p]$, backwards. The final result is the weight gradient $\frac{\partial f}{\partial \boldsymbol{w}}$, formed from a series of `vjp` computations.

Backpropagation (Rumelhart et al., 1986) (BP) is a specific case of RMAD tailored for neural networks. While RMAD's backward pass begins by $\frac{\partial f}{\partial y_p}$ set to 1, BP initializes from the gradient of the loss function: $\frac{\partial \mathcal{L}}{\partial y_p}$, which provides a semantically meaningful signal for optimization. The backward phase is preceded by a forward pass that computes the activations and the loss $\mathcal{L}$.

**Forward-mode Auto Differentiation (FMAD).** FMAD propagates directional derivatives through the neural network to compute Jacobian-vector products (`jvp`). FMAD analyzes how a small perturbation $\boldsymbol{v}$ in the weights $\boldsymbol{w}$ affects the outputs. Starting from $\delta y_1 = \frac{\partial y_1}{\partial w_1} v_1$, FMAD propagates changes forward as:

$$\delta y_i = \frac{\partial y_i}{\partial w_i} v_i + \frac{\partial y_i}{\partial y_{i-1}} \delta y_{i-1}, \quad \text{for } i \in [2, p] \tag{1}$$

until the final scalar perturbation in the loss $\delta\mathcal{L}$ is computed. Here, the *Jacobian* term $\frac{\partial y_i}{\partial w_i}$ reflects sensitivity to weight changes, and the perturbation *vector* is $v_i \in \boldsymbol{v}$, where $\boldsymbol{v}$ is typically sampled from $\mathcal{N}(0, I_d)$. The scalar $\delta\mathcal{L}$ is referred to as the `jvp`. Weight gradients (also called forward gradients)

Table 2: Big $\mathcal{O}$ bounds of gradient computation methods on (a) Convergence error, (b) Memory consumption, and (c) Compute cost (per-iteration). Let $c$ denote the memory required to store activations for a single layer, $c_h$ the maximum per-layer activation memory, $p$ the total number of layers, and $d$ the number of trainable parameters. While BP with checkpointing retains the fast convergence of BP with additional memory savings; both FMAD and ZO methods suffer from worse convergence and higher compute costs, with parallel variants further increasing memory consumption.

| Method | Convergence Error | Memory | Compute |
|---|---|---|---|
| BP | $\mathcal{O}(1/T)$ with $\eta \le \frac{1}{L}$ | $\mathcal{O}(cp)$ | $\mathcal{O}(d)$ |
| BP (with checkpointing) | | $\mathcal{O}(c\sqrt{p})$ | $\mathcal{O}(d\log p)$ |
| FMAD (Parallel) | $\mathcal{O}\left(\frac{1}{T\left[1 - \frac{L\eta}{2}\left(1 + \frac{d+1}{n}\right)\right]}\right)$ with $\eta < \frac{2}{L\left(1 + \frac{d+1}{n}\right)}$ | $\mathcal{O}(nc_h)$ | $\mathcal{O}(nd)$ |
| FMAD (Sequential) | | $\mathcal{O}(c_h)$ | $\mathcal{O}(nd)$ |
| ZO (Parallel) | $\mathcal{O}\left(\frac{1}{T\left[1 - \frac{L\eta}{2}\left(1 + \frac{d+1}{n}\right)\right]}\right) + \frac{Ld\eta^2}{2n}\mathcal{O}(\epsilon^2)$ | $\mathcal{O}(nc_h)$ | $\mathcal{O}(nd)$ |
| ZO (Sequential) | with $\eta < \frac{2}{L\left(1 + \frac{d+1}{n}\right)}$ | $\mathcal{O}(c_h)$ | $\mathcal{O}(nd)$ |

are computed as $\frac{\partial \mathcal{L}}{\partial w_i} = \texttt{jvp} \cdot v_i$. In contrast to BP, which propagates $\frac{\partial \mathcal{L}}{\partial y_i}$ backward ($i$ from $p$ to 1), FMAD propagates $\frac{\partial y_i}{\partial w_j}$ forward ($i$ from 1 to $p$, for all $j$).

**Zero-order (ZO) Finite Differences.** ZO optimization estimates gradients using only function $f$ evaluations, with no first-order derivative information required. These methods, including finite differences (Richardson, 1955; Malladi et al., 2023), perturb the weights and approximate gradients through changes in the loss values of the perturbed function evaluations. Given a perturbation direction $\boldsymbol{v} \sim \mathcal{N}(0, I_d)$, the gradient with respect to $w_i$ is approximated via:

$$\frac{\partial \mathcal{L}}{\partial w_i} \approx \frac{\mathcal{L}(f(\boldsymbol{w} + \epsilon\boldsymbol{v}, x), \hat{y}) - \mathcal{L}(f(\boldsymbol{w} - \epsilon\boldsymbol{v}, x), \hat{y})}{2\epsilon} \cdot v_i, \tag{2}$$

where $\epsilon$ is a small step size. This symmetric difference estimator requires two sequential forward passes per perturbation direction.

## 3 Convergence, Memory, and Compute Trade-offs

We next review the theoretical characteristics of BP, FMAD, and ZO optimization, focusing on their convergence, memory, and computational profiles. These methods have been analyzed individually in prior works (Malladi et al., 2023; Gautam et al., 2024; Guo et al., 2025; Chen et al., 2019), as well as in classic results on BP (Bottou et al., 2018; Garrigos and Gower, 2024) and automatic differentiation. The derivations of convergence bounds on a non-convex function $f$, for the three gradient computation approaches studied in this work are shown in Appendix I. Analysis on computation complexity is in Appendix H. Here, we compile the theoretical results into a common comparative framework to highlight their trade-offs under shared assumptions.

Table 2 shows how convergence behavior is affected by key parameters, including the trainable parameter dimensionality $d$ and the number of perturbations per iteration $n$. Although FMAD and ZO can achieve memory savings in certain regimes, they incur higher per-iteration compute costs and slower convergence in high-dimensional or low-perturbation settings. In contrast, BP (especially when paired with activation checkpointing) retains favorable convergence with competitive memory efficiency. These theoretical results provide intuitions for our empirical analysis in §4, where we quantify how these trade-offs manifest in large-scale training. We summarize theoretical comparisons of BP, FMAD, and ZO along three key axes:

**Observation 1: Accuracy.** FMAD and ZO introduce approximation noise and discretization effects, leading to higher convergence error than BP, especially in high-dimensional models or with limited perturbations. §4.2 empirically demonstrates that ZO suffers greater accuracy degradation than FMAD due to discretization error, and that both ZO and FMAD achieve lower accuracy than BP because of additional learning rate constraints, which are detailed in Appendix C.

**Observation 2: Convergence Speed.** Both FMAD and ZO require stricter learning-rate constraints than BP, which slows convergence as dimensionality grows or perturbation budgets shrink. §4.3 supports this observation by showing that FMAD and ZO converge more slowly and reach lower accuracy compared to BP.

**Observation 3: Memory-compute Trade-offs.** While able to reduce activation memory, FMAD and ZO incur $\mathcal{O}(nd)$ compute cost per iteration, and face a fundamental trade-off: parallel perturbations reduce runtime but increase memory, whereas sequential perturbations conserve memory but slow training. §4.4 corroborates these memory bounds and shows a breakdown of the memory consumption. §4.5 empirically validates the computation cost.

**A Note on Non-differentiable and Black-box Settings.** While it's claimed that ZO has utility in settings with non-differentiable objectives (Qiu et al., 2023; Rando et al., 2023) or limited model access (Nikolakakis et al., 2022; Lobanov et al., 2024), its applicability to large-scale model training is fundamentally constrained. In true black-box scenarios, it is often infeasible to perturb weights or query the loss values, making ZO methods impractical. In contrast, first-order methods such as BP and FMAD require access to model internals and automatic differentiation support, challenges that are largely engineering in nature and increasingly well-supported by modern frameworks. As such, the growing trend (Gautam et al., 2024; Guo et al., 2025) of applying ZO to train LLMs is misguided: the computational cost and degraded convergence significantly outweigh the memory gains.

## 4 EMPIRICAL EVALUATION

This section empirically compares the variants of BP, FMAD, and ZO optimization. We evaluate these methods across multiple axes, including (a) accuracy, (b) wallclock convergence time, (c) memory consumption, and (d) computation cost. For each of these dimensions, we also examine how different variance reduction strategies affect performance. Last but not least, we empirically show that variance reduction methods and adaptive optimizers fail to make FMAD and ZO converge reliably.

### 4.1 EXPERIMENTAL SETTINGS

**Datasets.** We evaluate gradient computation methods across a diverse set of **5 text-based tasks** and **2 vision-based tasks**. The 5 text-based tasks are (a) Gsm8K (text generation on math problems/next-word prediction) (Cobbe et al., 2021), (b) MMLU (multiple-choice question-answering covering various domains of knowledge) (Hendrycks et al., 2021), (c) AGNews (4-class news article text classification task) (Zhang et al., 2015), (d) BoolQ (boolean question-answering) (Clark et al., 2019), and (e) MultiRC (closed-book question-answering) (Khashabi et al., 2018). The 2 vision-based tasks are both based on visual question-answering: (a) VQAv2 (Goyal et al., 2019), and (b) TextVQA (Singh et al., 2019). Appendix B describes the datasets in detail, including the train/test splits.

**Models.** Our evaluation uses 5 models with a varying number of total parameters (listed in parentheses). For text-based tasks, on the billion-parameters scale, we use LLAMA 3.1 (8B) (Grattafiori et al., 2024) and OPT (1.3B, 6.7B, 13B) (Zhang et al., 2022). Additionally, we include medium-sized language models BERT (110M, 340M) (Devlin et al., 2019) and ROBERTA (125M, 355M) (Liu et al., 2020) to analyze performance variations across model sizes. For vision-based tasks, we use QWEN 2 VL (7B) (Qwen et al., 2025). To finetune these models, we use QLORA (Dettmers et al., 2023), where low-rank adapters are trainable while the rest of the weights are frozen and quantized to 4 bits. By default, we set the LORA rank $r = 1$ and scale $\alpha = 1$ to minimize the number of trainable parameters for FMAD and ZO. Appendix F.4 reports results on higher LORA ranks.

**Methods for Comparison.** We categorize the 16 gradient computation methods our evaluation compares into three groups: (a) **Backpropagation Methods:** BP-VANILLA (Rumelhart et al., 1986) (the standard implementation that stores all intermediate activations), BP-CHECKPOINTING (Chen et al., 2016) (reduces peak memory consumption by storing only a subset of activations and recomputing the rest during the backward pass), and BP-ACCUMULATE (uses gradient accumulation). (b) **Zero-order Methods**: ZO-VANILLA (Chen et al., 2019) (use a single perturbation to estimate gradients as in Equation 2), MEZO (Malladi et al., 2023) (incorporates a prompt-finetuning approach to convert classification tasks into next-word prediction with a constrained vocabulary), ZO-ACCUMULATE (applies gradient accumulation to reduce noise in gradient estimates), ZO-MULTIPLE (Feng et al., 2024) (averages gradient estimates from multiple perturbations per iteration to improve estimate stability), ZO-ADAPTIVE (adaptively selects perturbation directions based on prior gradients), ZO-

Table 3: BP, FMAD, and ZO variant accuracies (higher is better) across models and datasets. Subscripts show accuracy gaps from BP-VANILLA/CHKPT (CHKPT = CHECKPOINTING). While BP remains the most accurate, FMAD and ZO variants -ACCUMULATE and -MULTIPLE offer notable gains over their -VANILLA forms but still lag behind BP, especially on generation tasks, such as GSM8K. Appendix F.1 reports variance across runs. Darker shade ▦ = range of high accuracies, lighter shade ▢ = range of moderate accuracies, unshaded = range of low accuracies.

| Model + Dataset | LLAMA 3.1 (8B) | | | | | QWEN 2 VL (7B) | |
| --- | --- | --- | --- | --- | --- | --- | --- |
| Method | AGNews | BoolQ | MultiRC | GSM8K | MMLU | VQAv2 | TextVQA |
| No Finetuning | 23.5 | 51.6 | 52.8 | 27.3 | 51.1 | 73.2 | 71.1 |
| BP-VANILLA/CHKPT | 94.2 | 88.3 | 85.2 | 54.3 | 60.3 | 87.1 | 98.5 |
| BP-ACCUMULATE | 93.8$_{(-0.4)}$ | 87.9$_{(-0.4)}$ | 83.3$_{(-1.9)}$ | 33.1$_{(-21.1)}$ | 53.8$_{(-6.4)}$ | 86.3$_{(-0.7)}$ | 97.1$_{(-1.4)}$ |
| ZO-VANILLA | 73.6$_{(-20.6)}$ | 57.1$_{(-31.1)}$ | 57.2$_{(-28.0)}$ | 36.3$_{(-17.9)}$ | 54.7$_{(-5.6)}$ | 77.6$_{(-9.4)}$ | 72.9$_{(-25.6)}$ |
| ZO-ACCUMULATE | 85.8$_{(-8.4)}$ | 60.9$_{(-27.3)}$ | 60.3$_{(-24.8)}$ | 28.0$_{(-26.3)}$ | 55.2$_{(-5.1)}$ | 79.7$_{(-7.4)}$ | 73.1$_{(-25.4)}$ |
| ZO-MULTIPLE | 86.7$_{(-7.4)}$ | 60.0$_{(-28.2)}$ | 61.0$_{(-24.1)}$ | 35.8$_{(-18.5)}$ | 56.8$_{(-3.4)}$ | 81.5$_{(-5.6)}$ | 74.7$_{(-23.8)}$ |
| ZO-ADAPTIVE | 81.5$_{(-12.7)}$ | 57.4$_{(-30.9)}$ | 59.0$_{(-26.1)}$ | 30.2$_{(-24.1)}$ | 52.6$_{(-7.6)}$ | 79.5$_{(-7.5)}$ | 79.1$_{(-19.3)}$ |
| ZO-SVRG | 84.7$_{(-9.5)}$ | 62.6$_{(-25.7)}$ | 61.2$_{(-23.9)}$ | 32.1$_{(-22.2)}$ | 55.9$_{(-4.3)}$ | 79.1$_{(-7.9)}$ | 72.9$_{(-25.6)}$ |
| ZO-SPARSE | 64.5$_{(-29.6)}$ | 53.2$_{(-35.1)}$ | 55.3$_{(-29.8)}$ | 29.1$_{(-25.1)}$ | 51.4$_{(-8.9)}$ | 78.6$_{(-8.5)}$ | 73.8$_{(-24.7)}$ |
| MEZO | 80.5$_{(-13.7)}$ | 58.2$_{(-30.1)}$ | 60.4$_{(-24.8)}$ | — | — | — | — |
| FMAD-VANILLA | 80.5$_{(-13.7)}$ | 60.7$_{(-27.6)}$ | 61.4$_{(-23.8)}$ | 37.7$_{(-16.6)}$ | 55.8$_{(-4.5)}$ | 82.3$_{(-4.8)}$ | 78.3$_{(-20.2)}$ |
| FMAD-ACCUMULATE | 88.0$_{(-6.2)}$ | 70.3$_{(-17.9)}$ | 71.2$_{(-14.0)}$ | 30.8$_{(-23.5)}$ | 57.1$_{(-3.1)}$ | 83.7$_{(-3.4)}$ | 80.9$_{(-17.6)}$ |
| FMAD-MULTIPLE | 86.2$_{(-8.0)}$ | 64.4$_{(-23.8)}$ | 65.4$_{(-19.7)}$ | 40.5$_{(-13.8)}$ | 57.7$_{(-2.6)}$ | 82.9$_{(-4.2)}$ | 79.1$_{(-19.4)}$ |
| FMAD-ADAPTIVE | 78.5$_{(-15.7)}$ | 56.4$_{(-31.9)}$ | 58.2$_{(-27.0)}$ | 38.1$_{(-16.2)}$ | 56.3$_{(-3.9)}$ | 82.9$_{(-4.1)}$ | 78.2$_{(-20.3)}$ |
| FMAD-SVRG | 82.5$_{(-11.7)}$ | 64.6$_{(-23.7)}$ | 64.1$_{(-21.0)}$ | 35.4$_{(-18.9)}$ | 56.1$_{(-4.2)}$ | 83.0$_{(-4.0)}$ | 79.5$_{(-19.0)}$ |
| FMAD-SPARSE | 70.4$_{(-23.8)}$ | 56.9$_{(-29.4)}$ | 53.1$_{(-32.1)}$ | 30.3$_{(-23.9)}$ | 53.4$_{(-6.8)}$ | 80.3$_{(-6.7)}$ | 77.0$_{(-21.5)}$ |

SVRG (Liu et al., 2018) (applies stochastic variance reduction to correct noisy gradients), and ZO-SPARSE (Guo et al., 2025) (only updates top-1% parameters each iteration). (c) **Forward-mode AD Methods:** FMAD-VANILLA (Baydin et al., 2022), FMAD-ACCUMULATE, FMAD-MULTIPLE, FMAD-ADAPTIVE, FMAD-SVRG, and FMAD-SPARSE. The -VANILLA suffix denotes the original implementation according to Equation 1, while the other variants mirror the corresponding ZO method in (b), adapting similar strategies for the forward-mode setting. Appendix C describes these methods and their hyperparameters in detail.

**Metrics.** We evaluate the efficiency of the gradient computation methods using four metrics. (a) **Accuracy** at test-time assesses the efficacy of the learned models. (b) **Wallclock convergence time** (in minutes) measures the time each approach takes to achieve stable-state accuracy. (c) **Peak memory consumption** (in GBs) quantifies the maximum memory consumed during training. (d) **Computation cost** for each iteration and until convergence, in Tera Floating-Point Operations per Second (TFLOPs). Additionally, in our failure mode analysis, for ZO and FMAD approaches, Section 4.6 reports statistics, such as the mean of effective gradient norms and `jvp` values across iterations, capturing the instability of estimated gradients and its impact on optimization dynamics.

**Libraries and Hardware.** Our codebase is built using PyTorch (Paszke et al., 2019). Quantization uses AutoGPTQ (Frantar et al., 2022). We conducted all experiments involving billion-scale models across ZO and FMAD variants on a single Nvidia L40 GPU (48GB RAM). For experiments on OPT (13B) model, we used one Nvidia A100 (80GB RAM). For BERT and ROBERTA models, we used an Nvidia 2080ti (11GB RAM). We repeated each experiment three times with random seeds set to 0, 1, and 2 to ensure consistency and robustness. Our source code is available for replication [1].

## 4.2 COMPARISON ON ACCURACY

Accuracy is the primary metric of interest since any gradient computation method that reduces memory consumption or computational cost is of little practical value if it cannot match the predictive performance of BP. Table 3 presents accuracy results and Appendix F.1 shows variance across 3 executions.

**Backpropagation achieves significantly higher accuracy than FMAD-VANILLA and ZO-VANILLA.** Backpropagation, both in its standard form (BP-VANILLA) and with checkpointing (BP-CHKPT), consistently achieves the highest accuracy across all tasks. Among the alternatives, the

---
[1] https://anonymous.4open.science/r/Gradient_Estimation

-VANILLA forms of FMAD and ZO are most directly comparable to BP. Due to the inherent randomness in their perturbation-based gradients (see §3 Obs 1), both FMAD-VANILLA and ZO-VANILLA lag behind BP by 4.5–27.5% and 5.6–31.1% across datasets, respectively. Further, across all the datasets, FMAD-VANILLA outperforms ZO-VANILLA, with gains of 1.1–6.9%. This consistent margin illustrates FMAD's fundamental advantage: access to analytic first-order Jacobian-vector products (`jvp`), over ZO's reliance on noisy finite-difference estimates.

**Variance reduction approaches improve the accuracy of FMAD and ZO methods yet fall short of closing the gap with BP methods.** Both FMAD and ZO benefit from their -ACCUMULATE and -MULTIPLE variants, which reduce gradient noise by trading off higher compute or memory. FMAD-ACCUMULATE improves over FMAD-VANILLA by 1.4–9.8% across datasets, except on GSM8K (-6.9%), likely due to its need for smaller batch sizes. Similarly, ZO-ACCUMULATE boosts accuracy by 0.2–12.2%, with an 8.3% drop on GSM8K. FMAD-MULTIPLE improves by 0.6–5.7%, and ZO-MULTIPLE by 0.2–13.2%, with only a 0.5% drop on GSM8K.

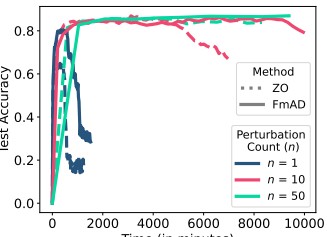 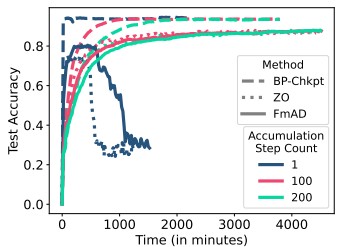

(a) Averaging gradients over multiple perturbations per iteration

(b) Accumulating and averaging gradients across iterations

Figure 2: Experiments (on AGNews dataset with LLAMA 3.1 8B model) with varying perturbation counts (-MULTIPLE) and accumulation steps (-ACCUMULATE) show that both strategies reduce gradient noise and improve convergence stability for FMAD and ZO. However, -MULTIPLE increases memory and compute costs, while -ACCUMULATE slows down convergence. Furthermore, as shown in (*right*), BP-CHECKPOINTING achieves the highest, most stable accuracy fastest. FMAD performs moderately but is unstable or slow, and ZO (with step size 1) suffers early collapse and fails to match BP's accuracy.

To understand the effects of these two variance reduction techniques, we vary the number of perturbations and accumulation steps. Figure 2a shows that increasing perturbation count ($n = 10, 50$) yields 5.7–7.7% (FMAD) and 13.2–14.0% (ZO) accuracy gains on AGNews, consistent with the observations of § 3. Similarly, Figure 2b shows that increasing accumulation steps (100, 200) yields 7.5–7.6% (FMAD) and 12.2–14.0% (ZO) gains. These improvements come at the cost of increased convergence time (sequential implementation of -MULTIPLE), memory (parallel implementation of -MULTIPLE), or slower updates (-ACCUMULATE). These trade-offs are discussed in §4.3 and 4.4.

**Other variance reduction approaches offer limited or inconsistent accuracy improvements for FMAD and ZO.** -ADAPTIVE often underperforms, with FMAD-ADAPTIVE trailing -VANILLA on BoolQ (-4.3%) and MultiRC (-3.2%), likely due to biased updates from gradient-informed perturbation sampling. -SPARSE performs worst overall, lagging -VANILLA by 1.2–10.1% (FMAD) and 1.9–9.0% (ZO), as random perturbations of early steps mislead saliency-based parameter selection. -SVRG improves classification accuracy by 4.0–11.1%, but failing on GSM8K (-4.2%) due to homogenized updates that weaken variance correction (see Appendix F.5.2). MEZO offers modest gains (1.0–6.9%) on classification but lacks applicability to generative and vision-language tasks.

**Accuracy gaps widen as trainable parameters or model size increases.** To further evaluate the impact of trainable parameter count on FMAD and ZO, we conducted additional experiments on medium-sized models (110M–350M parametered BERT and ROBERTA) and large-sized models (OPT 1.3B, 6.7B, with various LORA ranks, and 13B), as detailed in Appendices F.2 and F.4. Both FMAD and ZO still exhibit slower convergence and degraded performance compared to BP, with the gap widening as model size increases (especially in the case of BERT and ROBERTA). Experiments on changing the perturbation variance are presented in Appendix F.3.

### 4.3 COMPARISON ON WALLCLOCK CONVERGENCE TIME

Convergence time determines how quickly a trained model becomes feasible for practical use. We contextualize our analysis using Figure 2, which illustrates the time-to-accuracy curve of the -VANILLA methods. Figure 4 in Appendix F.1 includes results on the remaining datasets.

**Compared to ZO and FMAD, BP-CHECKPOINTING achieves the fastest convergence speed and highest accuracy.** Figure 2b (AGNews, batch size 40) compares test accuracy against wall-clock time. Since BP-VANILLA runs out of memory at this batch size, we instead report a smaller batch size (8) for a fair runtime comparison between BP-VANILLA and BP-CHECKPOINTING. At batch size 8, BP-VANILLA requires 804.4s/iter, while BP-CHECKPOINTING takes 936.3s/iter (∼1.2× slower per iteration). Despite this overhead, BP-CHECKPOINTING still outperforms FMAD by ∼1.2× per iteration and achieves 4.5–27.6% higher accuracy.

At batch size 40, where memory is the limiting factor, BP-CHECKPOINTING converges reliably with 1112.8s/iter. In comparison, FMAD requires 1286.5s/iter, and ZO is the fastest at 726.7s/iter (∼1.5× faster than BP-CHECKPOINTING). However, this runtime advantage does not translate to accuracy: BP-CHECKPOINTING reaches ∼94% accuracy, while FMAD and ZO fall short due to slower convergence and instability. Specifically, ZO suffers from approximation errors in gradient estimation, leading to accuracy degradation of 5.6–31.1% relative to BP-based methods.

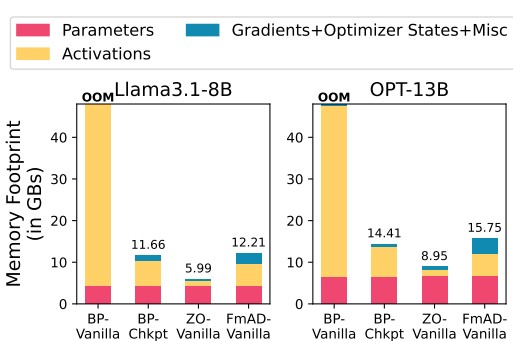

Figure 3: Breakdown of memory consumption of training (*left*) LLAMA 3.1 (8B) and (*right*) OPT (13B) models on AGNews. Although BP-CHECKPOINTING is 1.6–1.9× takes more memory than ZO, it takes far fewer iterations to achieve 4.5–31.1% higher accuracy (as shown in Figure 2b).

In terms of overall time-to-accuracy, BP-CHECKPOINTING achieves convergence 21.4–34.8% faster than alternatives. The gap with FMAD arises from its computational inefficiency: unlike BP, which reuses downstream gradients with a single matrix multiplication per layer, FMAD requires two matrix multiplications per layer for `jvp` evaluation (Eq. 1).

**Variance reduction improves convergence, but often slows down convergence time.** As shown in Figure 2a, -MULTIPLE variants (e.g., with $n = 10, 50$) yield smoother training and higher accuracy than their $n = 1$ counterparts. However, these gains come with a proportional increase in convergence time for sequential implementations as the runtime scales linearly with the number of perturbations. -ACCUMULATE variants (Figure 2b) improve accuracy without increasing per-iteration cost, as they amortize single-step estimates over multiple updates. However, the delay in parameter updates slows down overall convergence: with an accumulation window of 200, training is 14.8× and 10.9× slower for FMAD and ZO, respectively, than when trained without accumulation.

4.4   COMPARISON ON MEMORY CONSUMPTION

**Memory savings from FMAD and ZO come at the cost of accuracy and convergence speed.** Figure 3 shows that both FMAD and ZO reduce memory usage relative to BP-VANILLA, which runs out of memory (OOM) due to storing all activations. By contrast, FMAD and ZO store only the previous layer's activation, yielding a lower memory footprint. However, as seen in Figure 2b, these savings lead to significantly longer training times and degraded model performance. Meanwhile, BP-CHECKPOINTING uses 0.6–1.3GB less memory than FMAD, while delivering substantially faster convergence and 4.5–31.1% higher accuracy. Further, FMAD consumes 3.3–4.3× more activation memory than ZO. This overhead stems from the need to simultaneously store previous layer's intermediate activations for both the primary forward pass and the additional `jvp` computation.

**Variance reduction strategies introduce memory-accuracy trade-offs.** The -MULTIPLE variants improve gradient quality by evaluating multiple perturbations per step, but parallel implementations require linearly more memory. For instance, if one forward pass needs 1.26GB (ZO) or 5.41GB (FMAD) for activations, using $n$ perturbations inflates this to 1.26$n$GB or 5.41$n$GB, respectively. On the other hand, -ACCUMULATE amortizes these computations over time and introduces no memory overhead, though at the cost of slower convergence.

4.5   COMPARISON ON COMPUTATION COST

Computation cost in terms of FLOPS directly impacts energy consumption and determines whether training large models is feasible under given resources. Table 4 reports both per-iteration cost and total cost until convergence. (Table 2 summarized the theoretical bounds.)

**FMAD and ZO methods reduce per-iteration compute costs but incur significantly higher total compute due to slow convergence.** ZO-VANILLA incurs a relatively low per-iteration cost of 288.7 TFLOPs, approximately $0.7\times$ the cost of BP-CHECKPOINTING, because it only requires two forward passes per gradient estimate. However, this advantage is misleading: due to slow convergence, its total computation cost *until convergence* is $3.8\times$ higher than that of BP-CHECKPOINTING. FMAD-VANILLA shows a per-iteration cost nearly identical to BP-CHECKPOINTING, but its convergence is hindered by gradient estimates with high variance, leading to $3.2\times$ higher total compute costs.

**Multiple perturbations per iteration improves accuracy but linearly increases cost.** In ZO-MULTIPLE, using 10 perturbations per iteration leads to a $9.7\times$ increase in compute, showcasing the linear relationship between the number of perturbations and cost. In contrast, ZO-ACCUMULATE, which accumulates gradients across iterations without increasing perturbation count, maintains similar cost to ZO-VANILLA but still suffers from slow convergence. Similarly, for FMAD, when we increase the number of perturbations by $10\times$ to reduce gradient variance and improve accuracy, the cost increases by $20\times$ that of BP, as each `jvp` involves two matrix multiplications.

Table 4: Computation cost per iteration and until convergence (lower is better) for LLAMA 3.1 (8B) on AGNews dataset. BP-CHECKPOINTING remains by far the most compute-efficient; whereas the perturbation-based methods (ZO and FMAD), even their -ACCUMULATE variants, incur order-of-magnitude more TFLOPs to reach convergence.

| Method | TFLOPs per Iter. ($\downarrow$) | TFLOPs until Convergence ($\downarrow$) | # Iter. until Convergence |
|---|---|---|---|
| BP-CHECKPOINTING | 434.4 | $65.2 \times 10^4$ | $1.5 \times 10^3$ |
| ZO-VANILLA | 288.7 | $251.2 \times 10^4$ | $8.7 \times 10^3$ |
| ZO-MULTIPLE | 2886.8 | $2425.0 \times 10^4$ | $8.4 \times 10^3$ |
| ZO-ACCUMULATE | 288.7 | $2165.1 \times 10^4$ | $75.0 \times 10^3$ |
| FMAD-VANILLA | 432.0 | $207.4 \times 10^4$ | $4.8 \times 10^3$ |
| FMAD-MULTIPLE | 4320.3 | $4147.5 \times 10^4$ | $9.6 \times 10^3$ |

### 4.6 FAILURE MODE ANALYSIS

Here, we analyze why variance reduction methods and adaptive optimizers sometimes fail to make FMAD and ZO converge reliably.

**Cascading JVP Amplification with Adaptive Optimizers.** A key failure mode in FMAD arises with adaptive optimizers, such as ADAMW, triggering cascading amplification of Jacobian-vector products (`jvp`). On GSM8K, `jvp` magnitudes remain stable under SGD within $[-50, 50]$, but spike $8$–$10\times$ under ADAMW (Figure 9). These spikes produce large gradient updates, inflating weights and further amplifying `jvp` values, a positive feedback loop that can cause divergence or noisy updates.

**Gradient Variance and Magnitude Explains Performance Drops.** Effective gradient variance under ADAMW is $4$–$6\times$ higher than SGD, with peaks of $200$–$400$ in hidden layers of the LLaMA-7B subset. This instability correlates with $2$–$5\%$ lower final accuracy vs. BP with checkpointing, and some runs yield `NaN` gradients. Spikes typically appear after $50$–$100$ iterations, indicating accumulation from the rolling-average mechanism in adaptive optimizers.

**Non-Adaptive SGD Maintains Stability.** In contrast, SGD keeps `jvp` bounded and gradients closely track backpropagation, producing stable convergence (Figures 9a, 9b). These results highlight a critical interaction between optimizer choice and FMAD stability: adaptive optimizers can introduce harmful gradient artifacts in FMAD and ZO methods. Further details, including additional datasets, layer-wise analyses, and variance-reduction strategies, are provided in Appendix F.5.

## 5 CONCLUSION

While forward-mode AD (FMAD) and zero-order (ZO) optimization have been proposed as memory-efficient alternatives to backpropagation (BP), prior work lacked comparison with checkpointed BP and unified theoretical bounds. Our analysis closes these gaps, revealing that FMAD and ZO incur higher computational cost, slower convergence, and greater sensitivity to dimensionality and perturbation budgets. Even with enhancements like variance reduction, they remain less efficient and robust than BP with activation checkpointing. Empirical results on large models confirm that checkpointed BP consistently outperforms FMAD and ZO across accuracy, convergence speed, and compute cost – at comparable memory usage. These findings reaffirm checkpointed BP as the most practical strategy for memory-constrained training and clarify the limitations of FMAD and ZO.

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

# A  RELATED WORK

Here we review recent works on forward-mode AD and zero-order optimization methods; along with a discussion on various methods to refine the memory and time efficiency of backpropagation.

## A.1  FORWARD-MODE AD

The application of forward-mode automatic differentiation (FMAD) for training deep neural networks was first introduced in Forward-Gradient Descent (FGD) (Baydin et al., 2022), building on an earlier survey on automatic differentiation (Baydin et al., 2017). FGD demonstrated FMAD on a small-scale three-layer fully connected model and a four-layer convolutional network, claiming that FMAD can outperform backpropagation (BP) in speed and loss reduction per unit time. However, these claims remain unverifiable, as the implementation was never made publicly available, and subsequent independent evaluations (Belotti and Angioni, 2023) have found these results difficult to reproduce.

Beyond this initial demonstration, more recent efforts have attempted to improve FMAD's efficiency. *Can Forward Gradients Match Backpropagation?* (Fournier et al., 2023) seeks to enhance FMAD by generating more structured perturbations rather than relying on random sampling. This approach introduces local losses computed via small auxiliary networks to inform perturbation choices. However, training these auxiliary networks significantly increases memory consumption and computational overhead, undermining FMAD's intended efficiency advantage.

Other studies have focused on extending FMAD beyond first-order gradients. *Second-order FmAD* (Cobb et al., 2024) provides a formal framework for computing second-order gradients with FMAD, demonstrating improved optimization performance. However, this comes at a substantial computational cost, and experiments remain limited to small-scale benchmarks (e.g., a CNN with only 431K parameters), leaving open the question of whether second-order FMAD can scale competitively against BP. Similarly, Taylor-mode Auto Differentiation (Bettencourt et al., 2019) generalizes FMAD to compute higher-order gradients, yet the memory and time-to-convergence trade-offs compared to BP remain unexplored.

Several other works have proposed variations of FMAD without fundamentally addressing its inefficiencies. *Randomized Forward Gradient-based GD* (Shukla and Shin, 2023) provides a convergence analysis of FMAD using random perturbations but offers no new insights into its computational efficiency. PROJECTED-FG (Rostami and Kia, 2024) applies FMAD to memory-efficient Frank-Wolfe optimization but evaluates only small models, making its conclusions inapplicable to large-scale deep learning. *Beyond Backpropagation* (Flügel et al., 2024) investigates the use of multiple perturbations per iteration to improve forward-gradient computation but fails to identify why FMAD remains inferior to BP in practice.

A more recent large-scale application of FMAD appears in SPRY (Panchal et al., 2024), which employs FMAD for fine-tuning large models (ranging from 100K to 13B parameters) in a federated learning setting. By restricting each client to a small subset of weights, SPRY circumvents FMAD's poor performance in high-dimensional perturbations. However, even in this setting, FMAD exhibits slower convergence and higher variance than BP, further reinforcing its fundamental limitations.

For applications besides LLM training or finetuning, biological plausibility (Schmidt-Hieber, 2023; Xiao et al., 2024) has been proposed as a motivating factor for exploring alternative gradient estimation techniques. FMAD avoids the backward signal transport required by backpropagation and has therefore been considered more biologically plausible. Though FMAD still relies on first-order derivatives and engineered automatic differentiation, which limits its direct applicability to biological systems.

While prior works have demonstrated narrow successes of FMAD in specialized scenarios, none have systematically analyzed its fundamental computational constraints. Besides, the comparison of FMAD against a strong baseline of BP-CHECKPOINTING remains uncharted. Unlike these related studies, our work provides a principled theoretical and empirical investigation into the scalability bottlenecks of FMAD, explicitly comparing its memory and time complexity against BP-CHECKPOINTING. We also uncover failure modes of FMAD in deep networks, offering new insights into why it cannot consistently surpass BP in terms of both time-to-convergence and efficiency.

## A.2 ZERO-ORDER OPTIMIZATION

Zero-order (ZO) optimization has received significant attention, particularly in settings where first-order gradient information is unavailable or impractical to compute. Unlike FMAD, which has seen limited large-scale adoption, ZO methods have been actively explored in deep learning due to their applicability in scenarios such as adversarial attacks, black-box optimization, and gradient-free fine-tuning. Similar to FMAD, we also note that none of the works discussed below have made a comparison of their ZO-based variant against BP-CHECKPOINTING, an aspect which is fleshed out in this work.

MEZO (Malladi et al., 2023) and its extension MEZO-SVRG (Gautam et al., 2024) introduced memory-efficient ZO optimization strategies that regenerate random perturbations instead of storing them, effectively reducing memory overhead. These methods have demonstrated practical advantages in fine-tuning large language models (LLMs) for classification tasks without requiring explicit backpropagation. While they address memory constraints, they do not provide insights into the fundamental efficiency trade-offs between ZO and BP in terms of time-to-convergence, memory consumption, and attained accuracy; which are the key concerns of our work. A closely related line of work is HIZOO (Zhao et al., 2025), which proposes a forward-only second-order ZO optimizer that uses Hessian-informed perturbations to accelerate MeZO-style fine-tuning. While HIZOO successfully demonstrates reduced activation-memory usage relative to MEZO, its evaluation focuses primarily on memory rather than wall-clock convergence time or total compute cost—metrics that are central to our analysis. Moreover, the algorithm introduces additional second-order computations (via Hessian-related estimators), whose overhead is not thoroughly quantified.

Expanding on these efforts, DEEPZERO (Chen et al., 2024) proposed a ZO deep learning framework capable of training deep neural networks from scratch. By leveraging coordinate-wise gradient estimation (CGE) over randomized vector-wise estimation, DEEPZERO achieves improved accuracy and computational efficiency. Additionally, the introduction of sparsity-induced training, feature reuse, and forward parallelization brings ZO training closer to first-order methods, achieving state-of-the-art results on ResNet-20 trained on CIFAR-10. However, despite these advancements, ZO remains fundamentally limited by high variance and inefficient gradient estimation, resulting in slower convergence compared to BP, which is an issue we empirically validate in our benchmarks.

Other works, such as DZOVR (Chen et al., 2023) and ZO-SVRG (Liu et al., 2018), have attempted to improve ZO efficiency by incorporating Stochastic Variance Reduced Gradients (SVRG) (Johnson and Zhang, 2013). Similarly, research on ZO methods for non-convex and non-smooth optimization (Liu et al., 2024; Kornowski and Shamir, 2024; Balasubramanian and Ghadimi, 2018) has provided valuable theoretical insights. However, none of these studies systematically compare ZO to BP in terms of memory consumption, execution time, and scalability, leaving open the question of whether ZO can ever be a viable alternative. Our work explicitly addresses this gap by benchmarking these methods against BP and highlighting their structural inefficiencies.

Further, ZO-ADAMM (Chen et al., 2019) integrates an adaptive optimizer (ADAMM) into ZO, demonstrating improved stability. However, even with adaptive optimization, ZO struggles to match the convergence speed of BP, as shown in their experiments on a small-scale CNN. Additionally, work on ZO optimization in high-dimensional settings (Wang et al., 2018) has focused primarily on convergence properties rather than the computational and memory efficiency bottlenecks that limit ZO's practical scalability.

*Revisiting ZO* (Zhang et al., 2024) benchmarks the performance of large language models trained using BP, FMAD, and ZO optimization. Our work differs in several key ways: (a) We include comparisons against a backpropagation with checkpointing baseline, offering new insights into the memory-efficiency trade-offs among gradient computation methods. (b) Unlike *Revisiting ZO*, our study evaluates both time-to-convergence and overall computational cost, which are critical for understanding practical scalability. (c) We also provide an in-depth failure mode analysis, focusing on the behavior of Jacobian-vector products and their influence on model updates, an aspect unexplored in their work.

We note that like ZO with LLMs, ZO for biological systems (Schmidt-Hieber, 2023) would face scalability and convergence challenges when applied to high-dimensional models. Our study does not aim to contest the conceptual motivations behind these techniques; rather, we show their practical

limitations, in terms of computational cost and optimization performance, for large-scale models like LLMs.

### A.3 OPTIMIZATIONS ON BACKPROPAGATION

Backpropagation (BP) remains the dominant method for training deep neural networks due to its computational efficiency and well-optimized implementations. However, standard BP incurs high memory costs, as it requires storing intermediate activations for the entire computational graph during the forward pass. This limitation has motivated extensive research into memory-efficient variants of BP that aim to reduce memory consumption without significantly compromising training speed.

Checkpointing-based methods, such as REVERSIBLE RESIDUAL NETWORKS (Gomez et al., 2017) and ACTIVATION CHECKPOINTING (Chen et al., 2016), trade memory for recomputation by strategically discarding and later recomputing activations. These techniques have proven effective in reducing memory overhead, but they introduce additional computational costs. More recent approaches, such as EFFICIENT REMATERIALIZATION (Gruslys et al., 2016) and DYNAMIC PROGRAMMING-BASED ACTIVATION OFFLOADING (Beaumont et al., 2021), attempt to optimize checkpointing strategies to minimize recomputation overhead. Despite these advances, BP with checkpointing still follows the same fundamental backpropagation framework and benefits from computation reuse – an efficiency advantage that FMAD and ZO methods lack.

## B  DATASETS

In this section, we provide detailed descriptions of the datasets used in our experiments. For each dataset, we outline its origin, licensing, the version we have used, and task-specific characteristics, including the number of samples, sequence lengths and relevant domain or classification details.

**AGNews.**  The AG News dataset (Zhang et al., 2015) is derived from a corpus of 496,835 labeled news articles collected from over 2,000 web-based news sources published between 2004 and 2005. For this work, we use a widely adopted, cleaned, and balanced subset comprising 120,000 training samples and 7,600 test samples, evenly distributed across four categories: World, Sports, Business, and Science/Technology. We divide the test data into half to create validation and test splits. The dataset is primarily used for topic classification, which is also the focus of our study. The maximum sequence length for our experiments is set to 350 tokens during training. It is released under the Creative Commons CC0 1.0 Universal license, placing it in the public domain. We obtained the dataset via the Hugging Face Datasets library (Tunstall, 2022).

**BoolQ.**  The Boolean Questions (BoolQ) dataset (Clark et al., 2019) is a reading comprehension benchmark consisting of naturally occurring yes/no questions. Each instance includes a question, a passage (typically a paragraph from Wikipedia), and a binary answer ("yes" or "no") derived from the passage content. Unlike synthetic question-generation benchmarks, BoolQ features real user queries collected from Google search logs, making the task more reflective of real-world comprehension. The dataset contains approximately 9,427 question-passage training pairs, and 3,270 validation pairs. We divide the validation data into half to create the validation and test data splits for this work. The maximum sequence length for our experiments is set to 1200 tokens during training. BoolQ is released under the Creative Commons Share-Alike 3.0, which allows for flexible use, modification, and redistribution with appropriate attribution. Once again, Hugging Face Datasets was used to access BoolQ (del Moral, 2022a).

**MultiRC.**  The Multi-Sentence Reading Comprehension (MultiRC) (Khashabi et al., 2018) dataset is a benchmark corpus designed to evaluate machine reading comprehension over short paragraphs. Each example consists of a paragraph followed by one or more questions, with corresponding candidate answers that must be inferred from the text. In our setup, we frame the task as a binary classification problem, determining whether a given question-answer pair is correct or incorrect based on the paragraph content. The dataset contains approximately 6,000 multi-sentence questions drawn from over 800 distinct paragraphs. The maximum sequence length for our experiments is set to 1500 tokens during training. MultiRC is released under the MIT License, permitting broad use and redistribution with attribution. We accessed the dataset through Hugging Face (del Moral, 2022b).

**GSM8K.**   The Grade School Math 8K (GSM8K) dataset (Cobbe et al., 2021) is a high-quality benchmark for evaluating arithmetic reasoning and problem-solving abilities of language models. Each example consists of a single math word problem followed by a detailed, step-by-step answer. Designed to emphasize multi-step reasoning, the problems are written in natural language and reflect concepts typically found in grade school (middle school) curricula. The dataset contains 7,470 training examples and 1,319 test examples, all manually curated for clarity and correctness. The maximum sequence length for our experiments is set to 800 tokens during training. In this work, we use GSM8K as a text-to-text supervised learning task, where the input is the problem statement and the target is the final answer (without the reasoning steps). The dataset is publicly available under the MIT License, allowing broad reuse and modification with attribution. The dataset is available on Hugging Face (del Moral, 2022c).

**MMLU.**   The Massive Multitask Language Understanding (MMLU) (Hendrycks et al., 2021) dataset is a comprehensive benchmark designed to assess general knowledge and reasoning ability across a wide range of academic and professional subjects. It covers 57 diverse topics, including mathematics, history, law, medicine, and the sciences, with questions derived from standardized exams and expert-written materials. Each example is a multiple-choice question with four answer options, requiring both factual knowledge and reasoning skills. All four answer options are included in the prompt. The dataset consists of 99.8k training samples, 1.5k validation samples, and 14k test samples. The maximum sequence length for our experiments is set to 1500 tokens during training. MMLU is publicly available under the MIT License, allowing free use, modification, and distribution with appropriate credit. Its breadth and difficulty make it a challenging benchmark for evaluating finetuned language models. In line with rest of the datasets, we have used the Hugging Face Datasets version of MMLU (Phan, 2024).

**VQAv2.**   The Visual Question Answering v2.0 (VQAv2) (Goyal et al., 2019) dataset is a large-scale benchmark designed to evaluate a model's ability to understand and reason over both visual and textual inputs. Each example consists of an image (sourced primarily from the MS COCO dataset (Lin et al., 2014)) paired with a natural language question, and the task is to generate an accurate, typically short (often single-word), answer based on the visual content of the image.

VQAv2 addresses the language bias issues present in its predecessor (VQAv1) by ensuring that each question is associated with multiple images, such that the correct answer varies depending on the visual context. This structure encourages models to genuinely integrate image understanding rather than relying solely on question priors.

The dataset contains 443,757 training questions, 214,354 validation questions, and 447,793 test questions, associated with over 200,000 images. Each question has 10 human-provided answers, allowing for nuanced evaluation metrics such as accuracy based on answer consensus (Goyal, 2017). The maximum sequence length for our experiments is set to 100 tokens during training. VQAv2 is distributed under the 2-Clause BSD License, allowing for use and adaptation with attribution. We access the dataset through the VisualQA website (Goyal, 2017).

**TextVQA.**   The TextVQA (Text-based Visual Question Answering) dataset (Singh et al., 2019) is a vision-language benchmark specifically designed to evaluate a model's ability to read and reason about text within images. Unlike standard VQA tasks that focus on general object and scene understanding, TextVQA centers on questions where the answer relies on text present in the image itself; such as signs, labels, documents, product packaging, and storefronts.

Each example in the dataset includes an image, a natural language question, and a free-form textual answer. To correctly answer a question, models must integrate visual understanding with OCR (Optical Character Recognition) capabilities. TextVQA challenges systems to perform multimodal reasoning that spans spatial, linguistic, and visual modalities.

The dataset consists of approximately 28,408 questions associated with 14,987 images, split into: 21,953 training questions; 3,166 validation questions; and 3,289 test questions. Each question is annotated with 10 answers from human annotators to support consensus-based evaluation metrics. The maximum sequence length for our experiments is set to 100 tokens during training. TextVQA is publicly available under the CC BY 4.0 (Creative Commons Attribution 4.0 International License),

allowing flexible use, sharing, and adaptation with attribution. The dataset is available for access on Hugging Face (Preet, 2022).

## C  BASELINES AND HYPERPARAMETERS

**BP-VANILLA.**  This baseline (Rumelhart et al., 1986) uses a standard implementation of the training loop with backpropagation as the gradient computation method, without any modifications or enhancements. Due to out-of-memory (OOM) issues encountered with larger batch sizes, most experiments involving BP-VANILLA are conducted using smaller batches. Table 5 lists the hyperparameters.

Table 5: Hyperparameters related to BP-VANILLA, for all datasets.

|  | AGNews | BoolQ | MultiRC | GSM8K | MMLU | VQAv2 | TextVQA |
|---|---|---|---|---|---|---|---|
| Batch Size | 8 | 4 | 8 | 4 | 6 | 6 | 8 |
| Learning Rate | $10^{-3}$ | $10^{-3}$ | $10^{-3}$ | $10^{-5}$ | $10^{-4}$ | $10^{-4}$ | $10^{-4}$ |
| Optimizer | ADAMW | ADAMW | ADAMW | ADAMW | SGD Nesterov Momentum 0.9 | SGD | ADAMW |

**BP-CHECKPOINTING.**  BP-CHECKPOINTING (Chen et al., 2016) is identical to BP-VANILLA with one key difference: it employs activation checkpointing (also known as gradient checkpointing) to reduce memory consumption, allowing for larger batch sizes without incurring out-of-memory (OOM) errors. To ensure a fair comparison, the batch sizes used for BP-CHECKPOINTING match those used for the ZO and FMAD variants. The hyperparameters are given in Table 6.

Table 6: Hyperparameters related to BP-CHECKPOINTING and BP-ACCUMULATE, for all datasets.

|  | AGNews | BoolQ | MultiRC | GSM8K | MMLU | VQAv2 | TextVQA |
|---|---|---|---|---|---|---|---|
| Batch Size | 40 | 40 | 40 | 6 | 8 | 8 | 8 |
| Learning Rate | $10^{-3}$ | $10^{-3}$ | $10^{-3}$ | $10^{-5}$ | $10^{-4}$ | $10^{-4}$ | $10^{-4}$ |
| Optimizer | ADAMW | ADAMW | ADAMW | ADAMW | SGD Nesterov Momentum 0.9 | SGD | ADAMW |

**BP-ACCUMULATE.**  BP-ACCUMULATE follows the same training procedure as BP-CHECKPOINTING, but incorporates gradient accumulation to simulate larger effective batch sizes without exceeding memory constraints. Instead of updating model weights after every mini-batch, gradients are accumulated over multiple smaller batches and the update is performed after a fixed number of steps. At the end of the accumulation period, the summed gradients are averaged by dividing them by the number of accumulation steps. The hyperparameters are same as those of BP-CHECKPOINTING (see Table 6), with accumulation step count being 100 as default.

**ZO-VANILLA.**  ZO-VANILLA (Chen et al., 2019) implements a standard zero-order optimization approach, which estimates gradients using only function evaluations according to Equation 2, without requiring access to the model's internal, first-order gradients. Specifically, it perturbs the model parameters along randomly sampled directions and uses finite differences to approximate the gradient. We have used the memory-efficient perturbation trick of MEZO for all the ZO- variants, which includes storing the random seed and regenerating perturbations for forward pass evaluations, instead of persisting entire perturbations in the memory. For fair comparison, we use the same batch sizes as in BP-CHECKPOINTING and FMAD baselines. The hyperparameters are given in Table 7.

**ZO-ACCUMULATE.**  ZO-ACCUMULATE extends the ZO-VANILLA baseline by incorporating gradient accumulation to simulate larger effective batch sizes without exceeding memory constraints. Instead of estimating and applying a parameter update after each mini-batch, gradient approximations (based on finite differences) are accumulated over multiple steps and averaged before updating the model. This approach results in improved stability due to averaging out the noisy gradient estimates. The hyperparameters are same as with ZO-VANILLA, given in Table 7, with default accumulation window of 100.

Table 7: Hyperparameters related to ZO-VANILLA, for all datasets.

|  | AGNews | BoolQ | MultiRC | GSM8K | MMLU | VQAv2 | TextVQA |
|---|---|---|---|---|---|---|---|
| Batch Size | 40 | 40 | 40 | 6 | 8 | 8 | 8 |
| Learning Rate | $10^{-4}$ | $10^{-3}$ | $10^{-3}$ | $10^{-5}$ | $10^{-5}$ | $10^{-4}$ | $10^{-4}$ |
| Optimizer | ADAMW | ADAMW | SGD | SGD Nesterov Mmtm 0.9 | SGD Nesterov Mmtm 0.9 | ADAMW | SGD |
| Perturbation Step Size | $10^{-3}$ | $10^{-2}$ | $10^{-2}$ | $10^{-3}$ | $10^{-4}$ | $10^{-3}$ | $10^{-3}$ |

**ZO-MULTIPLE.** ZO-MULTIPLE (also shown in (Panchal et al., 2024; Xu et al., 2024; Feng et al., 2024)) builds on the ZO-VANILLA method by using multiple random perturbation directions per iteration, to improve the accuracy of the gradient estimate. Instead of relying on a single direction, this variant samples several perturbations and averages the resulting finite-difference approximations, leading to a lower-variance and more stable update. However, this approach increases the number of function evaluations per step. The hyperparameters are same as with ZO-VANILLA, given in Table 7, with default perturbation count per iteration of 10.

**ZO-ADAPTIVE.** ZO-ADAPTIVE enhances zero-order optimization by incorporating an adaptive perturbation strategy that aligns gradient estimates more closely with the true gradient direction over time. The optimization proceeds in two phases. In the *calibration phase* (typically the first iteration), multiple perturbation directions are sampled, and the one with the highest positive projected gradient is selected. This direction is assumed to have the smallest angle with the true gradient. This calibrated perturbation is then used to compute an initial gradient estimate. In the *adaptive phase* (subsequent iterations), new perturbations are sampled based on the previously estimated gradient, and a rolling average is maintained between the new perturbation and the historical gradient direction. This mechanism biases the search toward more promising directions while still allowing for exploratory variation. The hyperparameters are same as those of ZO-VANILLA, with the inclusion of sampling 4 perturbations during the calibration phase.

**ZO-SVRG.** ZO-SVRG (Liu et al., 2018) applies the principles of Stochastic Variance Reduced Gradient (SVRG) (Johnson and Zhang, 2013) to the zero-order optimization setting, aiming to improve convergence speed and stability by reducing the variance inherent in gradient estimates. The method alternates between two types of updates: full gradient estimation at a reference point (called a snapshot) and subsequent inner-loop updates that correct noisy estimates using control variates. In the zero-order context, both the snapshot gradient and the inner-loop updates are computed using finite-difference approximations along random perturbations. The variance reduction comes from reusing the snapshot gradient to correct each inner-step estimate. Besides the hyperparameters shown in Table 7, we use interval of 5 epochs to compute full gradients.

**ZO-SPARSE.** ZO-SPARSE (Guo et al., 2025) introduces sparsity into zero-order optimization by restricting gradient estimation and updates to only the top 1% of model parameters, selected based on their magnitude at each iteration. Unlike structured approaches such as LoRA, this method dynamically identifies and perturbs the most significant weights, those likely to contribute most to loss reduction. Hence, ZO-SPARSE focuses the optimization on a small, adaptive subset of parameters. This sparsity constraint reduces the dimensionality of the optimization problem, leading to fewer function evaluations. The hyperparameters are exactly the same as those of Table 7.

**MEZO.** MEZO (Malladi et al., 2023) builds on ZO-VANILLA, but with a key modification tailored for classification tasks using language models. Instead of relying on a separate classifier head, MEZO employs the language modeling (LM) head and masks out logits corresponding to vocabulary tokens that are not class labels. This approach is presented in the prompt-based fine-tuning strategy introduced by Gao et al. (2021). MEZO integrates this prompting technique with zero-order optimization, enabling effective gradient-free fine-tuning of large language models, although it is limited to the classification tasks. We use the same hyperparameters as ZO-VANILLA (see Table 7).

**FMAD-VANILLA.** FMAD-VANILLA implements the standard forward-mode automatic differentiation (Baydin et al., 2017; 2022) approach for computing gradients, more details are in §2. In this baseline, we use a straightforward implementation of forward-mode AD without any memory-saving strategies or structural optimizations. The hyperparameters used for FMAD-VANILLA are summarized in Table 8. Additionally, the variance of the Gaussian distribution used for perturbation sampling is fixed at 1 across all datasets.

Table 8: Hyperparameters related to FMAD-VANILLA, for all datasets.

|  | AGNews | BoolQ | MultiRC | GSM8K | MMLU | VQAv2 | TextVQA |
|---|---|---|---|---|---|---|---|
| Batch Size | 40 | 40 | 40 | 6 | 8 | 8 | 8 |
| Learning Rate | $10^{-3}$ | $10^{-4}$ | $10^{-4}$ | $10^{-5}$ | $10^{-5}$ | $10^{-4}$ | $10^{-4}$ |
| Optimizer | ADAMW | SGD | ADAMW | SGD Nesterov Mmtm 0.9 | SGD Nesterov Mmtm 0.9 | SGD | SGD |

**FMAD-ACCUMULATE.** FMAD-ACCUMULATE extends the standard forward-mode automatic differentiation by incorporating gradient accumulation to simulate larger batch sizes without increasing memory consumption. The same accumulation strategy is used in corresponding BP-ACCUMULATE and ZO baselines to maintain fairness in comparison. The hyperparameters are given in Table 8, with the addition of accumulation window of 100.

**FMAD-MULTIPLE.** FMAD-MULTIPLE enhances the basic forward-mode AD approach by using multiple perturbation directions per update to improve the stability and accuracy of gradient estimates. The setup closely mirrors that of ZO-MULTIPLE, with hyperparameters listed in Table 8. The only addition is the use of 10 perturbation count per iteration.

**FMAD-ADAPTIVE.** FMAD-ADAPTIVE mirrors the two-phase procedure described in ZO-ADAPTIVE, including the calibration phase for selecting an initial perturbation direction and the adaptive phase that updates this direction using a rolling average of past gradients. For full details, we refer the reader to the ZO-ADAPTIVE description. All hyperparameters remain consistent with Table 8, with calibration phase including 4 perturbations just like ZO-ADAPTIVE.

**FMAD-SVRG.** FMAD-SVRG adopts the same stochastic variance-reduced gradient (SVRG) framework used in ZO-SVRG, but applies it within the forward-mode AD setting. It alternates between full-gradient computation on a reference batch and variance-reduced updates on mini-batches, thereby reducing the noise in gradient estimates while maintaining computational efficiency. For details on the SVRG formulation, we refer the reader to the description of ZO-SVRG. Hyperparameters are in Table 8, with full gradients getting computed every 5 epochs (similar to ZO-SVRG).

**FMAD-SPARSE.** FMAD-SPARSE adopts the same sparsity strategy described in ZO-SPARSE, where only the top 1% of parameters (by magnitude) are selected for gradient updates during each iteration. As with the ZO-SPARSE variant, this method avoids techniques like LoRA and instead relies on direct selection of high-magnitude weights. For complete details on the sparsity mechanism, we refer the reader to the ZO-SPARSE description. All hyperparameters are in Table 8.

**A Note on the Theoretical vs. Empirical Learning Rate.** The theoretical convergence bound of ZO (Theorem I.8) has the condition of $\eta < \frac{2}{L(1+\frac{d+1}{n})}$. The condition becomes increasingly conservative as $L$ and $d$ scale, which is especially relevant for large models. This is a standard limitation of worst-case analysis: the bound is derived under minimal assumptions (e.g., global $L$-smoothness, worst-case variance), and thus prioritizes generality over tightness. In practice, we start with relatively large learning rates ($10^{-4}$ to $10^{-3}$) to measure the best-case time to convergence for ZO and FmAD. With adaptive optimizers like AdamW, the learning rate is automatically scaled down during training, often yielding stable and effective performance even when theoretical bounds are violated.

However, in line with the theory, we observe convergence failures (including NaNs or divergence, see Appendix F.5) when using non-adaptive optimizers such as SGD, especially under large $d/n$

ratios (typically around $10^5$) or for FmAD and ZO methods. These failures reinforce that while the theoretical bound is conservative, it qualitatively predicts instability when learning rates are too aggressive relative to dimensionality and batch size (see Appendices F.3 and F.5). That said, we do observe (especially in the zero-order case) that overly aggressive learning rates can lead to instability or degraded final performance, in line with the theoretical intuition. Hence, the theoretical rate serves as a safeguard for convergence analysis rather than a recommended training setting, and that practical hyperparameters typically benefit from empirical tuning beyond what the theory prescribes. Further discussion is provided in Corollary I.10.

## D    LIMITATIONS AND FUTURE WORK

While the aim of our work was to provide a comprehensive comparison of backpropagation (BP), forward-mode automatic differentiation (FMAD), and zero-order (ZO) optimization strategies, several limitations remain, which can serve as venues for a further exploration.

First, our experiments focus on deep models, and we did not systematically evaluate backpropagation with checkpointing (BP-CHECKPOINTING) on wider but shallower models. In principle, checkpointing may offer less benefit for such architectures. However, since wider and shallower models are relatively uncommon in practice, we chose not to extend our evaluations in that direction. Further, our checkpointing implementation operates at only one granularity (where which activations to checkpoint is not controlled by us) due to current Hugging Face library support, which limits finer control over which activations are saved or recomputed. Finer-grained checkpointing could reduce memory usage further and potentially narrow the memory efficiency gap between BP-CHECKPOINTING and ZO methods. However, this would come at the cost of increased runtime, introducing a different trade-off. Finally, while we focused on tuning and training LoRA layers, an important future direction would be to extend our comparison framework to full model finetuning. Such an extension would allow for a more complete characterization of the trade-offs between memory, time-to-convergence, and accuracy across different gradient computation strategies.

## E    BROADER IMPACT

Training deep learning models already carries a high environmental cost due to significant energy consumption. Our study shows that forward-mode AD and zero-order optimization, despite saving memory in some cases, require much longer training times and compute compared to backpropagation with checkpointing. This inefficiency leads to greater carbon emissions overall. Therefore, we show that optimizing for true computational efficiency (time-to-convergence and compute; along with memory consumption) is crucial for reducing the environmental footprint of large-scale training.

We also acknowledge that misinterpreting our results could lead to the premature dismissal of forward-mode AD or zero-order methods altogether. While they are not scalable replacements for backpropagation in large-scale training, they may still be uniquely suited for small models, non-differentiable tasks, or privacy-preserving settings where explicit gradients are inaccessible. Careful contextual understanding is necessary when applying our conclusions.

## F    ADDITIONAL RESULTS

### F.1    EXPERIMENTAL VARIANCE AND LOSS CURVES

Table 9 shows variance in reported accuracy numbers of Table 3. For each experiment, we performed three independent runs on seeds 0, 1, and 2. For each run, we computed the steady-state accuracy (averaged over the final evaluation steps). We then reported the mean (in Table 3) and variance (in Table 9) computed across these three steady-state accuracies. Furthermore, Figure 4 illustrates the training loss curves with respect to the training time, highlighting the convergence behavior. We have only showed the best-performing baselines to maintain clarity.

Figure 5 reports the mean gradient norm across all trainable parameters for all the datasets on LLAMA 3.1 (8B) model. These curves closely mirror the loss trajectories reported in Figure 4, exhibiting similar convergence tendencies across all methods. Specifically, BP-CHECKPOINTING shows the

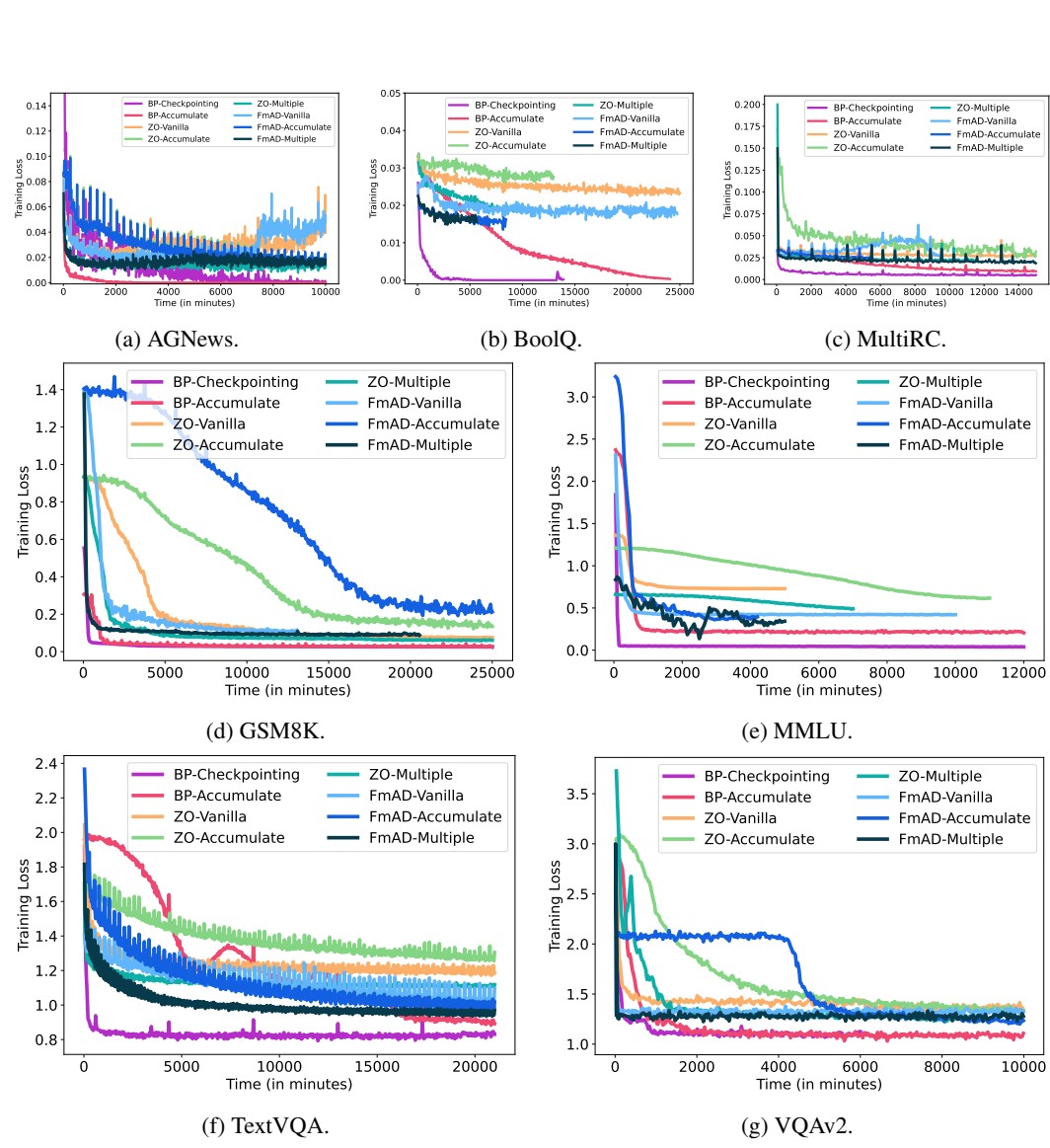

Figure 4: Training loss vs. training time (in minutes) for *(top)* training LLAMA 3.1 (8B) on three text classification datasets (AGNews, BoolQ, and MultiRC), and *(middle)* two text generation datasets (GSM8K and MMLU). *(bottom)* VQAv2 and TextVQA are used to train QWEN 2 VL (7B) on visual question-answering task.

Table 9: Experimental variance ($\pm$) of test accuracy across three runs with seeds 0, 1, and 2.

| Model + Dataset / Method | LLAMA 3.1 (8B) | | | | | QWEN 2 VL (7B) | |
|---|---|---|---|---|---|---|---|
| | AGNews | BoolQ | MultiRC | GSM8K | MMLU | VQAv2 | TextVQA |
| BP-VANILLA | 0.46 | 0.54 | 0.59 | 0.41 | 0.63 | 1.49 | 0.78 |
| BP-CHECKPOINTING | 0.45 | 0.56 | 0.62 | 0.39 | 0.61 | 1.52 | 0.77 |
| BP-ACCUMULATE | 0.67 | 0.78 | 0.84 | 0.79 | 0.69 | 1.71 | 0.98 |
| ZO-VANILLA | 0.98 | 0.76 | 0.8 | 0.55 | 0.95 | 1.22 | 0.89 |
| ZO-ACCUMULATE | 0.84 | 0.72 | 0.76 | 0.53 | 0.84 | 1.16 | 0.85 |
| ZO-MULTIPLE | 0.79 | 0.64 | 0.67 | 0.53 | 0.86 | 1.13 | 0.86 |
| ZO-ADAPTIVE | 1.02 | 0.95 | 1.13 | 0.84 | 0.83 | 0.95 | 0.78 |
| ZO-SVRG | 0.94 | 1.03 | 0.92 | 0.82 | 0.46 | 1.02 | 1.13 |
| ZO-SPARSE | 0.53 | 0.67 | 0.62 | 0.34 | 1.03 | 0.89 | 0.9 |
| MEZO | 0.86 | 0.73 | 0.73 | — | — | — | — |
| FMAD-VANILLA | 0.81 | 0.72 | 0.64 | 0.73 | 0.86 | 1.34 | 0.92 |
| FMAD-ACCUMULATE | 0.69 | 0.73 | 0.80 | 0.62 | 0.78 | 0.91 | 0.95 |
| FMAD-MULTIPLE | 0.85 | 0.77 | 0.89 | 1.04 | 0.96 | 0.74 | 0.83 |
| FMAD-ADAPTIVE | 1.63 | 1.25 | 1.34 | 0.95 | 1.52 | 1.11 | 1.31 |
| FMAD-SVRG | 1.42 | 0.96 | 0.89 | 1.02 | 1.44 | 1.05 | 1.29 |
| FMAD-SPARSE | 0.93 | 0.75 | 1.10 | 0.54 | 0.67 | 1.24 | 0.93 |

steepest and most stable decay in gradient norm, aligning with its superior convergence behavior in loss and accuracy. This strengthens the consistency between the theoretical observations of § 3 (which centers on the gradient norm) and our empirical findings of § 4.

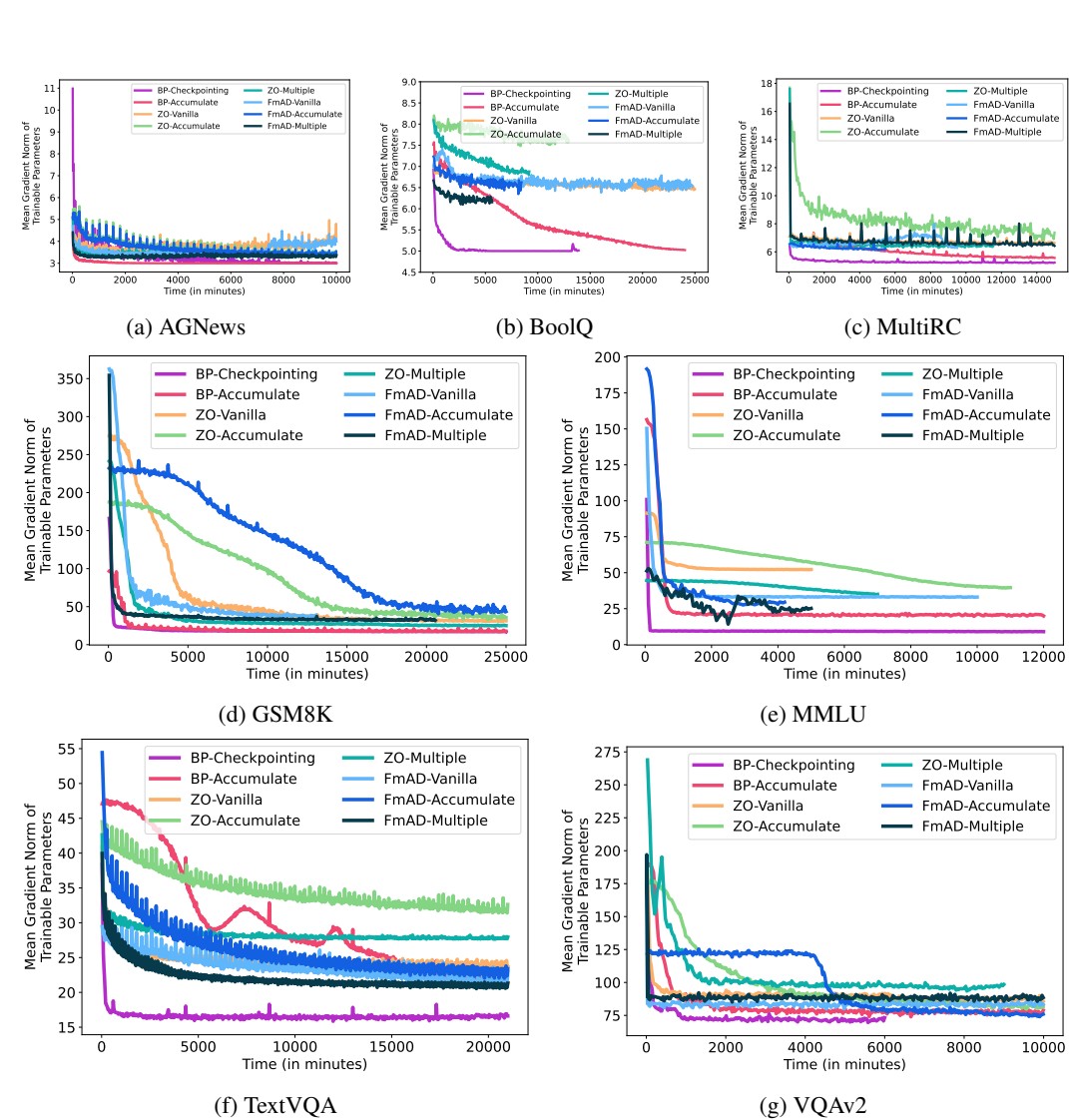

(a) AGNews    (b) BoolQ    (c) MultiRC

(d) GSM8K    (e) MMLU

(f) TextVQA    (g) VQAv2

Figure 5: Gradient norm vs. training time (in minutes) for *(top)* training LLAMA 3.1 (8B) on three text classification datasets (AGNews, BoolQ, and MultiRC), and *(middle)* two text generation datasets (GSM8K and MMLU). *(bottom)* VQAv2 and TextVQA are used to train QWEN 2 VL (7B) on visual question-answering task.

## F.2 EXPERIMENTS WITH MEDIUM-SIZED MODELS

The goal of these experiments was to investigate whether forward-mode automatic differentiation (FMAD) and zero-order (ZO) optimization could perform competitively when applied to medium-sized models, specifically BERT Base (110M), BERT Large (340M), ROBERTA Base (125M), and ROBERTA Large (350M). While FMAD and ZO have shown some promise on very small-scale problems in prior work (Cobb et al., 2024; Chen et al., 2019; Rostami and Kia, 2024), it remained an open question whether the convergence speed could scale reasonably with model sizes.

Figure 6 highlights a clear and consistent trend: backpropagation (with checkpointing) achieves superior convergence speed and final test accuracy, even for medium-sized models, compared to FMAD and ZO methods. Even for BERT Base (110M weights), FMAD and ZO lag significantly behind backpropagation in terms of convergence rate. While FMAD and ZO eventually approach a comparable final accuracy (with a gap of 0.74–1.66%) for BERT Base, they require substantially more training time to do so.

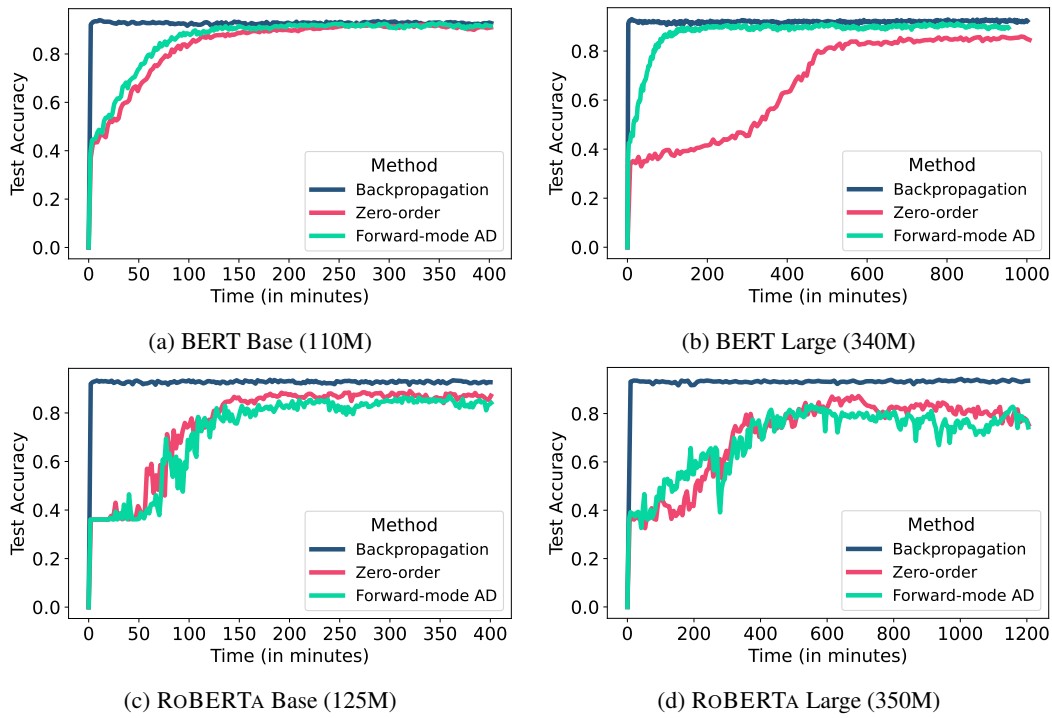

(a) BERT Base (110M)  (b) BERT Large (340M)

(c) ROBERTA Base (125M)  (d) ROBERTA Large (350M)

Figure 6: Accuracy versus training time comparison across Backpropagation (with checkpointing), Zero-order (ZO), and Forward-mode AD (FMAD) on BERT (Base and Large) and ROBERTA (Base and Large). Even at a smaller scale of trainable parameter count, ZO and FMAD either fail to reach to the accuracy of backpropagation (in case of ROBERTA), or takes longer to reach to the desired accuracy (in case of BERT).

As we scale to larger models, BERT Large and ROBERTA variants, the performance of FMAD and ZO deteriorates further. Both methods experience slower convergence, greater instability, and often plateau at lower final accuracies (with a drop of 1.19–6.71% for BERT Large, 6.76–7.62% for ROBERTA Base, 9.33–12.98% for ROBERTA Large) despite extensive training. ZO, in particular, struggles to reach acceptable performance, while FMAD shows increasingly volatile learning curves.

In summary, our experiments confirm that FMAD and ZO are fundamentally limited in their ability to compete with backpropagation in realistic settings. Their inefficiency becomes increasingly pronounced as we evaluate accuracy, along side memory consumption and time-to-convergence.

Table 10: Accuracy of BP, ZO, and FMAD under model size scaling: Only BP-CHECKPOINTING (abbreviated as BP-CHKPT) maintains high accuracy as model size increases.

Table 11: Accuracy as the LORA rank increases for OPT 6.7B: BP-CHECKPOINTING remains robust, while FMAD becomes unstable and ZO shows minimal gains.

| OPT Variants | Variant Size | Accuracy (↑) |
|---|---|---|
| BP-CHKPT | 1.3B | 94.08 |
| | 6.7B | 94.35 |
| | 13.0B | 94.51 |
| ZO-VANILLA | 1.3B | 73.16 |
| | 6.7B | 65.75 |
| | 13.0B | 71.00 |
| FMAD-VANILLA | 1.3B | 88.28 |
| | 6.7B | 87.50 |
| | 13.0B | 77.07 |

| OPT 6.7B | LORA Rank | Accuracy (↑) |
|---|---|---|
| BP-CHKPT | 1 | 94.35 |
| | 16 | 88.44 |
| | 32 | 85.54 |
| ZO-VANILLA | 1 | 65.75 |
| | 16 | 68.07 |
| | 32 | 68.97 |
| FMAD-VANILLA | 1 | 87.50 |
| | 16 | jvp = NaN |
| | 32 | jvp = NaN |

## F.3 CHANGING VARIANCE OF RANDOM PERTURBATION SAMPLING

We examine the effect of variance $\sigma^2$ of random perturbations which are sampled from Gaussian distribution $\mathcal{N}(0, \sigma^2)$ on the accuracy performance of FMAD and ZO. Figure 7 presents test accuracy over time for different values of $\sigma^2$, ranging from 1 to $10^{-2}$ for FMAD and from $10^{-2}$ to $10^{-4}$ for ZO.

The results reveal a strong sensitivity to the choice of variance: small variances reduce the diversity of perturbations, while large variances introduce excessive noise in high-dimensions, destabilizing training. Both FMAD and ZO achieve their best performance at intermediate values, $\sigma^2 = 1$ for FMAD and $\sigma^2 = 10^{-3}$ for ZO, which balance signal strength and noise. For ZO, reducing the variance from $10^{-3}$ to $10^{-4}$ results in a sharp accuracy drop of 13.75%. In contrast, FMAD shows a more gradual decline of 0.94% and 2.27% as $\sigma^2$ decreases from 1 to $10^{-1}$ and $10^{-2}$, respectively. The lower optimal variance for ZO arises from its gradient estimator, which includes an explicit division by the perturbation variance to scale the update magnitude (Equation 2). These findings suggest that simply reducing variance of the distribution from which perturbations are sampled does not result in better gradient estimates, nor does it improve convergence.

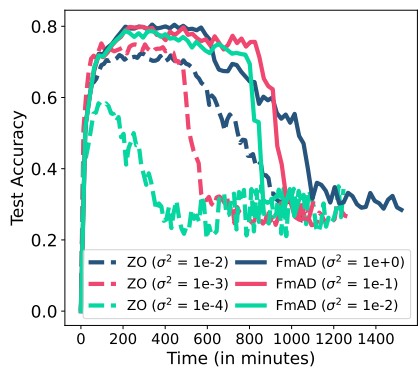

Figure 7: Changing variance $\sigma^2$ of random sampling of perturbations. Directly reducing randomness variance does not lead to reduced noise in the gradients.

## F.4 REDUCING TRAINABLE PARAMETER COUNT

We investigate how increasing the number of trainable parameters affects performance under BP, ZO and FMAD. Tables 10 and 11 present results across varying model sizes and LORA ranks, respectively. Further comparison of convergence time is available in Figure 8.

In Table 10, we evaluate BP-CHECKPOINTING, ZO-VANILLA, and FMAD-VANILLA on OPT model variants of size 1.3B, 6.7B, and 13B. As the model size increases, BP-CHECKPOINTING consistently maintains high accuracy of ∼94%. In contrast, ZO and FMAD exhibit noticeable drops in accuracy at larger model scales. Notably, FMAD achieves 88.28% accuracy on the 1.3B model but declines to 77.07% on the 13B model, showing degradation from scaling the count of trainable parameters. This result are consistent with our theoretical findings of convergence error bounded by the trainable parameter count (§3). Table 11 explores accuracy as a function of LORA rank for OPT 6.7B. While BP-CHECKPOINTING degrades gracefully as rank increases (likely due to overfitting), FMAD becomes unstable and fails to converge beyond rank 1, yielding NaN outputs for higher ranks. FMAD's instability at higher LORA ranks is due to inherent insta-

bility of perturbation-based gradient estimations, which we discuss in Appendix F.5. ZO, while stable, shows limited improvement with increased rank, reaching only 68.97% accuracy at rank 32.

## F.5 FAILURE MODE ANALYSIS

In order to understand why variance reduction methods or adaptive optimizers sometimes fail to make FMAD and ZO converge, or converge at a suboptimal accuracy; we present failure mode analysis with different optimizers and SVRG.

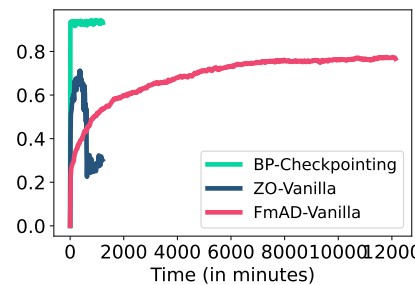

### F.5.1 CHALLENGES WITH OPTIMIZER CHOICE

Here we discuss a distinct failure mode of FMAD which has been frequently observed in our preliminary experiments: the computed Jacobian-vector products (jvp) abruptly spike in magnitude. These sudden surges lead to disproportionately large gradient updates, destabilizing training and hindering convergence. A similar failure mode has been observed in zero-order (ZO) methods, where the projected gradients, mathematically equivalent to FMAD's jvp values, exhibit comparable instability.

Figure 8: Comparison of convergence time among BP-CHECKPOINTING, FMAD-VANILLA, and ZO-VANILLA with OPT(13B) on AGNews dataset.

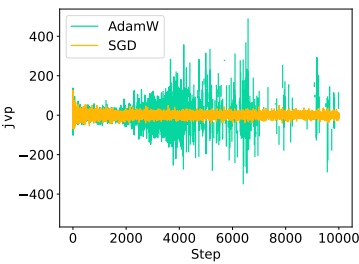

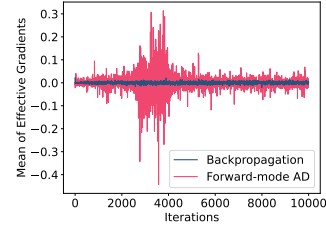

(a) ADAMW optimizer.

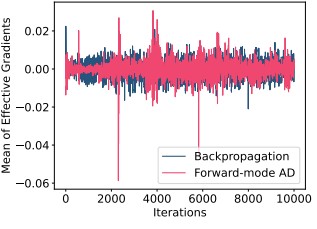

(b) SGD optimizer.

Figure 9: Effect of ADAMW and SGD optimizers on jvp values in FMAD on GSM8K dataset.

Figure 10: Mean of effective gradients of Backpropagation and Forward-mode AD with ADAMW and SGD optimizers on GSM8K dataset.

Figure 9 illustrates the impact of optimizer choice, specifically ADAMW (adaptive) versus SGD (non-adaptive), on jvp values in FMAD. Under SGD, jvp values remain bounded within a stable range of $[-50, 50]$ for the case of GSM8K dataset. However, with ADAMW, these values exhibit a gradual increase followed by sharp spikes for certain datasets including GSM8K. In some cases, the spikes reach $8-10\times$ higher magnitudes than the stable baseline observed with SGD.

Figures 9a and 9b further illustrate the implications of these spikes. Under ADAMW, the effective gradient magnitudes produced by FMAD exhibit substantially higher variance than those from backpropagation, indicating instability and less reliable gradient directions. These inflated updates also increase weight magnitudes, which in turn amplify subsequent jvp evaluations, since these depend on both the current weights and their perturbations. This positive feedback loop can lead to divergence and, eventually, NaN values in jvp computations, as observed in several FMAD runs in Table 11. Even when divergence does not occur, the resulting gradient updates can be excessively noisy or of high magnitude, leading to suboptimal convergence. In contrast, under SGD, the effective gradients computed by FMAD closely mirror those from backpropagation across most iterations, with stable behavior and no evidence of runaway magnitudes. We posit that this cascading rise in magnitude for the case of ADAMW is due to its adaptive nature, where a rolling average of historical and current gradients is computed each iteration, leading to amplification of higher magnitude gradients. In contrast, the impact of jvp spikes is diminished with non-adaptive SGD since the previous iteration's gradients would have limited effect (to only one iteration's gradient updates).

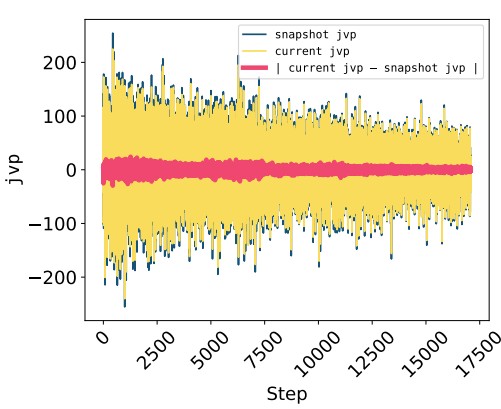 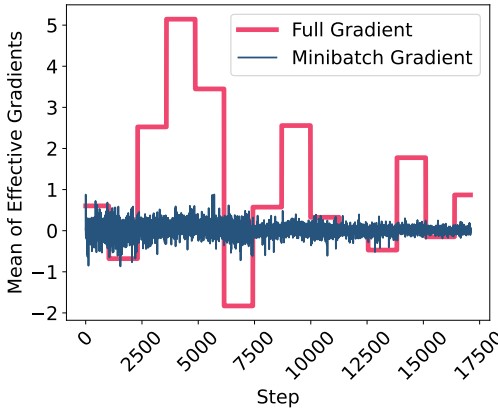

(a) The current iteration's `jvp`, `jvp` computed based on the snapshot weights, and the difference between these two values.

(b) Magnitude of the mean effective full gradients (computed at specific epoch intervals) and the current iteration's mini-batch gradient.

Figure 11: Impact of incorporating SVRG into FMAD on (a) `jvp` values and the mean of effective full-batch and (b) mini-batch gradients, evaluated on the GSM8K dataset.

This stark contrast highlights a critical interaction between optimizer choice and the numerical stability of FMAD. While ADAMW is widely favored for its adaptive learning rates and regularization capabilities, its use with FMAD (and by extension ZO) can introduce harmful gradient artifacts which are driven by uncontrolled `jvp` amplification. These results underscore the specific vulnerabilities in gradient estimation methods and point to a need for further study into stabilizing FMAD and ZO for more reliable deployment in large-scale training regimes.

### F.5.2    CHALLENGES WITH SVRG

In this section, we discuss a failure mode of SVRG observed in the context of text generation tasks. While SVRG improves performance for both ZO-VANILLA and FMAD-VANILLA baselines by 4.04–11.13 and 1.97–3.86, respectively, in many settings, it leads to performance degradation in certain sequence modeling tasks like GSM8K. Figure 11 illustrates the behavior of `jvp` values and the corresponding gradients when SVRG is applied to FMAD on the GSM8K dataset. In Figure 11a, we observe that the difference between the `jvp` computed on the current model weights and the one computed on the snapshot weights is minimal. Consequently, the control variate, the difference between mini-batch gradients at current and snapshot weights, has little impact relative to the magnitude of the full gradient.

This hypothesis is supported by Figure 11b, which shows that the mean of the effective full gradients remains consistently large, while the mini-batch gradient magnitudes are significantly smaller. Because the full gradients are updated only at periodic intervals (every 5 epochs in our case), their inflated magnitude dominates the update direction across multiple steps. This inflation stems from the accumulation of large `jvp` values during the summation of per-batch gradients, occasionally resulting in outlier gradients with extremely high norms. As a result, the SVRG mechanism fails to provide meaningful variance reduction and instead perpetuates overly large updates, ultimately degrading model performance.

A similar performance degradation was observed in ZO-SVRG (Liu et al., 2018), albeit on a smaller model with approximately 852K parameters. However, that work does not address the scalability challenges of SVRG-based methods in the context of zeroth-order optimization.

### F.5.3    IMPROVING STABILITY VIA MULTIPLE-PERTURBATION AND ACCUMULATED-GRADIENT

We further extend our analysis of `jvp` magnitudes and mean gradient values to the variance-reducing baselines -MULTIPLE (which samples multiple perturbations per iteration and averages the resulting

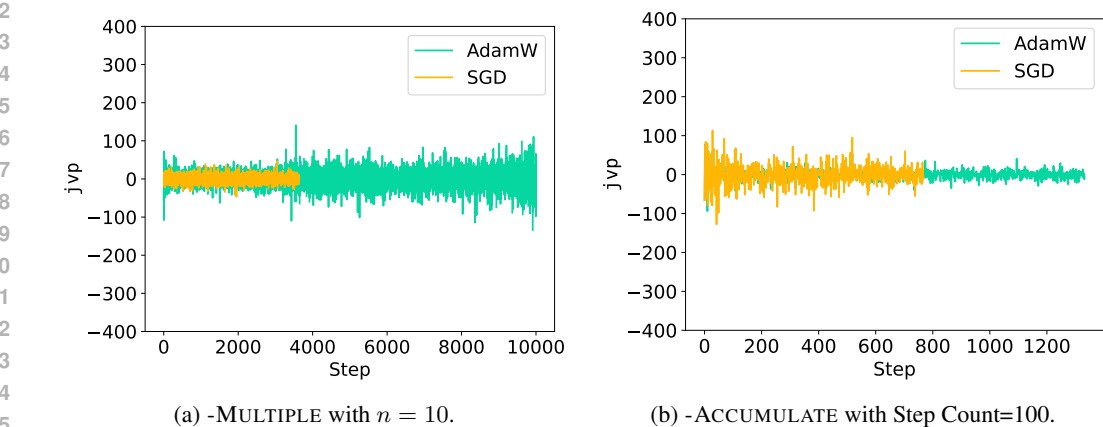

(a) -MULTIPLE with $n = 10$.  (b) -ACCUMULATE with Step Count=100.

Figure 12: Effect of ADAMW and SGD optimizers on `jvp` values in FMAD-MULTIPLE and FMAD-ACCUMULATE on GSM8K dataset.

gradients) and -ACCUMULATE (which uses a single perturbation per iteration but accumulates gradients over several steps before applying an update).

Figure 12 reports the corresponding `jvp` trajectories. Note that the SGD baseline contains fewer plotted steps because it converged substantially earlier than the other configurations, and the experiment was therefore terminated once convergence was reached. In contrast to the instability observed for FMAD-VANILLA in Figure 9, both baselines exhibit stable `jvp` magnitudes even under ADAMW. This stability, in turn, yields lower error and more reliable convergence. This behavior can be attributed to the inherent variance-reduction mechanisms in these baselines. In -MULTIPLE, averaging *multiple* jvp-induced gradient estimates suppresses the high-variance noise that otherwise interacts negatively with ADAMW's adaptive accumulators. Similarly, -ACCUMULATE delays updates and aggregates gradient signals across several steps, effectively smoothing out perturbation-induced fluctuations before the optimizer sees them. In both cases, the optimizer receives a more stable and lower-variance gradient stream, preventing the cascading amplification effects that cause jvp spikes in FMAD-VANILLA. As a result, these variance-reduction strategies mitigate the optimizer–noise interaction responsible for divergence, leading to substantially more stable training dynamics.

Figure 13 reports the mean gradient magnitudes. The curves for Backpropagation and FMAD are identical to those shown previously in Figure 10. In addition, we include the results for FMAD-MULTIPLE and FMAD-ACCUMULATE. Unlike the pronounced gradient-magnitude spikes observed in FMAD, both -MULTIPLE and -ACCUMULATE exhibit markedly steadier behavior under both optimizers ADAMW and SGD. Notably, ACCUMULATE displays the greatest stability. This is expected: accumulating gradients over several iterations before applying an update effectively averages out the perturbation-induced noise and prevents high-variance signals from being directly fed into the optimizer's adaptive state. As a result, ADAMW receives smoother, lower-variance updates, which suppresses the positive feedback loop responsible for the divergence in FMAD. In contrast, MULTIPLE exhibits a slight upward drift near the end of training when used with ADAMW. This behavior is consistent with the fact that, although multiple perturbations are averaged per iteration, the optimizer still processes an update at every step; thus, residual noise (especially as weights grow in magnitude) can accumulate in the adaptive moments and produce a mild increase in gradient scale. Nevertheless, this increase remains small relative to the uncontrolled spikes in FMAD-VANILLA, confirming that perturbation-level averaging substantially reduces variance. Finally, note that ACCUMULATE has fewer points plotted because it performs fewer parameter-update steps; gradients are accumulated locally and applied only periodically, resulting in a lower number of optimizer interactions reflected in the visualization.

## F.6 EFFECT OF PERTURBATION DISTRIBUTIONS AND NORMALIZATION STRATEGIES

We additionally experimented with perturbation sampling strategies: (a) Sampling from a normal distribution and using the perturbations as-is (unnormalized), (b) Sampling from a normal distri-

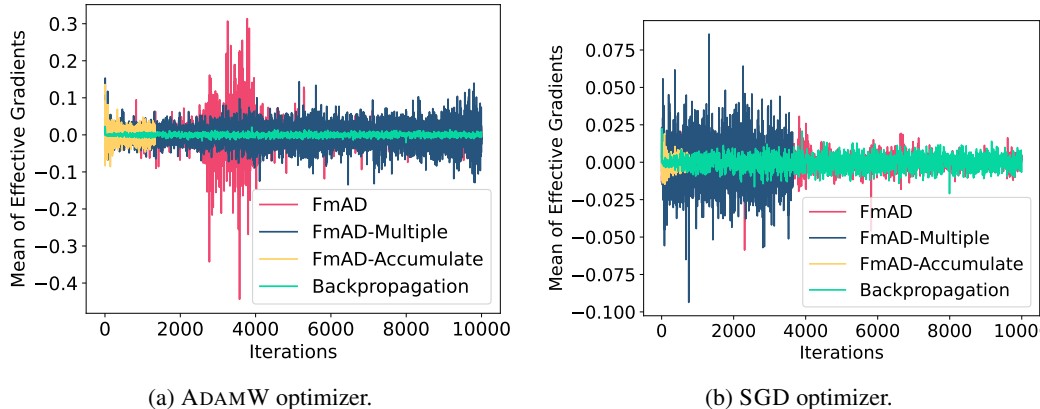

(a) ADAMW optimizer.  (b) SGD optimizer.

Figure 13: Mean of effective gradients of Backpropagation, FMAD, FMAD-MULTIPLE ($n = 10$), and FMAD-ACCUMULATE (Step Count=100) with ADAMW and SGD optimizers on GSM8K dataset.

bution and normalizing the perturbations, (c) Sampling from a uniform distribution and using the perturbations as-is (unnormalized), and (d) Sampling from a uniform distribution and normalizing the perturbations.

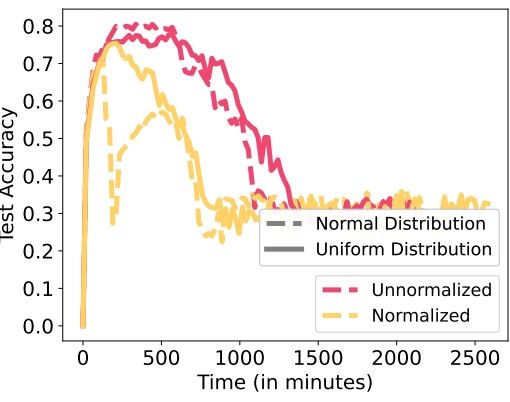

Figure 14: Finetuning LLAMA 3.1 (8B) on the AG-News dataset using FMAD, comparing perturbations drawn from normal vs. uniform distributions, with both normalized and unnormalized variants.

Our findings are as follows. Normalization consistently reduces accuracy for both the normal and uniform variants. This degradation arises because normalization forces every perturbation to have identical magnitude, eliminating natural variability in scale that carries useful information for estimating the local curvature of the loss landscape. By projecting all perturbations onto a fixed-radius hypersphere, the method reduces the effective signal-to-noise ratio of the `jvp` estimate and prevents larger, informative perturbations (particularly in high-curvature regions) from contributing to learning. As a result, the gradients become less expressive and exhibit higher relative variance, leading to poorer optimization.

Among the unnormalized variants, sampling from a normal distribution yields the strongest performance, with the unnormalized uniform distribution performing comparably closely, before both resulting in overfitting. The slight advantage of the normal distribution can be attributed to its heavier tails, which naturally introduce a broader range of perturbation magnitudes. This diversity more closely mimics the statistical structure of true gradients in large neural networks, allowing the estimator to explore directions of both small and moderately large curvature. In contrast, the unnormalized uniform distribution produces perturbations bounded within a fixed interval, limiting the range of effective step sizes and resulting in marginally less efficient gradient estimation.

### F.7 COMPARISON AGAINST SIGNZO

Table 12 shows a comparison of accuracy, memory usage, compute cost, and convergence time for BP-CHECKPOINTING, ZO, SIGNZO, ZO-ACCUMULATE, and ZO-MULTIPLE when finetuning LLAMA-3.1 (8B) on AGNews.

Table 12: Performance, memory, and efficiency trade-offs across BP-CHECKPOINTING, ZO baselines, and SIGNZO for finetuning LLAMA-3.1 (8B) on AGNews.

|  | Accuracy | Memory Consumption (in GB) | Compute Cost (in FLOPs) | Wallclock Convergence Time (in seconds) |
|---|---|---|---|---|
| BP-CHECKPOINTING | 93.8% | 11.66 | 65.2 x $10^4$ | 16,691 |
| ZO | 73.6% | 5.99 | 251.2 x $10^4$ | 21,074 |
| SIGNZO | 82.6% | 5.99 | 251.9 x $10^4$ | 56,892 |
| ZO-ACCUMULATE | 85.8% | 5.99 | 2165.1 x $10^4$ | 181,510 |
| ZO-MULTIPLE | 86.7% | 5.99 | 2425 x $10^4$ | 201,747 |

**Accuracy Comparison:** Although SIGNZO improves stability relative to vanilla ZO (as reflected in its smoother learning trajectory in the Figure 15) its overall accuracy performance remains significantly below the backpropagation baseline. On AGNews with LLAMA3.1 (8B), SIGNZO reaches 82.6% accuracy, which is a noticeable improvement over the 73.6% achieved by standard ZO but still far from the 93.8% obtained via backpropagation. This gap indicates that the sign-based estimator, while stabilizing the update direction, does not provide sufficient gradient resolution to match the fidelity of true gradients.

SIGNZO also underperforms compared to the variance-reducing methods (ZO-MULTIPLE and ZO-ACCUMULATE). ZO-MULTIPLE and ZO-ACCUMULATE reach 86–87% accuracy, and although they require larger FLOPs and longer runtimes, they converge to higher-quality solutions.

**Memory, Computation cost, and Convergence Time Comparison:** SIGNZO matches the memory footprint of other ZO baselines (5.99 GB) and maintains similar FLOP-level compute costs. However, its wall-clock convergence time is substantially longer ($\approx$56.9k seconds), more than 2.7$\times$ slower than ZO and 3.4$\times$ slower than BP-Checkpointing. The longer convergence time stems from the fact that stabilizing noisy ZO directions via sign compression requires more optimization steps to make meaningful progress. Although Table 12 reports only wall-clock time, SIGNZO and ZO have identical

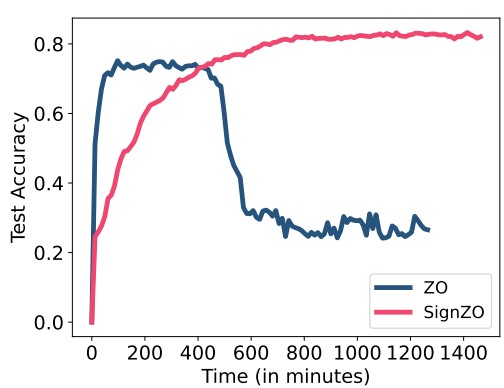

Figure 15: SignZO against ZO for training LLAMA 3.1 (8B) on AGNews dataset.

per-iteration runtime, the only difference between them is that SIGNZO applies a sign-compressed update during `optimizer.step()`, which does not affect iteration cost. Consequently, time on the x-axis is effectively proportional to the number of optimization steps, allowing us to conclude that the longer wall-clock time directly reflects the larger number of iterations required for convergence.

Overall, SIGNZO improves upon naïve ZO in terms of final accuracy (82.6% vs. 73.6%), but does so by requiring substantially more computation: although its per-iteration FLOPs are nearly identical to ZO, its wall-clock convergence time is 2.7$\times$ longer (56.9k s vs. 21.1k s). Compared to BP-Checkpointing, SIGNZO achieves a markedly smaller memory footprint (5.99 GB vs. 11.66 GB), but only by trading off both efficiency and performance, requiring $\approx$3.4$\times$ longer time to converge, $\approx$3.9$\times$ more compute, and yielding 11.2 percentage points lower accuracy. Furthermore, while SIGNZO converges faster than the variance-reduced ZO-Accumulate and ZO-Multiple baselines, those methods achieve higher accuracies (85.8% and 86.7%), reinforcing the broader trend observed in our paper: stability alone is not sufficient, effective ZO training at scale also requires variance-reduction mechanisms to improve both accuracy and efficiency.

### F.8 SENSITIVITY TO PERTURBATION BUDGET FOR OPT13B

Figure 16 shows the training of the OPT (13B) model on the AGNews dataset using ZO with varying perturbation budgets $n$.

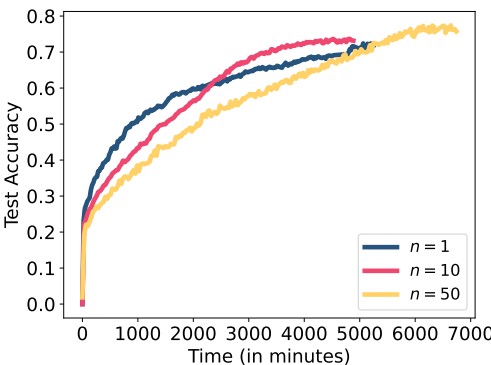

For $n = 1$, convergence occurs around 5000 minutes, reaching an accuracy of 71%. With $n = 10$, convergence also occurs near 5000 minutes, with a slight improvement in accuracy to 72%. Increasing the budget further to $n = 50$ improves the final accuracy to 75.5%, but convergence is delayed until approximately 6500 minutes. This behavior can be explained by the trade-off between gradient estimate quality and computational overhead: larger perturbation budgets reduce the variance of the `jvp` estimates, leading to more accurate gradients and higher final accuracy, but in this case, the perturbations are applied sequentially due to hardware limitations, which increases wall-clock time per iteration. Additionally, small increases in $n$ (e.g., from 1 to 10) yield only modest accuracy gains because even a few perturbations are sufficient to capture enough directional information for effective early-stage training. Overall, these results highlight a trade-off between perturbation budget, convergence speed, and final accuracy: larger budgets improve the quality of gradient estimates at the cost of increased computation time, particularly when sequential execution is required.

Figure 16: Training the OPT (13B) model on the AGNews dataset using ZO with varying perturbation budgets $n$. Larger budgets reduce variance in the `jvp` estimates, improving final accuracy, but sequential application of perturbations under resource constraints increases wallclock convergence time.

## G SIGNAL PROPAGATION FOR GRADIENT COMPUTATION

A key difference between BP and FMAD/ZO methods lies in how they propagate the loss signal to compute weight updates. BP computes the derivative of the loss $\mathcal{L}$ with respect to each weight $w_i$, effectively mapping changes in the loss to precise updates in the parameter space. The gradients of the intermediate activations, computed during the backward pass, are also derived from $\delta\mathcal{L}$, allowing the loss signal to guide every stage of the update. This direct path from the loss to the parameters makes BP a **loss-to-weights** approach, where the signal flows backward through the network in a structured and deterministic way.

In contrast, both FMAD and ZO adopt a **weights-to-loss** perspective: they estimate how perturbations in the weights, $\delta\boldsymbol{w} = \boldsymbol{v}$; affect the loss, $\delta\mathcal{L}$. The forward-mode Jacobian-vector product (`jvp`) and the ZO projected gradient scalar both incorporate the resulting change in the loss, but they do so indirectly. Specifically, they multiply $\delta\mathcal{L}$ by the perturbation direction $\boldsymbol{v}$ to approximate weight gradients (as detailed in § 2). However, in these approaches, the intermediate changes $\delta y_p$ (which influence $\delta\mathcal{L}$) are driven by the initial perturbations $\delta w_p \sim \mathcal{N}(0, I_d)$; not by the loss. As a result, the variance introduced at the input level through the perturbations propagates forward through the network, ultimately contaminating the gradient signal. This lack of an explicit loss-driven mechanism for shaping activation gradients leads to noisier gradient updates. Consequently, FMAD and ZO require stricter step size constraints (see Theorems I.8 and I.9) and exhibit degraded convergence behavior.

Moreover, both FMAD and ZO optimization methods incur additional noise and estimation error compared to backpropagation. This noise is not just a side effect, it is an inherent consequence of using random perturbations to estimate gradients. In both FMAD and ZO, the injection of perturbations $\delta\boldsymbol{w} \sim \mathcal{N}(0, I_d)$ is core to the algorithmic process, and the resulting activation ($\delta y_i$) and loss variations ($\delta\mathcal{L}$) carry this randomness forward. Therefore, the gradient estimates vary depending on the sampled perturbation, making noise a deterministic outcome of the method itself.

In essence, while BP precisely channels loss information to guide weight updates, FMAD and ZO rely on stochastic approximations that make their updates fundamentally noisy and less targeted.

# H    COMPUTATIONAL COMPLEXITY

In this section, we analyze the computational complexity of different methods used to compute gradients in a neural network setting. We begin with a one-layer neural network, providing a detailed breakdown of the computational cost for the forward pass, backpropagation, zero-order optimization, and forward-mode automatic differentiation. Understanding these complexities is essential for evaluating the efficiency of gradient computation methods, especially in resource-constrained environments. Empirical computational cost of the gradient computation methods is shown in §4.5.

## H.1    BASICS

In this section, we analyze the computational complexity of a one-layer neural network $f$ with weight matrix $w \in \mathbb{R}^{d \times m}$. The network takes an input $x \in \mathbb{R}^d$ and produces an output $y \in \mathbb{R}^m$. While we focus on a single-layer setting for clarity, the analysis naturally extends to a deep neural network with $L$ layers, each with weight matrix $w_\ell$ for $\ell \in [L]$.

**Forward Pass.**    Since all three gradient computation methods share the same forward pass, we first establish its computational complexity. The forward pass consists of a matrix multiplication $y = xw$, where $x$ has dimensions $1 \times d$ and $w$ has dimensions $d \times m$. This results in a computational complexity of $\mathcal{O}(dm)$.

## H.2    BACKPROPAGATION

Backpropagation requires computing the gradient of the loss $\mathcal{L}$ with respect to the weights, given by

$$\frac{\partial \mathcal{L}}{\partial w} = \frac{\partial \mathcal{L}}{\partial y} \cdot \frac{\partial y}{\partial w}.$$

The first term, $\frac{\partial \mathcal{L}}{\partial y}$, involves differentiating the loss with respect to the output, which has a computational complexity of $\mathcal{O}(m)$. The second term, $\frac{\partial y}{\partial w}$, follows from the linear transformation $y = xw$, contributing a complexity of $\mathcal{O}(dm)$. The final gradient computation involves the multiplication of a $1 \times m$ matrix with an $m \times d$ matrix, resulting in an additional complexity of $\mathcal{O}(dm)$.

Although activation functions introduce constant factors, 3 for the last layer and 5 for intermediate layers, these constants do not affect the asymptotic complexity. Hence, the overall computational complexity of backpropagation remains $\mathcal{O}(dm)$.

## H.3    BACKPROPAGATION WITH CHECKPOINTING

Checkpointing builds on standard backpropagation by trading memory for additional computation. Instead of storing all intermediate activations, only selected layers are checkpointed, and discarded activations are recomputed as needed during the backward pass.

This recomputation introduces an overhead, resulting in a total compute complexity of $\mathcal{O}(dm \log p)$ for a network with $p$ layers Griewank and Walther (2000). Here, the $\log p$ factor reflects the optimal checkpointing schedule, capturing the additional cost of recomputing intermediate activations while still reducing peak memory usage compared to standard backpropagation. In this way, checkpointing offers a controlled trade-off between memory efficiency and computational overhead, extending the base $\mathcal{O}(dm)$ cost of standard backpropagation.

## H.4    ZERO-ORDER OPTIMIZATION

The zero-order optimization method with central finite differences involves perturbing the weights twice, evaluating at $(\boldsymbol{w} + \epsilon \boldsymbol{v})$ and $(\boldsymbol{w} - \epsilon \boldsymbol{v})$, where $\boldsymbol{v} \in \mathbb{R}^{d \times m}$ is a randomly sampled perturbation and $\epsilon \in \mathbb{R}$ is a small step size. The element-wise multiplication $\epsilon \boldsymbol{v}$ incurs a computational cost of

$\mathcal{O}(dm)$, as does the addition and subtraction with $w$. Since each perturbation requires evaluating the function at the perturbed points, the function evaluations $f(\boldsymbol{w} \pm \epsilon \boldsymbol{v})$ also contribute a complexity of $\mathcal{O}(dm)$.

With $n$ such perturbations per iteration, the total computational cost sums to $\mathcal{O}(ndm)$, where $n$ is the number of perturbations used in each iteration.

Compared to the forward pass on the original weights $w$, zero-order adds a constant of 4, which gets absorbed in $\mathcal{O}(ndm)$.

### H.5 Forward-Mode AD

The jvp (Jacobian-vector product) computation incurs a complexity of $\mathcal{O}(dm)$, as it partially computes $\frac{\partial \mathcal{L}}{\partial w}$. The resulting jvp is then multiplied with the perturbation vector $\boldsymbol{v}$ to obtain the weight gradient for $\boldsymbol{w}$. Since $\boldsymbol{v}$ has dimensions $d \times m$, this multiplication also has a computational complexity of $\mathcal{O}(dm)$.

Repeating this process $n$ times for $n$ perturbations per iteration leads to a total computational cost of $\mathcal{O}(ndm)$.

## I Proofs of Convergence Bounds

This section includes the details on upper error bounds of all three gradient computation methods: Backpropagation, Zero-order optimization, and Forward-mode Auto Differentiation.

### I.1 Basics

All examples of gradient computation methods are based on a function $f$, which, in the context of machine learning, corresponds to a neural network. This function $f$ is composed of nested functions $f_i$, $i \in [p]$; where each function corresponds to an intermediate output (or activation) $y_i = f_i(w_i, y_{i-1})$, generated from the input weights $w_i$ and previous activation $y_{i-1}$. $y_0$ is set to $x$, which can be data points in ML. We assume that $x$ is fixed, for the ease of exposition. The input weights are represented by the vector $\boldsymbol{w} = w_1, w_2, \ldots, w_p$, where each $w_{[1,\ldots,p]} \in \mathbb{R}^{[m_1,\ldots,m_p]}$. The intermediate outputs, or activations, are denoted by $\boldsymbol{y} = y_1, \ldots, y_p$. The final output is $y = y_p = f(\boldsymbol{w}, x) \in \mathbb{R}^n$, where typically $n << m_i$ for all $i \in [p]$. The loss function $\mathcal{L}(y, \hat{y}) \in \mathbb{R}$ is then computed to measure the difference between the predicted output $y$ and the true target values $\hat{y}$.

With gradient descent, one update to the weights $\boldsymbol{w}$ looks like this,

$$\boldsymbol{w}_{t+1} \leftarrow \boldsymbol{w}_t - \eta \nabla f(\boldsymbol{w}_t), \tag{3}$$

where $t$ is the iteration count, and $\eta$ is the learning rate.

The objective is to minimize $f(\boldsymbol{w})$: $\min_{\boldsymbol{w} \in \mathbb{R}^d} f(\boldsymbol{w})$.

**Definition I.1** (Optimality Gap). The optimality gap at iteration $t$ is defined as the difference between the function value at the current iterate $\boldsymbol{w}_t$ and the function value at an optimal solution $\boldsymbol{w}^*$:

$$\Delta_t = f(\boldsymbol{w}_t) - f(\boldsymbol{w}^*) \tag{4}$$

The optimality gap quantifies how far the current function value is from the optimal value. In convergence analysis, the goal is to show that this gap decreases over iterations.

### I.2 Assumptions

**Assumption I.2** (Smoothness). Let $f : \mathbb{R}^d \to \mathbb{R}$ be $L$-smooth, meaning that its gradient is Lipschitz continuous with constant $L > 0$. That is, for all $\boldsymbol{w}, \boldsymbol{w}' \in \mathbb{R}^d$, the function $f$ satisfies,

$$||\nabla f(\boldsymbol{w}') - \nabla f(\boldsymbol{w})|| \leq L||\boldsymbol{w}' - \boldsymbol{w}||. \tag{5}$$

### I.3 LEMMAS

**Lemma I.3** (Bias of Gradient Estimate of the Central Finite Difference). *Let $g(\boldsymbol{v})$ be the gradient estimate obtained using the central finite difference method with a perturbation vector $\boldsymbol{v} \sim \mathcal{N}(0, I_d)$. Then, the expectation of the estimator satisfies*

$$\mathbb{E}_{\boldsymbol{v}}[g(\boldsymbol{v})] = \nabla f(\boldsymbol{w}), \tag{6}$$

*implying that the central finite difference gradient estimator is unbiased up to first-order error terms. Furthermore, the second moment of the estimator satisfies*

$$\mathbb{E}_{\boldsymbol{v}}\left[\|g(\boldsymbol{v})\|^2\right] = \|\nabla f(\boldsymbol{w})\|^2 (d+2) + \mathcal{O}(\epsilon^2)d. \tag{7}$$

*Proof.* We start with one evaluation of the central finite difference, for a perturbation $\boldsymbol{v} \in \mathbb{R}^d$. The full derivation is given in Theorem I.8,

$$g(\boldsymbol{v}) = \left((\boldsymbol{v}^\top \nabla f(\boldsymbol{w}))\boldsymbol{v} + \mathcal{O}(\epsilon)\boldsymbol{v}\right). \tag{8}$$

In order to measure the bias of the above gradient estimate, we take expectation with respect to the randomness of $\boldsymbol{v}$,

$$\mathbb{E}_{\boldsymbol{v}}[g(\boldsymbol{v})] = \mathbb{E}_{\boldsymbol{v}}\left[(\boldsymbol{v}^\top \nabla f(\boldsymbol{w}))\boldsymbol{v} + \mathcal{O}(\epsilon)\boldsymbol{v}\right] \tag{9}$$

$$= \nabla f(\boldsymbol{w})\mathbb{E}_{\boldsymbol{v}}\left[\boldsymbol{v}^\top \boldsymbol{v}\right] + \mathcal{O}(\epsilon)\mathbb{E}_{\boldsymbol{v}}\left[\boldsymbol{v}\right] \tag{10}$$

$$= \nabla f(\boldsymbol{w})I_d + \mathcal{O}(\epsilon) \cdot 0 \quad (\text{since } \boldsymbol{v} \sim \mathcal{N}(0, I_d)) \tag{11}$$

$$\therefore \mathbb{E}_{\boldsymbol{v}}[g(\boldsymbol{v})] = \nabla f(\boldsymbol{w}). \tag{12}$$

This shows that the estimator is unbiased.

Now, we analyze the second moment of the estimator:

$$\mathbb{E}_{\boldsymbol{v}}\left[\|g(\boldsymbol{v})\|^2\right] = \mathbb{E}_{\boldsymbol{v}}\left[\left\|(\boldsymbol{v}^\top \nabla f(\boldsymbol{w}))\boldsymbol{v} + \mathcal{O}(\epsilon)\boldsymbol{v}\right\|^2\right] \tag{13}$$

$$= \mathbb{E}_{\boldsymbol{v}}\left[(\boldsymbol{v}^\top \nabla f(\boldsymbol{w}))^2\|\boldsymbol{v}\|^2\right] + \mathcal{O}(\epsilon^2)\mathbb{E}_{\boldsymbol{v}}\left[\|\boldsymbol{v}\|^2\right]. \tag{14}$$

Using the known expectation property of Gaussian vectors:

$$\mathbb{E}\left[\boldsymbol{v}\boldsymbol{v}^\top \|\boldsymbol{v}\|^2\right] = (d+2)I_d, \tag{15}$$

we obtain:

$$\mathbb{E}_{\boldsymbol{v}}\left[\|g(\boldsymbol{v})\|^2\right] = \mathbb{E}_{\boldsymbol{v}}\left[\text{Tr}((\boldsymbol{v}\boldsymbol{v}^\top)\nabla f(\boldsymbol{w})\nabla f(\boldsymbol{w})^\top \boldsymbol{v}\boldsymbol{v}^\top)\right] + \mathcal{O}(\epsilon^2)d \tag{16}$$

$$= \text{Tr}(\nabla f(\boldsymbol{w})\nabla f(\boldsymbol{w})^\top)(d+2) + \mathcal{O}(\epsilon^2)d \tag{17}$$

$$= \|\nabla f(\boldsymbol{w})\|^2 (d+2) + \mathcal{O}(\epsilon^2)d. \tag{18}$$

$\square$

**Lemma I.4** (Variance of Gradient Estimate of the Central Finite Difference). *Let $\hat{g}(\boldsymbol{v})$ be the central finite difference gradient estimator using $n$ perturbations $\boldsymbol{v}_1, \ldots, \boldsymbol{v}_n \sim \mathcal{N}(0, I_d)$, given by*

$$\hat{g}(\boldsymbol{w}) = \frac{1}{n}\sum_{i=1}^{n} g(\boldsymbol{v}_i), \quad \text{where} \quad g(\boldsymbol{v}_i) = \frac{f(\boldsymbol{w} + \epsilon\boldsymbol{v}_i) - f(\boldsymbol{w} - \epsilon\boldsymbol{v}_i)}{2\epsilon}\boldsymbol{v}_i. \tag{19}$$

*Assuming the finite-difference step $\epsilon$, the variance of the estimator satisfies:*

$$Var[\hat{g}(\boldsymbol{w})] = \frac{1}{n}\left(\|\nabla f(\boldsymbol{w})\|^2 (d+1) + \mathcal{O}(\epsilon^2)d\right). \tag{20}$$

*This result shows that the variance of the gradient estimator scales as $\mathcal{O}((d+1)/n)$, which quantifies how the dimension $d$ and the number of samples $n$ influence the estimator's variance.*

*Proof.* We can derive the variance of the estimator

$$\hat{g}(\boldsymbol{w}) = \frac{1}{n}\sum_{i=1}^{n}g(\boldsymbol{v}_i), \text{ with } g(\boldsymbol{v}_i) = \frac{f(\boldsymbol{w}+\epsilon\boldsymbol{v}_i) - f(\boldsymbol{w}-\epsilon\boldsymbol{v}_i)}{2\epsilon}\boldsymbol{v}_i$$

by computing

$$\text{Var}[\hat{g}(\boldsymbol{w})] = \mathbb{E}_{\boldsymbol{v}}[||\hat{g}(\boldsymbol{w})||^2] - ||\mathbb{E}_{\boldsymbol{v}}[\hat{g}(\boldsymbol{w})]||^2. \tag{21}$$

For clarity, in the following we assume that the finite-difference step is chosen so that the bias is negligible (i.e. the estimator is unbiased according to Lemma I.3, so that $\mathbb{E}_{\boldsymbol{v}}[g(\boldsymbol{v}_i)] = \nabla f(\boldsymbol{w})$ and $\mathbb{E}_{\boldsymbol{v}}[\hat{g}(\boldsymbol{w})] = \nabla f(\boldsymbol{w})$. (We can add the higher-order remainder later.)

We write the second moment (squared norm) of $\hat{g}$ as

$$\mathbb{E}_{\boldsymbol{v}}[||\hat{g}(\boldsymbol{w})||^2] = \mathbb{E}_{\boldsymbol{v}}\left[\left\|\frac{1}{n}\sum_{i=1}^{n}g(\boldsymbol{v}_i)\right\|^2\right] = \frac{1}{n^2}\mathbb{E}_{\boldsymbol{v}}\left[\sum_{i=1}^{n}\sum_{j=1}^{n}g(\boldsymbol{v}_i)^\top g(\boldsymbol{v}_j)\right]. \tag{22}$$

We split the sum into diagonal and off-diagonal parts:

$$\mathbb{E}_{\boldsymbol{v}}[||\hat{g}(\boldsymbol{w})||^2] = \frac{1}{n^2}\left(\sum_{i=1}^{n}\mathbb{E}_{\boldsymbol{v}}[||g(\boldsymbol{v}_i)||^2] + \sum_{i\neq j}\mathbb{E}[g(\boldsymbol{v}_i)^\top g(\boldsymbol{v}_j)]\right). \tag{23}$$

Since the $g_i$ are independent,

$$\mathbb{E}_{\boldsymbol{v}}[g(\boldsymbol{v}_i)^\top g(\boldsymbol{v}_j)] = \mathbb{E}_{\boldsymbol{v}}[g(\boldsymbol{v}_i)]^\top \mathbb{E}_{\boldsymbol{v}}[g(\boldsymbol{v}_j)] = \nabla f(\boldsymbol{w})^\top \nabla f(\boldsymbol{w}) = ||\nabla f(\boldsymbol{w})||^2 \quad \text{for } i \neq j. \tag{24}$$

Thus,

$$\mathbb{E}_{\boldsymbol{v}}[||\hat{g}(\boldsymbol{w})||^2] = \frac{1}{n^2}(n\mathbb{E}_{\boldsymbol{v}}[||g(\boldsymbol{v})||^2] + n(n-1)||\nabla f(\boldsymbol{w})||^2). \tag{25}$$

Plugging the above result, along with the result derived from Lemma I.3 on $||\mathbb{E}_{\boldsymbol{v}}[\hat{g}(\boldsymbol{w})]||^2 = ||\nabla f(\boldsymbol{w})||^2$, in Equation 21,

$$\text{Var}[\hat{g}(\boldsymbol{w})] = \frac{1}{n^2}\left(n\mathbb{E}_{\boldsymbol{v}}[||g(\boldsymbol{v})||^2] + n(n-1)||\nabla f(\boldsymbol{w})||^2\right) - ||\nabla f(\boldsymbol{w})||^2 \tag{26}$$

$$= \frac{1}{n}\left(\mathbb{E}_{\boldsymbol{v}}[||g(\boldsymbol{v})||^2] - ||\nabla f(\boldsymbol{w})||^2\right). \tag{27}$$

The second moment of the estimator was derived in Lemma I.3, in Equation 18. We use that result in the above equation as follows,

$$\text{Var}[\hat{g}(\boldsymbol{w})] = \frac{1}{n}\left(||\nabla f(\boldsymbol{w})||^2(d+2) + \mathcal{O}(\epsilon^2)d - ||\nabla f(\boldsymbol{w})||^2\right) \tag{28}$$

$$= \frac{1}{n}\left(||\nabla f(\boldsymbol{w})||^2(d+1) + \mathcal{O}(\epsilon^2)d\right) \tag{29}$$

This leads us to a variance bound that scales as $\frac{d+1}{n}$ times $||\nabla f(\theta)||^2$ (plus a $\mathcal{O}(\epsilon^2)$ contribution), which exhibits the dependence of variance of the estimator $\hat{g}(\boldsymbol{w})$ on the dimension $d$ and the number of samples $n$. □

> The key difference between the above two lemmas and the next two lemmas is that the central finite difference estimator introduces a small $O(\epsilon)$ bias due to numerical approximation, whereas the forward-mode AD estimator is exactly unbiased. Additionally, the second moment of the central finite difference estimator includes an extra $O(\epsilon^2)d$ term, which is absent in forward-mode AD, making the latter more precise.

**Lemma I.5** (Bias of Gradient Estimate of Forward-mode Auto Differentiation)**.** *Let $g(\boldsymbol{v})$ be the gradient estimate obtained using the central finite difference method with a perturbation vector $\boldsymbol{v} \sim \mathcal{N}(0, I_d)$. Then, the expectation of the estimator satisfies*

$$\mathbb{E}_{\boldsymbol{v}}[g(\boldsymbol{v})] = \nabla f(\boldsymbol{w}), \tag{30}$$

*implying that the central finite difference gradient estimator is unbiased. Furthermore, the second moment of the estimator satisfies*

$$\mathbb{E}_{\boldsymbol{v}}\left[||g(\boldsymbol{v})||^2\right] = ||\nabla f(\boldsymbol{w})||^2(d+2). \tag{31}$$

*Proof.* We start with one evaluation of forward-mode auto differentiation, for a perturbation $\boldsymbol{v} \in \mathbb{R}^d$. The full derivation is given in Theorem I.9,

$$g(\boldsymbol{v}) = \left(\boldsymbol{v}^\top \nabla f(\boldsymbol{w})\right) \boldsymbol{v}. \tag{32}$$

In order to measure the bias of the above gradient estimate, we take expectation with respect to the randomness of $\boldsymbol{v}$,

$$\mathbb{E}_{\boldsymbol{v}}\left[g(\boldsymbol{v})\right] = \mathbb{E}_{\boldsymbol{v}}\left[(\boldsymbol{v}^\top \nabla f(\boldsymbol{w}))\boldsymbol{v}\right] \tag{33}$$

$$= \nabla f(\boldsymbol{w})\mathbb{E}_{\boldsymbol{v}}\left[\boldsymbol{v}^\top \boldsymbol{v}\right] = \nabla f(\boldsymbol{w})I_d \tag{34}$$

$$\therefore \mathbb{E}_{\boldsymbol{v}}\left[g(\boldsymbol{v})\right] = \nabla f(\boldsymbol{w}). \tag{35}$$

This shows that the estimator is unbiased.

Now, we analyze the second moment of the estimator:

$$\mathbb{E}_{\boldsymbol{v}}\left[\|g(\boldsymbol{v})\|^2\right] = \mathbb{E}_{\boldsymbol{v}}\left[\left\|(\boldsymbol{v}^\top \nabla f(\boldsymbol{w}))\boldsymbol{v}\right\|^2\right] = \mathbb{E}_{\boldsymbol{v}}\left[(\boldsymbol{v}^\top \nabla f(\boldsymbol{w}))^2\|\boldsymbol{v}\|^2\right]. \tag{36}$$

Using the known expectation property of Gaussian vectors:

$$\mathbb{E}\left[\boldsymbol{v}\boldsymbol{v}^\top \|\boldsymbol{v}\|^2\right] = (d+2)I_d, \tag{37}$$

We obtain:

$$\mathbb{E}_{\boldsymbol{v}}\left[\|g(\boldsymbol{v})\|^2\right] = \mathbb{E}_{\boldsymbol{v}}\left[\text{Tr}((\boldsymbol{v}\boldsymbol{v}^\top)\nabla f(\boldsymbol{w})\nabla f(\boldsymbol{w})^\top \boldsymbol{v}\boldsymbol{v}^\top)\right] \tag{38}$$

$$= \text{Tr}(\nabla f(\boldsymbol{w})\nabla f(\boldsymbol{w})^\top)(d+2) = \|\nabla f(\boldsymbol{w})\|^2(d+2). \tag{39}$$

$\square$

**Lemma I.6** (Variance of Gradient Estimate of Forward-mode Auto Differentiation). *Let $\hat{g}(\boldsymbol{v})$ be the central finite difference gradient estimator using $n$ perturbations $\boldsymbol{v}_1, \ldots, \boldsymbol{v}_n \sim \mathcal{N}(0, I_d)$, given by*

$$\hat{g}(\boldsymbol{w}) = \frac{1}{n}\sum_{i=1}^n g(\boldsymbol{v}_i), \quad where \quad g(\boldsymbol{v}_i) = (\boldsymbol{v}_i^\top \nabla f(\boldsymbol{w}))\boldsymbol{v}_i. \tag{40}$$

*Assuming the finite-difference step $\epsilon$, the variance of the estimator satisfies:*

$$Var[\hat{g}(\boldsymbol{w})] = \frac{1}{n}\left(\|\nabla f(\boldsymbol{w})\|^2(d+1)\right). \tag{41}$$

*This result shows that the variance of the gradient estimator scales as $\mathcal{O}((d+1)/n)$, which quantifies how the dimension $d$ and the number of samples $n$ influence the estimator's variance.*

*Proof.* We can derive the variance of the estimator

$$\hat{g}(\boldsymbol{w}) = \frac{1}{n}\sum_{i=1}^n g(\boldsymbol{v}_i), \text{ with } g(\boldsymbol{v}_i) = (\boldsymbol{v}_i^\top \nabla f(\boldsymbol{w}))\boldsymbol{v}_i$$

by computing

$$\text{Var}[\hat{g}(\boldsymbol{w})] = \mathbb{E}_{\boldsymbol{v}}[\|\hat{g}(\boldsymbol{w})\|^2] - \|\mathbb{E}_{\boldsymbol{v}}[\hat{g}(\boldsymbol{w})]\|^2. \tag{42}$$

The estimator is unbiased according to Lemma I.5, so that $\mathbb{E}_{\boldsymbol{v}}[g(\boldsymbol{v}_i)] = \nabla f(\boldsymbol{w})$ and $\mathbb{E}_{\boldsymbol{v}}[\hat{g}(\boldsymbol{w})] = \nabla f(\boldsymbol{w})$.

We write the second moment (squared norm) of $\hat{g}$ as

$$\mathbb{E}_{\boldsymbol{v}}[\|\hat{g}(\boldsymbol{w})\|^2] = \mathbb{E}_{\boldsymbol{v}}\left[\left\|\frac{1}{n}\sum_{i=1}^n g(\boldsymbol{v}_i)\right\|^2\right] = \frac{1}{n^2}\mathbb{E}_{\boldsymbol{v}}\left[\sum_{i=1}^n\sum_{j=1}^n g(\boldsymbol{v}_i)^\top g(\boldsymbol{v}_j)\right]. \tag{43}$$

We split the sum into diagonal and off-diagonal parts:

$$\mathbb{E}_{\boldsymbol{v}}[\|\hat{g}(\boldsymbol{w})\|^2] = \frac{1}{n^2}\left(\sum_{i=1}^n \mathbb{E}_{\boldsymbol{v}}[\|g(\boldsymbol{v}_i)\|^2] + \sum_{i \neq j}\mathbb{E}[g(\boldsymbol{v}_i)^\top g(\boldsymbol{v}_j)]\right). \tag{44}$$

Since the $g_i$ are independent,

$$\mathbb{E}_{\boldsymbol{v}}[g(\boldsymbol{v}_i)^\top g(\boldsymbol{v}_j)] = \mathbb{E}_{\boldsymbol{v}}[g(\boldsymbol{v}_i)]^\top \mathbb{E}_{\boldsymbol{v}}[g(\boldsymbol{v}_j)] = \nabla f(\boldsymbol{w})^\top \nabla f(\boldsymbol{w}) = ||\nabla f(\boldsymbol{w})||^2 \quad \text{for } i \neq j. \quad (45)$$

Thus,

$$\mathbb{E}_{\boldsymbol{v}}[||\hat{g}(\boldsymbol{w})||^2] = \frac{1}{n^2}(n\mathbb{E}_{\boldsymbol{v}}[||g(\boldsymbol{v})||^2] + n(n-1)||\nabla f(\boldsymbol{w})||^2). \quad (46)$$

Plugging the above result, along with the result derived from Lemma I.3 on $||\mathbb{E}_{\boldsymbol{v}}[\hat{g}(\boldsymbol{w})]||^2 = ||\nabla f(\boldsymbol{w})||^2$, in Equation 42,

$$\text{Var}[\hat{g}(\boldsymbol{w})] = \frac{1}{n^2}\left(n\mathbb{E}_{\boldsymbol{v}}[||g(\boldsymbol{v})||^2] + n(n-1)||\nabla f(\boldsymbol{w})||^2\right) - ||\nabla f(\boldsymbol{w})||^2 \quad (47)$$

$$= \frac{1}{n}\left(\mathbb{E}_{\boldsymbol{v}}[||g(\boldsymbol{v})||^2] - ||\nabla f(\boldsymbol{w})||^2\right). \quad (48)$$

The second moment of the estimator was derived in Lemma I.5, in Equation 39. We use that result in the above equation as follows,

$$\text{Var}[\hat{g}(\boldsymbol{w})] = \frac{1}{n}\left(||\nabla f(\boldsymbol{w})||^2(d+2) - ||\nabla f(\boldsymbol{w})||^2\right) \quad (49)$$

$$= \frac{1}{n}\left(||\nabla f(\boldsymbol{w})||^2(d+1)\right) \quad (50)$$

This leads us to a variance bound that scales as $\frac{d+1}{n}$ times $||\nabla f(\theta)||^2$, which exhibits the dependence of variance of the estimator $\hat{g}(\boldsymbol{w})$ on the dimension $d$ and the number of samples $n$. $\qquad\square$

### I.4 THEOREMS

The given analysis for all gradient computation methods is for **a non-convex objective** $f$.
We begin by reiterating the descent lemma applied to gradients computed by backpropagation.

**Theorem I.7** (Error Bound of Backpropagation). *Let $f : \mathbb{R}^d \to \mathbb{R}$ be a differentiable, L-smooth function. Consider the gradient descent update rule:*

$$\boldsymbol{w}_{t+1} = \boldsymbol{w}_t - \eta\nabla f(\boldsymbol{w}_t),$$

*where $\eta$ is the step size (learning rate). Suppose $0 < \eta \leq \frac{1}{L}$. Then, after $T$ iterations, the minimum gradient norm satisfies the following bound:*

$$\min_{t\in[T]}||\nabla f(\boldsymbol{w}_t)||^2 \leq \frac{2L}{T}\left(f(\boldsymbol{w}_1) - f(\boldsymbol{w}_T)\right),$$

*This bound shows that gradient descent achieves an $\mathcal{O}(\frac{1}{T})$ convergence rate in terms of gradient norm, which is the optimal rate for first-order methods in smooth optimization.*

*Proof.* Using Assumption I.2, we apply the smoothness condition, which gives the following quadratic upper bound:

$$f(\boldsymbol{w}_{t+1}) \leq f(\boldsymbol{w}_t) + \nabla f(\boldsymbol{w}_t)^\top(\boldsymbol{w}_{t+1} - \boldsymbol{w}_t) + \frac{L}{2}||\boldsymbol{w}_{t+1} - \boldsymbol{w}_t||^2 \quad (51)$$

Substituting the gradient descent update rule Equation 3, we obtain,

$$\therefore f(\boldsymbol{w}_{t+1}) \leq f(\boldsymbol{w}_t) - \eta||\nabla f(\boldsymbol{w}_t)||^2 + \frac{L}{2}\eta^2||\nabla f(\boldsymbol{w}_t)||^2 \quad (52)$$

Rearranging the above terms,

$$\therefore f(\boldsymbol{w}_{t+1}) - f(\boldsymbol{w}_t) \leq -\eta||\nabla f(\boldsymbol{w}_t)||^2 + \frac{L\eta^2}{2}||\nabla f(\boldsymbol{w}_t)||^2 = -\left(\eta - \frac{L\eta^2}{2}\right)||\nabla f(\boldsymbol{w}_t)||^2 \quad (53)$$

To ensure progress in minimizing $f(\boldsymbol{w})$, we need the term $(1 - \frac{L\eta}{2})$ to be positive. Hence we assume $\eta \leq \frac{1}{L}$, along with $0 < \eta$,

$$\therefore f(\boldsymbol{w}_{t+1}) - f(\boldsymbol{w}_t) \leq -\frac{1}{2L}||\nabla f(\boldsymbol{w}_t)||^2 \quad (54)$$

Summing over $t = 1$ to $t = T$,

$$\sum_{t=1}^{T} \left(f(\boldsymbol{w}_{t+1}) - f(\boldsymbol{w}_t)\right) \leq -\frac{1}{2L} \sum_{t=1}^{T} ||\nabla f(\boldsymbol{w}_t)||^2 \tag{55}$$

The left-hand side forms a telescoping sum, resulting in

$$f(\boldsymbol{w}_{T+1}) - f(\boldsymbol{w}_1) \leq -\frac{1}{2L} \sum_{t=1}^{T} ||\nabla f(\boldsymbol{w}_t)||^2 \tag{56}$$

$$\frac{1}{T} \sum_{t=1}^{T} ||\nabla f(\boldsymbol{w}_t)||^2 \leq \frac{2L}{T} \left(f(\boldsymbol{w}_1) - f(\boldsymbol{w}_{T+1})\right) \tag{57}$$

Using the definition of optimality gap from Equation I.1,

$$\min_{t \in [T]} ||\nabla f(\boldsymbol{w}_t)||^2 \leq \frac{2L}{T} \left(f(\boldsymbol{w}_t) - f(\boldsymbol{w}_T)\right) \tag{58}$$

Thus, the optimality gap reduces at a rate of $\mathcal{O}(\frac{1}{T})$, given $\eta \leq \frac{1}{L}$. $\qquad \square$

Next, we will give a similar treatment to the gradients derived from zero-order finite differences,

**Theorem I.8** (Error Bound of Zero-Order Optimization). *Consider a function $f : \mathbb{R}^d \to \mathbb{R}$ that is L-smooth. Let the central finite-difference gradient estimator with $n$ perturbations per iteration, where each perturbation vector $\boldsymbol{v}_i$ is sampled independently from $\mathcal{N}(0, I_d)$ and step size $\eta$ be*

$$\hat{g}(\boldsymbol{w}_t) = \frac{1}{n} \sum_{i=1}^{n} \left(\frac{f(\boldsymbol{w}_t + \epsilon \boldsymbol{v}_i) - f(\boldsymbol{w}_t - \epsilon \boldsymbol{v}_i)}{2\epsilon} \boldsymbol{v}_i\right).$$

*Then, the expected average squared gradient norm is bounded by*

$$\frac{1}{T} \sum_{t=1}^{T} ||\nabla f(\boldsymbol{w}_t)||^2 \leq \frac{f(\boldsymbol{w}_1) - f(\boldsymbol{w}_T)}{\eta T \left[1 - \frac{L\eta}{2}\left(1 + \frac{d+1}{n}\right)\right]} + \frac{Ld\eta^2}{2n} \mathcal{O}(\epsilon^2), \tag{59}$$

*To ensure convergence, the step size must satisfy*

$$\eta < \frac{2}{L\left(1 + \frac{d+1}{n}\right)}. \tag{60}$$

*This result highlights how the convergence rate depends on the dimension d, the number of perturbations $n$, and the perturbation magnitude $\epsilon$. Specifically, a larger $d$ or a smaller $n$ increases the bound, implying slower convergence.*

*Proof.* The central finite-difference gradient estimator for $n$ perturbations per iteration is

$$\hat{g}(\boldsymbol{w}_t) = \frac{1}{n} \sum_{i=1}^{n} \left(\frac{f(\boldsymbol{w}_t + \epsilon \boldsymbol{v}_i) - f(\boldsymbol{w}_t - \epsilon \boldsymbol{v}_i)}{2\epsilon} \boldsymbol{v}_i\right) \tag{61}$$

where each $\boldsymbol{v}_i \in \mathbb{R}^d$ is a perturbation drawn from a Gaussian distribution $\mathcal{N}(0, 1)$.

Assuming that $f$ is sufficiently smooth so that the following Taylor expansions are valid,

$$f(\boldsymbol{w} + \epsilon \boldsymbol{v}) = f(\boldsymbol{w}) + \epsilon \boldsymbol{v}^\top \nabla f(\boldsymbol{w}) + O(\epsilon^2), \text{ and} \tag{62}$$

$$f(\boldsymbol{w} - \epsilon \boldsymbol{v}) = f(\boldsymbol{w}) - \epsilon \boldsymbol{v}^\top \nabla f(\boldsymbol{w}) + O(\epsilon^2) \tag{63}$$

Subtracting the above two expansions yields:

$$f(\boldsymbol{w} + \epsilon \boldsymbol{v}) - f(\boldsymbol{w} - \epsilon \boldsymbol{v}) = 2\epsilon \boldsymbol{v}^\top \nabla f(\boldsymbol{w}) + \mathcal{O}(\epsilon^2) \tag{64}$$

Plugging the above result in Equation 61,

$$\hat{g}(\boldsymbol{w}_t) = \frac{1}{n} \sum_{i=1}^{n} \underbrace{\left((\boldsymbol{v}_i^\top \nabla f(\boldsymbol{w}))\boldsymbol{v}_i + \mathcal{O}(\epsilon)\boldsymbol{v}_i\right)}_{g(\boldsymbol{v}_i)} \tag{65}$$

*Now we will use the derived $\hat{g}(\boldsymbol{w}_t)$ in the descent lemma:*

Similar to Theorem I.7, using the Assumption I.2, we apply the smoothness on $f$:

$$f(\boldsymbol{w}_{t+1}) \le f(\boldsymbol{w}_t) + \nabla f(\boldsymbol{w}_t)^\top (\boldsymbol{w}_{t+1} - \boldsymbol{w}_t) + \frac{L}{2}||\boldsymbol{w}_{t+1} - \boldsymbol{w}_t||^2 \tag{66}$$

For the gradient descent update with central finite differences, we set the model update rule as

$$\boldsymbol{w}_{t+1} = \boldsymbol{w}_t - \eta \hat{g}(\boldsymbol{w}_t). \tag{67}$$

Plugging the model update rule into the smoothness inequality,

$$f(\boldsymbol{w}_{t+1}) \le f(\boldsymbol{w}_t) - \eta \nabla f(\boldsymbol{w}_t)^\top \hat{g}(\boldsymbol{w}_t) + \frac{L\eta^2}{2}||\hat{g}(\boldsymbol{w}_t)||^2 \tag{68}$$

Taking expectation conditioned on $\boldsymbol{v} \sim \mathcal{N}(0, I_d)$,

$$f(\boldsymbol{w}_{t+1}) \le f(\boldsymbol{w}_t) - \eta \nabla f(\boldsymbol{w}_t)^\top \underbrace{\mathbb{E}_{\boldsymbol{v}}[\hat{g}(\boldsymbol{w}_t)]}_{\text{Term}_1} + \frac{L\eta^2}{2} \underbrace{\mathbb{E}_{\boldsymbol{v}}[||\hat{g}(\boldsymbol{w}_t)||^2]}_{\text{Term}_2} \tag{69}$$

Solving **Term**$_1$ and **Term**$_2$ separately,

**Term**$_1$: From Lemma I.3, we get $\mathbb{E}[g(\boldsymbol{v})] = \nabla f(\boldsymbol{w})$, which also gets us

$$\mathbb{E}[\hat{g}(\boldsymbol{w})] = \frac{1}{n} \sum_{i=1}^{n} \mathbb{E}[g(\boldsymbol{v})] = \nabla f(\boldsymbol{w})$$

**Term**$_2$: The error of $\hat{g}(\boldsymbol{w})$ is measured by $\delta$,

$$||\hat{g}(\boldsymbol{w})||^2 = ||\nabla f(\boldsymbol{w}) + \delta||^2 = ||\nabla f(\boldsymbol{w})||^2 + 2\nabla f(\boldsymbol{w})^\top \delta + ||\delta||^2 \tag{70}$$

Taking expectation and noting that $\mathbb{E}[\delta] = 0$ and $\mathbb{E}[||\delta||^2] = \text{Var}[\hat{g}(\boldsymbol{w})]$,

$$\mathbb{E}_{\boldsymbol{v}}[||\hat{g}(\boldsymbol{w})||^2] = ||\nabla f(\boldsymbol{w})||^2 + \text{Var}[\hat{g}(\boldsymbol{w})] \tag{71}$$

Using Lemma I.4 to get the bound of $\text{Var}[\hat{g}(\boldsymbol{w})]$,

$$\mathbb{E}_{\boldsymbol{v}}[||\hat{g}(\boldsymbol{w})||^2] = ||\nabla f(\boldsymbol{w})||^2 + \frac{d+1}{n}||\nabla f(\boldsymbol{w})||^2 + \frac{d}{n}\mathcal{O}(\epsilon^2) \tag{72}$$

**Back to Equation 69, plugging in Term$_1$ and Term$_2$:**

$$f(\boldsymbol{w}_{t+1}) \le f(\boldsymbol{w}_t) - \eta \nabla f(\boldsymbol{w}_t)^\top \nabla f(\boldsymbol{w}_t) + \frac{L\eta^2}{2}\left(\left(1 + \frac{d+1}{n}\right)||\nabla f(\boldsymbol{w})||^2 + \frac{d}{n}\mathcal{O}(\epsilon^2)\right) \tag{73}$$

$$= f(\boldsymbol{w}_t) - \eta ||\nabla f(\boldsymbol{w}_t)||^2 + \frac{L\eta^2}{2}\left(1 + \frac{d+1}{n}\right)||\nabla f(\boldsymbol{w})||^2 + \frac{Ld\eta^2}{2n}\mathcal{O}(\epsilon^2) \tag{74}$$

Grouping the terms involving $||\nabla f(\boldsymbol{w})||^2$,

$$f(\boldsymbol{w}_{t+1}) \le f(\boldsymbol{w}_t) - \eta\left[1 - \frac{L\eta}{2}\left(1 + \frac{d+1}{n}\right)\right]||\nabla f(\boldsymbol{w}_t)||^2 + \frac{Ld\eta^2}{2n}\mathcal{O}(\epsilon^2) \tag{75}$$

This inequality shows that, provided the step size $\eta$ is small enough so that,

$$1 - \frac{L\eta}{2}\left(1 + \frac{d+1}{n}\right) > 0$$

Summing the inequality over epochs $t = 1$ to $T$:

$$f(\boldsymbol{w}_T) - f(\boldsymbol{w}_1) \le -\eta\left[1 - \frac{L\eta}{2}\left(1 + \frac{d+1}{n}\right)\right]\sum_{t=1}^{T}||\nabla f(\boldsymbol{w}_t)||^2 + \frac{Ld\eta^2 T}{2n}\mathcal{O}(\epsilon^2) \tag{76}$$

Rearranging the terms give us,

$$\sum_{t=1}^{T} ||\nabla f(\boldsymbol{w}_t)||^2 \leq \frac{f(\boldsymbol{w}_1) - f(\boldsymbol{w}_T)}{\eta \left[1 - \frac{L\eta}{2}\left(1 + \frac{d+1}{n}\right)\right]} + \frac{Ld\eta^2 T}{2n}\mathcal{O}(\epsilon^2) \tag{77}$$

Dividing by $T$ gives the bound on the average squared gradient norm:

$$\frac{1}{T}\sum_{t=1}^{T} ||\nabla f(\boldsymbol{w}_t)||^2 \leq \frac{f(\boldsymbol{w}_1) - f(\boldsymbol{w}_T)}{\eta T \left[1 - \frac{L\eta}{2}\left(1 + \frac{d+1}{n}\right)\right]} + \frac{Ld\eta^2}{2n}\mathcal{O}(\epsilon^2) \tag{78}$$

To ensure that $1 - \frac{L\eta}{2}\left(1 + \frac{d+1}{n}\right) > 0$, the step size $\eta$ must be chosen so that

$$\eta < \frac{2}{L\left(1 + \frac{d+1}{n}\right)}. \tag{79}$$

As the dimension $d$ increases (or as the number of samples $n$ decreases), the factor

$$\frac{L\eta}{2}\left(1 + \frac{d+1}{n}\right)$$

increases. This makes

$$1 - \frac{L\eta}{2}\left(1 + \frac{d+1}{n}\right)$$

smaller, which in turn makes the entire bound larger. In other words, a larger $d$ (or a smaller $n$) results in a worse (higher) error bound. This interplay of $d$ and $n$ also puts limitations on the order of $\eta$, keeping the learning rate quite small for stable learning. $\qquad\square$

Moving on, we will get the convergence bound of the gradients derived from forward-mode auto differentiation.

> The key difference between the two theorems is that the error bound for zero-order optimization includes an additional $\mathcal{O}(\epsilon^2)$ term due to the finite-difference approximation, whereas the bound for forward-mode AD is exact and free from such errors. This makes forward-mode AD theoretically more efficient, as it avoids the additional error introduced by numerical differentiation while maintaining the same dependency on dimension $d$ and number of perturbations $n$.

**Theorem I.9** (Error Bound of Forward-mode Auto Differentiation). *Consider a function $f : \mathbb{R}^d \to \mathbb{R}$ that is $L$-smooth. Let the forward-mode AD gradient estimator with $n$ perturbations per iteration, where each perturbation vector $\boldsymbol{v}_i$ is sampled independently from $\mathcal{N}(0, I_d)$ and step size $\eta$ be*

$$\hat{g}(\boldsymbol{w}_t) = \frac{1}{n}\sum_{i=1}^{n} \left(\left(\boldsymbol{v}_i^\top \nabla f(\boldsymbol{w}_t)\right)\boldsymbol{v}_i\right).$$

*Then, the expected average squared gradient norm is bounded by*

$$\frac{1}{T}\sum_{t=1}^{T} ||\nabla f(\boldsymbol{w}_t)||^2 \leq \frac{f(\boldsymbol{w}_1) - f(\boldsymbol{w}_T)}{\eta T \left[1 - \frac{L\eta}{2}\left(1 + \frac{d+1}{n}\right)\right]}, \tag{80}$$

*To ensure convergence, the step size must satisfy*

$$\eta < \frac{2}{L\left(1 + \frac{d+1}{n}\right)}. \tag{81}$$

*This result highlights how the convergence rate depends on the dimension $d$, the number of perturbations $n$, and the perturbation magnitude $\epsilon$. Specifically, a larger $d$ or a smaller $n$ increases the bound, implying slower convergence.*

*Proof.* The forward-mode AD gradient estimator for $n$ perturbations per iteration is

$$\hat{g}(\boldsymbol{w}_t) = \frac{1}{n} \sum_{i=1}^{n} \left( (\nabla \boldsymbol{v}_i^\top f(\boldsymbol{w}_t)) \boldsymbol{v}_i \right) \tag{82}$$

where each $\boldsymbol{v}_i \in \mathbb{R}^d$ is a perturbation drawn from a Gaussian distribution $\mathcal{N}(0,1)$.

We will use $\hat{g}(\boldsymbol{w}_t)$ in the descent lemma. Similar to Theorem I.8, using the Assumption I.2, we apply the smoothness on $f$:

$$f(\boldsymbol{w}_{t+1}) \leq f(\boldsymbol{w}_t) + \nabla f(\boldsymbol{w}_t)^\top (\boldsymbol{w}_{t+1} - \boldsymbol{w}_t) + \frac{L}{2} ||\boldsymbol{w}_{t+1} - \boldsymbol{w}_t||^2 \tag{83}$$

For the gradient descent update with forward-mode AD, we set the model update rule as

$$\boldsymbol{w}_{t+1} = \boldsymbol{w}_t - \eta \hat{g}(\boldsymbol{w}_t). \tag{84}$$

Plugging the model update rule into the smoothness inequality,

$$f(\boldsymbol{w}_{t+1}) \leq f(\boldsymbol{w}_t) - \eta \nabla f(\boldsymbol{w}_t)^\top \hat{g}(\boldsymbol{w}_t) + \frac{L\eta^2}{2} ||\hat{g}(\boldsymbol{w}_t)||^2 \tag{85}$$

Taking expectation conditioned on $\boldsymbol{v} \sim \mathcal{N}(0, I_d)$,

$$f(\boldsymbol{w}_{t+1}) \leq f(\boldsymbol{w}_t) - \eta \nabla f(\boldsymbol{w}_t)^\top \underbrace{\mathbb{E}_{\boldsymbol{v}}[\hat{g}(\boldsymbol{w}_t)]}_{\text{Term}_1} + \frac{L\eta^2}{2} \underbrace{\mathbb{E}_{\boldsymbol{v}}[||\hat{g}(\boldsymbol{w}_t)||^2]}_{\text{Term}_2} \tag{86}$$

Solving **Term$_1$** and **Term$_2$** separately,

**Term$_1$:**   From Lemma I.5, we get $\mathbb{E}[g(\boldsymbol{v})] = \nabla f(\boldsymbol{w})$, which also gets us

$$\mathbb{E}[\hat{g}(\boldsymbol{w})] = \frac{1}{n} \sum_{i=1}^{n} \mathbb{E}[g(\boldsymbol{v})] = \nabla f(\boldsymbol{w})$$

**Term$_2$:**   The error of $\hat{g}(\boldsymbol{w})$ is measured by $\delta$,

$$||\hat{g}(\boldsymbol{w})||^2 = ||\nabla f(\boldsymbol{w}) + \delta||^2 = ||\nabla f(\boldsymbol{w})||^2 + 2\nabla f(\boldsymbol{w})^\top \delta + ||\delta||^2 \tag{87}$$

Taking expectation and noting that $\mathbb{E}[\delta] = 0$ and $\mathbb{E}[||\delta||^2] = \text{Var}[\hat{g}(\boldsymbol{w})]$,

$$\mathbb{E}_{\boldsymbol{v}}[||\hat{g}(\boldsymbol{w})||^2] = ||\nabla f(\boldsymbol{w})||^2 + \text{Var}[\hat{g}(\boldsymbol{w})] \tag{88}$$

Using Lemma I.6 to get the bound of $\text{Var}[\hat{g}(\boldsymbol{w})]$,

$$\mathbb{E}_{\boldsymbol{v}}[||\hat{g}(\boldsymbol{w})||^2] = ||\nabla f(\boldsymbol{w})||^2 + \frac{d+1}{n} ||\nabla f(\boldsymbol{w})||^2 \tag{89}$$

**Back to Equation 86, plugging in Term$_1$ and Term$_2$:**

$$f(\boldsymbol{w}_{t+1}) \leq f(\boldsymbol{w}_t) - \eta \nabla f(\boldsymbol{w}_t)^\top \nabla f(\boldsymbol{w}_t) + \frac{L\eta^2}{2} \left(1 + \frac{d+1}{n}\right) ||\nabla f(\boldsymbol{w})||^2 \tag{90}$$

$$= f(\boldsymbol{w}_t) - \eta ||\nabla f(\boldsymbol{w}_t)||^2 + \frac{L\eta^2}{2} \left(1 + \frac{d+1}{n}\right) ||\nabla f(\boldsymbol{w})||^2 \tag{91}$$

Grouping the terms involving $||\nabla f(\boldsymbol{w})||^2$,

$$f(\boldsymbol{w}_{t+1}) \leq f(\boldsymbol{w}_t) - \eta \left[1 - \frac{L\eta}{2}\left(1 + \frac{d+1}{n}\right)\right] ||\nabla f(\boldsymbol{w}_t)||^2 \tag{92}$$

This inequality shows that, provided the step size $\eta$ is small enough so that,

$$1 - \frac{L\eta}{2}\left(1 + \frac{d+1}{n}\right) > 0$$

Summing the inequality over epochs $t = 1$ to $T$:

$$f(\boldsymbol{w}_T) - f(\boldsymbol{w}_1) \le -\eta \left[ 1 - \frac{L\eta}{2} \left( 1 + \frac{d+1}{n} \right) \right] \sum_{t=1}^{T} ||\nabla f(\boldsymbol{w}_t)||^2 \tag{93}$$

Rearranging the terms give us,

$$\sum_{t=1}^{T} ||\nabla f(\boldsymbol{w}_t)||^2 \le \frac{f(\boldsymbol{w}_1) - f(\boldsymbol{w}_T)}{\eta \left[ 1 - \frac{L\eta}{2} \left( 1 + \frac{d+1}{n} \right) \right]} \tag{94}$$

Dividing by $T$ gives the bound on the average squared gradient norm:

$$\frac{1}{T} \sum_{t=1}^{T} ||\nabla f(\boldsymbol{w}_t)||^2 \le \frac{f(\boldsymbol{w}_1) - f(\boldsymbol{w}_T)}{\eta T \left[ 1 - \frac{L\eta}{2} \left( 1 + \frac{d+1}{n} \right) \right]} \tag{95}$$

To ensure that $1 - \frac{L\eta}{2} \left( 1 + \frac{d+1}{n} \right) > 0$, the step size $\eta$ must be chosen so that

$$\eta < \frac{2}{L \left( 1 + \frac{d+1}{n} \right)}. \tag{96}$$

As the dimension $d$ increases (or as the number of samples $n$ decreases), the factor

$$\frac{L\eta}{2} \left( 1 + \frac{d+1}{n} \right)$$

increases. This makes

$$1 - \frac{L\eta}{2} \left( 1 + \frac{d+1}{n} \right)$$

smaller, which in turn makes the entire bound larger. In other words – similar to zero-order method – a larger $d$ (or a smaller $n$) results in a worse (higher) error bound. This interplay of $d$ and $n$ also puts limitations on the order of $\eta$, keeping the learning rate quite small for stable learning. $\qquad\square$

**Corollary I.10** (Convergence Rate of ZO under Standard Parameter Choices). *Under the assumptions of Theorem I.8, the zeroth-order method achieves the well-known $\mathcal{O}(d/T)$ convergence rate when the parameters are chosen according to either of the following equivalent strategies:*

1. *Setting the step size to $\eta = \Theta \left( \frac{1}{L(1+\frac{d+1}{n})} \right)$, which yields the rate by balancing the contraction factor in the denominator term; or*

2. *Using the two-point estimator ($n = 1$) with perturbation radius $\epsilon = \mathcal{O}(T^{-1/4})$, so that $\epsilon^2 = \mathcal{O}(T^{-1/2})$ and the variance term becomes $\mathcal{O}(d/T)$.*

*Both parameterizations recover*

$$\min_{t \in [T]} ||\nabla f(w_t)||^2 = \mathcal{O}\left( \frac{d}{T} \right).$$

*While the first approach modulates the learning rate $\eta$, the second adapts the perturbation scale $\epsilon$; in practice both routes give consistent rates, though excessively small $\eta$ (scaling as $1/d$) may be less practical in high dimensions.*

*Proof.* Start from the bound in Theorem I.8:

$$\frac{1}{T} \sum_{t=1}^{T} ||\nabla f(w_t)||^2 \le \frac{f(w_1) - f(w_T)}{\eta T \left[ 1 - \frac{L\eta}{2} \left( 1 + \frac{d+1}{n} \right) \right]} + \frac{Ld\eta^2}{2n} \mathcal{O}(\epsilon^2).$$

We treat the two parameterizations separately.

**(1) Step-size choice.** Set the denominator factor to a constant by choosing

$$1 - \frac{L\eta}{2}\left(1 + \frac{d+1}{n}\right) = \tfrac{1}{2}, \quad \text{so} \quad \eta = \Theta\left(\frac{1}{L(1 + \frac{d+1}{n})}\right).$$

With this choice the first term scales as

$$\frac{f(w_1) - f(w_T)}{\eta T[\cdots]} = \Theta\left(\frac{1}{\eta T}\right) = \Theta\left(\frac{L(1 + \frac{d+1}{n})}{T}\right).$$

When $d \gg n$ this is $\Theta(d/T)$, so the first term already yields $\mathcal{O}(d/T)$. The second term becomes

$$\frac{Ld\eta^2}{2n}\mathcal{O}(\epsilon^2) = \mathcal{O}\left(\frac{Ld}{n} \cdot \frac{1}{L^2(1 + \frac{d+1}{n})^2}\epsilon^2\right) = \mathcal{O}\left(\frac{d}{Ln(1 + \frac{d+1}{n})^2}\epsilon^2\right),$$

which is typically smaller than the first term for reasonable (non-growing) $\epsilon$; hence the overall rate is dominated by $\mathcal{O}(d/T)$.

**(2) Smoothing-radius choice (two-point / $n = 1$).** Take $n = 1$ and set $\epsilon = \mathcal{O}(T^{-1/4})$, so $\epsilon^2 = \mathcal{O}(T^{-1/2})$. Keeping the same $\eta$ scale as above (or any constant-in-$T$ $\eta$ satisfying the step-size constraint), the first term is again $\mathcal{O}(1/(\eta T))$. With $\eta = \Theta(1/(L(1 + (d+1)/n))) \approx \Theta(1/(Ld))$ this yields $\mathcal{O}(d/T)$. The second term becomes

$$\frac{Ld\eta^2}{2}\mathcal{O}(\epsilon^2) = \mathcal{O}\left(Ld \cdot \frac{1}{L^2d^2} \cdot T^{-1/2}\right) = \mathcal{O}\left(\frac{1}{Ld}T^{-1/2}\right),$$

which is negligible compared to $\mathcal{O}(d/T)$ for typical $T$ and moderate $L$. Thus both choices give the stated $\mathcal{O}(d/T)$ rate. $\qquad\square$

**Discussion.**

- **Two equivalent levers.** The corollary emphasizes two ways to recover the classical $\mathcal{O}(d/T)$ bound: (i) scale down the learning rate $\eta$ (reviewer's route), or (ii) scale the perturbation radius $\epsilon$ with $T$ (the alternate route used in our original derivation). Both are valid theoretically and lead to the same asymptotic dependence on $d$ and $T$.

- **Dominant term and constants.** In the parameter regimes of interest the first term (the $1/(\eta T)$-type term) typically dominates and yields the $\Theta(d/T)$ dependency; the variance/truncation term involving $\epsilon^2$ is often smaller when $\epsilon$ is chosen to decay suitably with $T$.

- **Practicality.** Although setting $\eta = \Theta(1/d)$ recovers the rate, such tiny learning rates become impractical as model size grows (since $\eta \to 0$ with $d \to \infty$). The alternative of shrinking $\epsilon$ (e.g., $\epsilon = T^{-1/4}$ gives $\epsilon = 0.1$ at $T = 100$ and $\epsilon = 0.03$ at $T = 1000$) is often more feasible in practice, but it reduces signal-to-noise in finite-sample regimes and may require larger sample or perturbation budgets to get stable estimates.

