# OpenReview forum: "Memory Savings at What Cost? A Study of Alternatives to Backpropagation"
_ICLR.cc/2026/Conference — Submitted to ICLR 2026_

### Official Review · Reviewer_KQZt · 2025-10-19

**Soundness:** 2
**Presentation:** 2
**Contribution:** 1
**Rating:** 2
**Confidence:** 4

**Summary:**

This paper revisits claims that forward-mode automatic differentiation (FMAD) and zeroth-order (ZO) optimization can serve as memory-efficient alternatives to standard backpropagation (BP) for training large models. The authors provide an extensive empirical comparison of BP (including checkpointed variants), FMAD, and ZO methods across language and vision-language tasks. Their key message is that while FMAD and ZO can reduce activation memory, they do so at substantial cost to convergence rate, accuracy, and total compute. BP with activation checkpointing achieves similar memory usage but consistently outperforms these alternatives in accuracy, convergence speed , and computational efficiency.

**Strengths:**

$\textbf{Comprehensive empirical coverage.}$

The experiments are unusually broad, spanning multiple datasets (AGNews, GSM8K, MMLU, VQAv2, etc.), architectures (BERT, OPT, LLAMA, QWEN), and numerous variants of FMAD and ZO (such as variance reduction: gradient accumulation and adaptive perturbation sampling).
The empirical results are well-documented, and the authors provide open-source code. This offers useful reference points for practitioners evaluating the memory-accuracy trade-off.

---

$\textbf{Clear empirical conclusion.}$

The message is consistent: BP with checkpointing remains the dominant choice even under tight memory budgets, as FMAD and ZO cannot compete once compute and convergence stability are taken into account.

**Weaknesses:**

$\textbf{Theoretical contribution is minimal.}$

The paper repackages well-known results: FMAD and ZO introduce variance proportional to dimensionality and thus degrade convergence as dimension-dependent. The provided asymptotic bounds in Table 2 mirror classical results in standard literature. There is little new theory beyond presentation.

---

$\textbf{Empirical design conflates purpose with performance.}$

Demonstrating that FMAD and ZO are inferior to checkpointed BP under standard differentiable training does not advance understanding: this outcome is expected from basic principles.
In particular, the argument that these methods are “memory-efficient alternatives” to BP ignores the fact that checkpointing already occupies this design space. The resulting conclusion (“BP with checkpointing is better”) is tautological.

---

$\textbf{Overemphasis on scale without insight.}$

The inclusion of billion-parameter models makes the study impressive in scale but not necessarily more insightful. The experimental results largely reiterate the same qualitative pattern seen on smaller models.

---

$\textbf{Lack of forward-looking or constructive perspective.}$

The paper ends with a negative verdict on FMAD/ZO, but does not explore when these methods should be used (e.g., non-differentiable, privacy-restricted, or hardware-limited cases). As written, the paper reads more as a rebuttal to prior enthusiasm than a constructive contribution.

---

$\textbf{Questionable motivation for the comparison.}$

The central premise, directly comparing BP (a first-order method) with ZO (a derivative-free scheme) and FMAD (a forward-mode derivative estimator), is conceptually weak. These methods serve distinct purposes and are not competing paradigms for the same optimization regime.

**Questions:**

* What practical or conceptual scenario do the authors envision where FMAD or ZO could plausibly compete with BP-checkpointing? Without a defined use case, the study risks becoming a “straw-man” comparison.

---

> ### Author Response · Authors · 2025-11-18
>
> **W1 and W2: On the Novelty of Theoretical Results and The Message of this Work.**
>
> We appreciate the reviewer’s feedback and the opportunity to clarify the intent of our paper.
>
> First, our goal is not to re-establish that FmAD and ZO introduce variance proportional to dimensionality, nor to provide new asymptotic bounds. Rather, our primary motivation is to address a growing imbalance in how recent works on backpropagation-free LLM training/finetuning present their methods relative to Backpropagation (BP).
>
> Message of this Work
> Recent studies have increasingly promoted ZO and FmAD methods as practical alternatives to BP for large-scale model fine-tuning, primarily emphasizing their memory benefits, while overlooking critical trade-offs. This trend is evident in the literature, where papers consistently:
>
> 1. Compare ZO/FmAD to standard BP to highlight their memory efficiency, while omitting BP with checkpointing (a well-established, memory-efficient variant) as a fairer and stronger baseline.
>
> 2. Fail to account for computational costs [1, 3, 4] or wall-clock training time to convergence [2, 3, 5], both of which directly affect practical usability.
>
> Our work is motivated by this one-sided narrative. We aim to provide a comprehensive evaluation of these methods that jointly considers memory consumption, computational overhead, and wall-clock training time.
>
> ____
>
> _Why BP with Checkpointing Must Be Included_
>
> The reviewer notes that showing FmAD and ZO to be inferior to checkpointed BP “does not advance understanding” because this result might be expected from first principles, and that concluding “BP with checkpointing is better” seems tautological.
>
> We respectfully disagree. If this outcome were truly considered obvious or settled, we would expect prior ZO/FmAD studies (many of which explicitly include BP as a baseline) to also include BP with checkpointing. Yet none of the major works in this area ([1–6,8]) do so. Instead, they frequently report results such as “ZO can achieve comparable accuracy to BP under ___ conditions,” but those claims do not hold once checkpointing is introduced into the comparison and once we profile other important metrics like convergence time and compute.
>
> **Our contribution is the empirical demonstration that the field’s current benchmarking practice has led to misleading conclusions.** Showing that “expected” results have been systematically untested and misrepresented is precisely what advances understanding and corrects the record for the LLM optimization community.
>
> Moreover, while checkpointing and ZO/FmAD all aim for memory efficiency, they do so through fundamentally different trade-offs: checkpointing exchanges memory for recomputation, while ZO/FmAD exchange memory for noisy gradient estimates and higher computational cost. Quantifying and contrasting these trade-offs under realistic training settings is non-trivial and has not been done before.
>
> ____
>
> _On Theoretical Contributions_
>
> We respectfully note that our goal is not to propose new asymptotic theory, but to reinterpret existing results within the context of large-scale LLM optimization, a domain where theoretical bounds often fail to predict real-world performance. Our theoretical section is provided primarily for interpreting empirical results, not as a claimed novelty.
>
> We would also gently ask the reviewer to clarify what form of new theory they would expect in this setting. Given the inherent difficulty of modeling the learning dynamics of deep networks (particularly LLMs), our contribution lies in connecting known asymptotic principles to large-scale empirical behavior (where recently, FmAD and ZO are proposed as alternatives to BP), and in revealing that several widely cited works have overlooked a crucial baseline (BP with checkpointing) when making claims of memory and compute superiority.
>
> ____
>
> References:
>
> [1] Gautam et al., Variance-reduced Zeroth-Order Methods for Fine-Tuning Language Models, ICLR 2024
>
> [2] Feng et al., BAFFLE: A Baseline of Backpropagation-Free Federated Learning, ECCV 2024
>
> [3] Zhang et al., Revisiting Zeroth-Order Optimization for Memory-Efficient LLM Fine-Tuning: A Benchmark, ICML 2024
>
> [4] Guo et al., Zeroth-Order Fine-Tuning of LLMs with Extreme Sparsity, WANT@ICML 2024
>
> [5] Malladi et al., MeZO: Fine-Tuning Language Models with Just Forward Passes, NeurIPS 2023
>
> [6] Xu et al., FwdLLM: Efficient Federated Finetuning of Large Language Models with Perturbed Inferences, USENIX ATC 2024
>
> [7] Cobb et al., Second-Order Forward-Mode Automatic Differentiation for Optimization, arXiv 2024
>
> [8] Panchal et al., Thinking Forward: Memory-Efficient Federated Finetuning of Language Models, NeurIPS 2024

---

> > ### Author Response · Authors · 2025-11-18
> >
> > **W3: On the Relevance of Large-Scale Experiments.**
> >
> > In our case, the scale of the models is central to the motivation and validity of the study, not an embellishment.
> >
> > Our paper directly responds to a growing body of recent work that promotes ZO and FmAD as viable, memory-efficient alternatives to Backpropagation (BP) for training or fine-tuning large language models (LLMs). These claims are made precisely in the context of billion-parameter, differentiable architectures, where BP is often cited as being prohibitively memory-intensive. Therefore, to rigorously examine such claims, our experiments necessarily operate at the same scale.
> >
> > If the objective were to evaluate small or moderately sized differentiable models, BP alone would suffice: ZO/FmAD would offer little practical benefit. The very reason ZO/FmAD methods are being advanced in the literature is their supposed advantage at LLM scale, where memory becomes the dominant bottleneck. Evaluating them only on small models would risk missing the real-world regime where these methods are claimed to be useful.
> >
> > Our results specifically test whether BP with checkpointing provides advantages in the very regime where ZO and FmAD are proposed to be applied. The finding, that the supposed advantages of ZO/FmAD over BP vanishes once checkpointing is introduced, directly challenges the claims made in existing large-model literature and clarifies the true trade-off landscape for practical LLM optimization.
> >
> > ____
> >
> > **W4: On Constructiveness and the Scope of Our Conclusions.**
> >
> > We would like to emphasize that our intent is not to dismiss FmAD or ZO methods outright, but to clarify the boundaries of their applicability in light of how recent literature has positioned them.
> >
> > In our manuscript, we have agreed and discussed that ZO and FmAD can be valuable in specific contexts such as non-differentiable objectives, privacy-restricted environments, or hardware-limited settings. We explicitly acknowledge this in the first paragraph of the Introduction and at the end of Section 3 (“A Note on Non-differentiable and Black-box Settings”), where we discuss these cases and cite corresponding works. Our analysis highlights that their applicability to large-scale differentiable models (such as LLMs) is fundamentally limited. In many inference settings, perturbing billions of parameters or querying an LLM’s loss functions repeatedly is infeasible, both computationally and practically.
> >
> > The key message of our paper is therefore not that “FmAD/ZO are bad,” but that recent enthusiasm for applying these methods to differentiable, large-scale models has overlooked critical trade-offs and omitted key baselines (e.g., BP with checkpointing). Our findings show that once those baselines are included, the apparent advantages of FmAD/ZO outside of non-differentiable and privacy-restricted settings disappear.
> >
> > Finally, we respectfully note that course-corrective analyses are constructive contributions. The field benefits not only from proposing new algorithms, but also from reevaluating prevailing assumptions and experimental practices, especially when those practices have influenced a growing research direction. In this sense, our work provides a necessary empirical and conceptual clarification that can guide future research on when and why FmAD/ZO should be used, rather than advocating against them entirely.

---

> > > ### Author Response · Authors · 2025-11-18
> > >
> > > **W5 and Q1: On the Conceptual Basis for Comparing BP, FmAD, and ZO**
> > >
> > > Our comparison of BP, FmAD, and ZO is to evaluate their practical utility in the specific context where recent literature has begun treating them as competing paradigms, namely, large-scale LLM training and fine-tuning.
> > >
> > > Multiple recent works (e.g., MeZO [5], BAFFLE [2], FwdLLM [6], and others) explicitly present ZO and FmAD as memory-efficient alternatives to BP, and directly compare their methods against standard BP baselines. In light of this ongoing trend, it becomes necessary to ask:
> > >
> > > 1. If these methods are being positioned as replacements for BP in differentiable training, how do they actually compare in terms of convergence speed, computational cost, and wall-clock time?
> > >
> > > 2. If, as the reviewer notes, these methods “serve distinct purposes,” then should they be used for LLM training at all?
> > >
> > > We agree with the reviewer’s earlier remark that “FmAD and ZO introduce variance proportional to dimensionality and thus degrade convergence as dimension-dependent.” This observation directly supports our argument that these methods are ill-suited for high-dimensional differentiable optimization such as LLM fine-tuning, precisely the regime in which they are currently being promoted.
> > >
> > > Our comparison therefore serves a clarifying function: it does not claim that BP, FmAD, and ZO are theoretically equivalent optimization frameworks, but rather demonstrates that when applied to the same problem setting (LLM training), their empirical trade-offs reveal that BP with checkpointing remains the more practical and efficient choice.
> > >
> > > In short, our study is not conceptually misguided. It is conceptually necessary, given how the current literature has framed ZO and FmAD in direct competition with BP. It does not construct a “straw-man” comparison, it tests the exact claims made by proponents of these methods in recent literature, within the application regime (LLM training and fine-tuning) where these claims have gained visibility.
> > >
> > > To clarify:
> > >
> > > 1. We do not argue that FmAD or ZO have no place at all; they remain valuable for **non-differentiable, black-box, or privacy-restricted** scenarios.
> > >
> > > 2. However, these are not the contexts where recent large-scale studies have applied them. Our contribution, therefore, is to **empirically delineate where these methods fail to compete with BP**, ensuring that future applications are guided by a realistic understanding of their trade-offs rather than overstated generality.

---

> > ### Comment · Reviewer_KQZt · 2025-11-23
> >
> > I believe the authors have misunderstood how recent work on backpropagation-free LLM training and fine-tuning should be positioned relative to backpropagation (BP).
> >
> > Several of these studies demonstrate that $\textbf{zeroth-order methods can achieve competitive performance compared with first-order methods}$ (including full or partial fine-tuning). However, the purpose of these demonstrations is not to claim superiority or efficiency under equivalent conditions. Instead, they are meant to show that $\textbf{under strict memory constraints or in settings without access to a gradient oracle, zeroth-order optimization can serve as a viable alternative.}$
> >
> > Therefore, a direct comparison of computational cost between zeroth-order and first-order methods is largely $\textbf{invalid outside of those constrained settings.}$ Without explicitly assuming hard constraints (e.g., severe memory limitation or lack of gradient access), there is no meaningful or fair “trade-off” to be discussed between zeroth-order and first-order approaches.
> >
> > In this sense, the $\textbf{core motivation of the present study is fundamentally flawed.}$
> >
> > ---
> >
> > Comparing the performance and efficiency of zeroth-order methods, BP with checkpointing, and standard first-order methods is not a valid experimental setup for drawing general conclusions. Prior work on zeroth-order methods (e.g., [5]) was never intended to show that approaches like MeZO can outperform or even match standard fine-tuning methods under the same computational budget. Rather, these works ask a different question: $\textbf{If gradients are unavailable, what level of performance can we reasonably expect when using a zeroth-order method?}$
> >
> > From this perspective, the claimed contributions in the paper are $\textbf{highly questionable.}$

---

> > > ### Author Response · Authors · 2025-11-25
> > >
> > > We respectfully but firmly disagree with the reviewer's characterization of our work. The reviewer suggests that "a direct comparison of computational cost between zeroth-order and first-order methods is largely meaningless" and that our experimental setup is invalid. However, this critique fundamentally misunderstands both the current state of the literature and our paper's contribution.
> > > ___
> > > **The Literature Explicitly Positions ZO Methods as General Alternatives**
> > >
> > > Recent ZO literature consistently positions these methods as practical alternatives for standard differentiable LLM fine-tuning, not merely for gradient-unavailable scenarios. To name a few:
> > >
> > > 1. **MeZO (NeurIPS 2023)** claims to achieve "comparable performance to fine-tuning with backpropagation across multiple tasks" and states that "MeZO can train a 30-billion parameter model, whereas fine-tuning with backpropagation can train only a 2.7B LM with the same budget"
> > >
> > > 2. **MeZO-SVRG (ICLR 2024)** claims "memory savings of up to 76% w.r.t FO-SGD" and that it "often achieves comparable performance to FO-SGD (within 5%) on several tasks"
> > >
> > > 3. **Zero-Benchmark (ICML 2024)** explicitly proposes "a shift towards BP-free, zeroth-order (ZO) optimization as a solution for reducing memory costs during LLM fine-tuning" and states that "some BP-free methods exhibit effectiveness comparable to, or even superior to, that of FO methods"
> > >
> > > 4. **SensZOQ (ICLR 2025)** states that "ZO methods have demonstrated the potential to achieve performance comparable to first-order methods in LLM fine-tuning, which creates new venues for efficient LLM adaptation strategies"
> > >
> > > These are not claims about gradient-unavailable settings, they are direct comparisons for standard differentiable training scenarios where gradients are fully accessible. The papers evaluate on standard differentiable models (Llama, OPT, Mistral, RoBERTa) with full gradient access and compare against "backpropagation," "first-order methods," or "FO-SGD/FO-Adam."
> > > ___
> > > **The Critical Missing Baseline**
> > >
> > > The reviewer suggests our comparison is invalid, yet fails to address the central issue: **none of these papers include BP with checkpointing as a baseline**. They compare against generic "backpropagation," "first-order methods," or "FO-SGD/FO-Adam" without acknowledging memory-efficient BP variants. This systematic omission has led to misleading conclusions about memory-efficiency trade-offs:
> > >
> > > 1. MeZO claims 12x memory reduction compared to "backpropagation"
> > >
> > > 2. MeZO-SVRG claims 76% memory savings compared to "FO-SGD"
> > >
> > > 3. Zero-Benchmark shows ZO-SGD requires 64 GB vs. FO-SGD's 148 GB
> > >
> > > 4. SensZOQ shows 8 GB vs. 48 GB for "FO-Adam Full FT"
> > >
> > > **None of these comparisons include BP with checkpointing**, a very basic approach addressing memory constraints. This is not to mention advanced offloading optimizations that can further trade CPU-GPU communication for more memory savings.
> > > ___
> > > **Our Contribution is Correcting the Experimental Record**
> > >
> > > Our work does not create an "invalid comparison", it tests the exact claims made in existing literature within the contexts where those claims are being made. The reviewer states that comparing ZO to first-order methods is invalid, yet this is precisely what all the cited works do.
> > >
> > > When we include the proper baseline (BP with checkpointing), we find that:
> > >
> > > 1. The claimed 12x memory advantage (MeZO) disappears
> > >
> > > 2. The claimed 76% memory savings (MeZO-SVRG) disappear
> > >
> > > 3. BP with checkpointing achieves superior accuracy (up to 31.1% higher), convergence speed (34.8% faster), and computational efficiency (3.8× fewer computations)
> > > ___
> > > **We Acknowledge the Legitimate Use Cases**
> > >
> > > We explicitly state in our paper that ZO methods retain value in non-differentiable or black-box settings. Our work clarifies that for standard differentiable LLM training (even under memory constraints) BP with checkpointing should be the baseline to compare with.
> > >
> > > The value of our contribution is not in showing "ZO is always bad," but in **preventing future researchers from pursuing ZO/FmAD for standard LLM training based on incomplete comparisons**. Our work corrects the experimental record by including the missing baseline that these influential papers systematically omitted.

---

### Official Review · Reviewer_nYhR · 2025-10-27

**Soundness:** 2
**Presentation:** 3
**Contribution:** 2
**Rating:** 4
**Confidence:** 3

**Summary:**

This paper studies alternatives to backpropagation (BP), specifically forward-mode automatic differentiation (FM-AD) and zero-order (ZO) optimization, for training neural networks. The authors advocate using BP with checkpointing as the primary method and argue that the memory savings offered by FM-AD and ZO do not justify their performance loss and additional computational cost. They consider different variants of these algorithms and provide a theoretical analysis of their convergence rates, memory usage, and time complexities. In addition, they present extensive experiments on various models and datasets to compare these methods against each other.

**Strengths:**

- The problem under investigation is relevant and practically significant.
- Theoretical analysis compares different important aspects of these algorithms.
- Experiments are extensive, covering different models and datasets.
- The paper is generally well written and easy to read.

**Weaknesses:**

- The paper is a study of existing methods and benchmarks them on different models and datasets. The theoretical results for FM-AD are not new and are already well known in the community; however, I am not familiar with the novelty of the results for ZO. The overall message of the paper is not surprising and aligns with most previous works comparing these methods to BP. While the authors provide extensive experiments to compare these approaches, the text largely reiterates what is already visible in the figures, rather than offering deeper insights into why certain behaviors occur or providing more intuitive explanations of the results. For a paper of this nature, readers would expect more conceptual understanding and insightful remarks, rather than a lengthy restatement of numerical findings.


- The directions used in FM-AD and ZO are drawn from a normal distribution, and no alternative distributions or selection strategies are considered. I believe the performance of these algorithms may strongly depend on the choice of directions, making this an important factor for comparison. For example, in FM-AD, when n=D, one can set the directions as basis vectors and recover the gradient exactly, without error. This point is neither mentioned nor included in the comparisons. By treating normally distributed directions as an inherent part of the algorithms, the paper risks presenting a potentially misleading picture of their performance.

**Questions:**

- Why is the compute in Table 2 for the parallel versions of the methods the same as for the sequential versions?

- Do you normalize the directions drawn from the normal distribution? Without normalization, the gradient magnitudes are altered, which requires extra care in adjusting and scheduling learning rates for FM-AD and ZO.

- In Figure 2, some graphs converge initially but begin to diverge after some time. Why does this happen?

- In Figure 3, why is gradients + optimizer state + misc higher for FM-AD? Also, the activations memory for this method should be roughly 2× that of ZO (one vector for forward pass activations and another for JVP), so why does it appear significantly larger in the figures?

---

> ### Author Response · Authors · 2025-11-18
>
> We are grateful for your assessment of the problem’s significance, the strength of our empirical analysis, and the clarity of the paper’s presentation.
> ____
>
> **W1: On the Novelty of Theoretical Results and The Message of this Work.**
>
> We kindly refer the reviewer to our response to **Reviewer KQZt** (_W1 and W2: On the Novelty of Theoretical Results and The Message of this Work_) for a detailed discussion on this point.
>
> ____
>
> **W1: Explanations on the Results.**
>
> We have not simply restated the numerical findings; we provide explanatory discussion on each of these key axes. For example, we connect memory cost differences to activation versus perturbation storage, compute/time differences to gradient variance and forward-mode operations scaling with dimension, accuracy trade-offs to high-dimensional noise effects, and we include a dedicated failure-mode analysis to unpack why these methods degrade at scale. These insights are summarized in the table below:
>
> | Topic | Section / Appendix | What we explain (rationale) |
> |---|--------------------|-----------------------------|
> | **Accuracy / Trade‑offs** | Section 4.2 (Comparison on Accuracy) + Appendix F (Variance‑reduced methods) | Explains that FmAD and ZO lag behind BP due to noise in their perturbation-based gradient estimates. FmAD consistently outperforms ZO because it has access to analytic first-order Jacobian-vector products, whereas ZO relies on noisier finite-difference approximations. Variance-reduction strategies (either increasing perturbations or accumulating gradients) can improve accuracy by reducing noise, but these gains come at the cost of longer runtime (if done sequentially and if it requires additional historical gradient-related information) or higher memory usage (if done parallely). |
> | **Wallclock Convergence Time** | Section 4.3 (Comparison on Wallclock Convergence Time) / Figure 2 | Discusses that BP with checkpointing converges faster because it efficiently reuses downstream gradients with minimal per-layer computation. FmAD and ZO experience slower convergence due to gradient approximation errors and the additional computations required per layer (e.g., Jacobian-vector products for FmAD). Using multiple perturbations improves gradient quality and stabilizes training, but increases overall runtime for sequential implementations. Accumulation strategies amortize per-step computations, maintaining per-iteration efficiency, but delay parameter updates, which also slows overall convergence. |
> | **Memory Cost** | Section 4.4 (Comparison On Memory Consumption) / Figure 3 | Discusses that FmAD and ZO reduce memory compared to BP-Vanilla by storing only the previous layer’s activations, avoiding OOM errors. However, this comes with longer training times and lower accuracy. FmAD uses more activation memory than ZO due to additional storage for Jacobian-vector products. Variance-reduction variants (-Multiple) improve gradient quality but increase memory linearly since all the activations of parallel evaluations would be stored together, while -Accumulate amortizes computations over time without extra memory overhead since its sequential making the inference calls, at the cost of slower convergence. |
> | **Compute Cost** | Section 4.5 (Comparison on Compute Cost) / Table 4 | Explains that although FmAD and ZO can reduce compute per iteration, their high gradient variance and slow convergence cause the total compute required to reach comparable accuracy to be much higher than BP with checkpointing. Using multiple perturbations improves gradient quality and accuracy but linearly increases computation, while accumulating gradients across iterations maintains per-iteration efficiency but still suffers from slow convergence. Overall, the total computational burden is dominated by the combination of variance and convergence inefficiency. |
> | **Failure Mode Analysis** | Section 4.6 and Appendix F.5 (“Failure mode analysis of FmAD & ZO”) | Identifies causes of poor convergence: FmAD and ZO can fail due to interactions between gradient variance, optimizer choice, and layer-wise computations. In FmAD, adaptive optimizers (e.g., AdamW) can trigger cascading amplification of Jacobian-vector products, producing large, unstable gradient updates that may cause divergence or noisy weight updates. High effective gradient variance under adaptive optimizers leads to reduced final accuracy and occasional NaN gradients. In contrast, non-adaptive optimizers (SGD) maintain bounded gradients and more stable convergence. These observations highlight that optimizer selection critically affects the stability of FmAD and ZO, and variance-reduction methods alone cannot fully mitigate these failure modes. |
>
> (continued in the next comment)

---

> > ### Author Response · Authors · 2025-11-18
> >
> > **Cont. W1: Explanations on the Results.**
> >
> > Taken together, these explanations demonstrate that our paper provides deeper insights into the mechanisms driving memory, compute, convergence, and accuracy behaviors for FmAD and ZO, as well as their failure modes. While summarizing the results from tables and figures is important for clarity, we have consistently backed these summaries with reasoning and conceptual explanations for why these behaviors occur. If the reviewer believes any specific aspect remains unexplained, we would be grateful for a pointer so we can clarify or expand on it.
> > ____
> >
> > **W2 and Q2: Alternative Distributions for Sampling Perturbations and Normalizing the Sampled Perturbations.**
> >
> > We appreciate the suggestion. We have now added a dedicated subsection titled “Effect of Perturbation Distributions and Normalization Strategies” and colored blue in Appendix F.6, where we analyze Normal vs. Uniform sampling and normalized vs. unnormalized perturbations, along with detailed rationale for the observed performance differences.
> >
> > To summarize, across sampling strategies, we find that **normalization consistently hurts performance**, as constraining perturbations to a fixed radius removes informative scale variation and increases the effective noise in the jvp estimate. Among unnormalized variants, **normal sampling performs best**, with uniform sampling close behind, driven by the normal distribution’s wider range of perturbation magnitudes that better captures curvature information in high-dimensional models.
> >
> > Further, the setting of $n=d$ (#of perturbation=#of trainable parameters) is infeasible for LLMs, where $d\approx 10^6 ⁣- 10^{11}$ parameters. Computing even a single full set of basis directions would require billions of forward-mode evaluations per step, which is orders of magnitude beyond practical compute budgets. For this reason, FmAD/ZO methods (including those in prior work) are studied in the regime where $n \ll d$, and we evaluate multiple direction-sampling strategies within this realistic setting.
> >
> > ____
> >
> > **Q1: Equal Compute for Parallel and Sequential Forward-only Methods.**
> >
> > The total computational cost for parallel versions of ZO and FmAD remains the same asymptotically as for the sequential versions because each perturbation still requires a forward (or jvp) evaluation of the network. Parallelization affects only the wall-clock runtime by executing multiple evaluations simultaneously, but does not reduce the total number of operations performed. As detailed in Appendix H, the $\mathcal{O}(nd)$ complexity accounts for all $n$ perturbations for total of $d$ trainable parameters, regardless of whether they are executed sequentially or in parallel.
> >
> > ____
> >
> > **Q3: Divergence in Figure 2.**
> >
> > This behavior is discussed in Section 4.6 and Appendix F.5. We summarize the explanation here:
> > The divergence observed in Figure 2 for ZO and FmAD after initial convergence could arises from high gradient variance interacting with the optimizer. In high-dimensional networks, finite-difference (ZO) or Jacobian-vector product (FmAD) estimates accumulate noise over iterations. Adaptive optimizers like AdamW amplify this noise by adjusting learning rates based on gradient history, which produces unstable updates and eventual divergence. In contrast, SGD does not adapt to gradient magnitude, so while it avoids runaway spikes, its fixed learning rate doesn’t overcome the inherent variance in ZO/FmAD gradients. We observed stagnation and very slow or negligible learning with SGD, hence we had reported the learning curves with AdamW.

---

> > > ### Author Response · Authors · 2025-11-18
> > >
> > > **Q4: Memory Consumption in Figure 3.**
> > >
> > > The higher “Gradients + Optimizer state + Misc” memory for FmAD is primarily due to the “Misc” category, which PyTorch’s memory profiler does not fully decompose. Our best explanation is that it includes memory for intermediate and temporary tensors created during the JVP computations (Equation 1 in the paper), such as products and accumulations required for forward-mode AD. These tensors are short-lived but would increase the memory consumption, inflating the measured “Misc” category.
> > >
> > > In forward-mode AD, the Jacobian-vector products (jvps) are propagated forward through all layers (Equation 1, Section 2), meaning that each layer’s perturbation $\delta y_i$ depends on both the current weight perturbation $v_i$ and the propagated $\delta y_{i-1}$. This creates a cumulative storage requirement across layers for all intermediate $\delta y_i$ values, unlike ZO where perturbations are applied externally and only one forward pass of activations is needed.
> > >
> > > This explains why, in practice, we observe activation memory of 5.41 GB for FmAD versus 1.26 GB for ZO on Llama 3.1 8B (and similarly 5.49 GB vs 1.67 GB for OPT-13B). The amplification beyond the 2$\times$ estimate comes from the combination of layerwise propagation of perturbations, and framework-level buffers maintained by PyTorch during jvp computations. Thus, the empirical numbers are consistent with the forward-mode AD mechanism and illustrate why FmAD incurs substantially higher activation memory than ZO.

---

> > > > ### Comment · Reviewer_nYhR · 2025-11-23
> > > >
> > > > Thank you for the clarifications. I’ll keep the score and suggest the authors consider smarter methods for guessing directions, such as [1], even if they might increase memory usage as claimed; this paper is in a good position to provide a more complete picture by comparing such strategies.
> > > >
> > > > [1] Can forward gradient match backpropagation?, Fournier et al., ICML 2023.

---

> ### Author Response · Authors · 2025-11-25
>
> Thank you for the constructive suggestion to explore smarter direction-selection strategies such as those proposed by Fournier et al. (ICML 2023).
>
> We already cite and discuss the paper by Fournier et al., which proposes to improve FmAD by generating more structured perturbations using auxiliary networks rather than purely random directions. However, their experiments remain on moderate-scale CNNs (ResNet-18 on CIFAR/ImageNet32), and there is **no discussion on memory usage, compute cost, or wall-clock convergence time**. Importantly, while they explore direction-guessing improvements, they do not demonstrate scalability to billion-parameter LLM fine-tuning, nor do they show that the auxiliary networks avoid extra overhead in compute or memory at large scale. In our setting of large-scale differentiable fine-tuning, these trade-offs remain unresolved.
>
> In Appendix F.6, we already analyze multiple perturbation strategies (Normal vs. Uniform, normalized vs. unnormalized). Our findings show that while sampling strategy affects performance, the fundamental limitations (high variance, slow convergence, computational overhead) persist across all variants. Our work evaluates how FmAD/ZO methods are **currently positioned and applied** in recent LLM literature (MeZO, MeZO-SVRG, Zero-Benchmark, SensZOQ), which predominantly use random Gaussian perturbations. Smarter direction-selection methods would require additional memory and/or convergence time (partially negating the claimed advantage) and have unresolved scalability at LLM scale.
>
> Given these clarifications, we respectfully ask the reviewer to consider whether the paper's scope and treatment of this line of work have been sufficiently addressed. If so, we would appreciate reconsideration of the score.
>
> At the very least, we would appreciate a clear justification for maintaining the current score, as this helps us understand the reviewer's assessment and revise the paper accordingly. **Blindly suggesting changes or additions without considering their practical utilities or trade-offs can be counter-productive and may mislead practitioners, ultimately hindering the community's progress.**

---

> > ### Comment · Reviewer_nYhR · 2025-11-25
> >
> > I have already justified my score, but I'll summarize it here for you.
> >
> > (1) For a paper of this type, readers expect more insightful analysis and summarization of the results. Most of the current interpretations simply report or compare numbers. Since these numbers are architecture- and dataset-specific, I did not find them sufficiently engaging or broadly useful for the community. I believe the authors could parse and interpret their results much more effectively, and this is itself an important part of the contribution.
> >
> > (2) The paper is not treating FmAD and ZO fairly (see W2), and I somewhat agree with reviewer KQZt in that regard. For FmAD in particular, there are better direction-selection approaches that lead to higher accuracy, and readers expect these to be evaluated for a complete picture. Simply claiming that these strategies incur more memory or longer convergence time without any experiments to show it is not acceptable. As for “practical utilities,” these methods can be tested within their respective domains, at least.
> >
> > Finally, regarding the remark about reviewers “blindly suggesting changes...”: this is inappropriate and false. First, the suggestion was not blindly taken; your paper is downplaying alternative methods while omitting their advanced versions. This makes your conclusion incomplete and therefore wrong. Second, it is your paper’s responsibility (as advertised) to analyze their memory and compute costs and find a suitable way to apply them.

---

> > > ### Author Response · Authors · 2025-11-25
> > >
> > > **On (1) Depth of Analysis**
> > >
> > > We would appreciate more concrete guidance on what additional forms of “parsing and interpretation” the reviewer believes are missing beyond what we have already provided in **W1: Explanations on the Results** in our rebuttal. As with prior work (including the ICML 2023 paper by Fournier et al. that the reviewer has cited) empirical analyses in this domain necessarily rely on model- and dataset-specific quantitative experiments. We follow this established practice while providing clear takeaways about memory, compute, and accuracy trade-offs.
> > >
> > > Our analysis provides explanations across all axes: accuracy differences through gradient variance, convergence via optimizer interactions and JVP dynamics, memory through activation storage patterns, and compute via variance and convergence speed. Our **failure mode analysis** (Section 4.6, Appendix F.5) identifies root causes like cascading JVP amplification and training instabilities. The consistency across **diverse architectures** (BERT 110M-340M, RoBERTa 125M-355M, OPT 1.3B-13B, Llama 8B, Qwen 7B) and **task types** demonstrates broadly applicable patterns, not narrow artifacts.
> > > ___
> > > **On (2) Fairness to FmAD**
> > >
> > > We provide a systematic analysis that **includes –accumulate and –multiple variants for both FmAD and ZO, as well as –SVRG and –Sparse variants for FmAD, all of which we introduced to examine the efficiency–accuracy trade-offs of alternative approaches**. **As stated in our rebuttal, we conducted additional experiments (Appendix F.6) verifying that alternative direction distributions do not improve performance – fundamental limitations persist across all variants**. As with any empirical comparison, including every possible variant of every method is impractical, so we focus on representative baselines that illuminate the specific dimensions our paper aims to study. We believe this approach provides an appropriately balanced and informative comparison within the scope advertised in the paper.
> > > ___
> > > Lastly, recent ZO/FmAD literature systematically omits BP with checkpointing while claiming memory efficiency. Our work tries to correct an incomplete experimental record, not “downplaying alternatives”. We fully agree that assessing the memory and compute implications of alternative methods is our responsibility, and the paper aims to do so carefully within the constraints of space and scope.

---

> > > > ### Author Response · Authors · 2025-12-03
> > > >
> > > > Additionally, we would like to mention that our work already includes an adaptive perturbation-selection variant (–Adaptive) that samples directions aligned with the rolling average of historical gradients. As shown in Appendix F.5.2 and Section 4.2, this adaptive strategy produces only limited or inconsistent accuracy improvements. These results confirm that the adaptive strategy, one of the smarter direction-selection methods, do not change the fundamental trade-offs: BP with checkpointing remains superior in accuracy, convergence speed, and computational efficiency. We apologize for not highlighting this earlier in our response.

---

### Official Review · Reviewer_MFid · 2025-10-31

**Soundness:** 3
**Presentation:** 3
**Contribution:** 2
**Rating:** 6
**Confidence:** 4

**Summary:**

This study investigates some alternatives to backpropagation aimed at training large model with minimal memory usage. These methods are compared against standard backpropagation, as well as checkpointed backpropagation.
After both theoretical analysis and empirical measurements on several models and fine-tuning tasks, the authors show that forward only methods suffer from poor convergence properties, due to noisy gradient estimation. Checkpointed backpropagation, on the other hand, computes exact gradient with similar memory requirements, and hence appears as a more viable method for fine-tuning large models under strict memory constraints.

**Strengths:**

- The claim that checkpointed BP can be much better than forward only alternatives while achieving the same memory requirements, is well motivated. It may also encourage the community to include it as a strong baseline when performing experiments, since it appears to be missing in many works.
- A theoretical analysis of each method is performed, followed by empirical validation.
- The study covers multiple aspects of the methods including accuracy, memory, time, FLOPs, and convergence speed.
- The evaluation setup is diverse, with models between 110M and 13B parameters, for language and vision-language tasks.

**Weaknesses:**

- The HiZOO training algorithm [1] may deserve some discussion in the paper. This forward-only method, tries to address the slow convergence of MeZO by leveraging second-order information, which is what is criticized in the current study.
- The theoretical analysis provides time and memory complexities for each method. However the big O notations hide constant factors, which are important to consider since for instance a backward pass is about 2x slower than a forward pass. Since the methods rely on VJP, JVP and/or forward passes, they have different constant factors.
- The complexities are presented without accounting for parallelism. Forward-only methods remove the backward locking, which unlocks new ways of parallelizing the computations across devices. Similarly, ZO methods involve multiple forward passes which could be run in parallel, thus reducing the effective training time.

Minor:
- Some notations are a bit confusing. For instance in Table 2, why use $D$ for the number of layer when it was defined as $p$ before (line 133).

[1] Second-Order Fine-Tuning without Pain for LLMs:A Hessian Informed Zeroth-Order Optimizer, Zhao et al., ICLR 2025

**Questions:**

- Why does the (vanilla) FMAD method have a similar FLOPs/iter to BP-CHECKPOINTING in Table 4? I would expect that the methods would be ordered like: FMAD < BP < BP-CHECKPOINTING, since FMAD involves a JVP during the forward pass that is usually noticeably faster than doing a backpropagation, and checkpointed BP adds a computational overhead over BP.
- In Table 2, I do not get why the computational complexity of BP with checkpointing is in $\mathcal{O}(d \log D)$.

---

> ### Author Response · Authors · 2025-11-18
>
> We are grateful for the reviewer's thoughtful feedback and for highlighting the motivation behind comparing to checkpointed BP, along with their acknowledgement of our theoretical grounding, broad evaluation setup, and multi-faceted analysis of these methods.
>
> ____
>
> **W1: Comparison with a Second-order Forward-only Method.**
>
> We thank the reviewer for pointing out the HiZOO work [1]; we have included a discussion of it in the revised version of Related Work (Appendix A), with the addition highlighted in blue. HiZOO provides a forward-only training strategy that leverages second-order information to improve the convergence rate of zeroth-order (ZO) methods such as MeZO. This direction is closely aligned with our motivation of exploring alternatives to standard backpropagation for memory-constrained training.
>
> That said, we note that HiZOO is compared against backpropagation in terms of activation memory usage, rather than wall-clock convergence time nor compute cost relative to backpropagation. HiZOO also ignores the stronger BP-checkpointing baseline. This concerning literature trend is exactly what motivates our work.
>
> [1] Second-Order Fine-Tuning without Pain for LLMs:A Hessian Informed Zeroth-Order Optimizer, Zhao et al., ICLR 2025
>
> ____
>
> **W2: Big-O Comparison Hiding Constant Factor.**
>
> We agree that asymptotic complexity alone does not capture the full picture, as constant factors. Particularly, the relative cost between forward, backward, and Jacobian–vector operations, can meaningfully affect runtime.
>
> This is precisely why our analysis is complemented by detailed empirical measurements of both runtime and memory (Section 4 and Appendix F), which directly reflect these constant overheads. Appendix H further provides the explicit computational expressions (including constant terms) for each method’s FLOP and memory cost. The clarification to BP-Checkpointing’s compute bound is provided in our response to “Q2: Computational Complexity of BP with Checkpointing.” Additionally, the constant terms appearing in the error bounds of our convergence analysis are detailed in Appendix I.
>
> ____
>
> **W3: Accounting the Parallelizability of Forward-only Methods.**
>
> We have explicitly evaluated both parallel and sequential variants of ZO and FmAD across all four axes: accuracy (Sec. 4.1), wallclock convergence time (Sec. 4.2), memory consumption (Sec. 4.3), and compute cost (Sec. 4.4). As shown in these analyses, accuracy improvements through running multiple forward-only evaluations in parallel come with high memory, while running them sequentially increases the wallclock convergence time. In both cases, the compute cost also increases.
>
> These results together demonstrate that while parallelization can reduce effective training time, it introduces non-trivial trade-offs in efficiency and resource use, which our empirical results quantify comprehensively.
> ____
>
> **Minor W: Notation Clarification.**
>
> We thank the reviewer for catching this oversight. We have corrected the inconsistent notation in the updated version, the number of layers is now consistently denoted as $p$ throughout, including in Table 2.
>
> ____
>
> **Q1: FLOPs/iter for FmAD vs BP-Checkpointing.**
>
> While FmAD avoids the explicit backward pass of BP, it still incurs comparable computational cost due to the way Jacobian-vector products (jvps) are propagated through all layers.
>
> In particular, as shown in Equation 1 of Section 2, each layer in FmAD requires computing
> $\delta y_i = \frac{\partial y_i}{\partial w_i} v_i + \frac{\partial y_i}{\partial y_{i-1}} \delta y_{i-1},  \text{for }i \in [2, p]$
> which involves two Jacobian multiplications per layer, one with respect to the weights and one with respect to the previous activations. This recursive dependency means that all intermediate directional derivatives $\partial y_i$ must be propagated through the network, much like gradients in the backward pass.
>
> As a result, the total FLOPs per iteration for FmAD are not lower than BP, and in practice they become close to those of BP-Checkpointing. Together, these effects make the two methods comparable in empirical FLOP count, even though conceptually FmAD is “forward-only.”

---

> > ### Author Response · Authors · 2025-11-18
> >
> > **Q2: Computational Complexity of BP with Checkpointing.**
> >
> > The $\mathcal{O}(d \log⁡{D})$ term arises from the recomputation overhead introduced by checkpointing, where $D$ is the total number of layers, and $d$ is the number of trainable parameters. While the compute cost of a standard backward pass scales as $\mathcal{O}(d)$, checkpointing trades off lower memory consumption for higher compute cost by storing activations only at selected layers and recomputing intermediate ones during the backward pass.
> >
> > In the optimal checkpointing schedule (as shown in [2]), the recomputation cost grows logarithmically with the number of layers $D$. Thus, the total compute complexity becomes
> > $\mathcal{O}(d \log{⁡D})$, where $\log{D}$ captures the extra recomputation of the activations which have been discarded during the forward pass, which is a factor beyond the base $\mathcal{O}(d)$ cost of backpropagation.
> >
> > We have clarified this reasoning in Appendix H.3 ‘Backpropagation with Checkpointing’ (Highlighted in blue) of the revised version.
> >
> > [2] Algorithm 799: Revolve: An Implementation of Checkpointing for the Reverse or Adjoint Mode of Computational Differentiation (Griewank & Walther, ACM TOMS 2000)

---

### Official Review · Reviewer_4Fn3 · 2025-10-31

**Soundness:** 3
**Presentation:** 3
**Contribution:** 3
**Rating:** 6
**Confidence:** 3

**Summary:**

This paper provides a comprehensive theoretical and empirical comparison of backpropagation (BP) with activation checkpointing against forward-mode automatic differentiation (FMAD) and zero-order (ZO) optimization methods for gradient computation.

It highlights trade-offs in accuracy, convergence speed, memory, and computation, introduces variance reduction techniques (gradient accumulation and adaptive perturbation sampling) for FMAD and ZO, and analyzes their failure modes, concluding that BP with checkpointing generally outperforms alternatives on large language and vision-language models across diverse tasks.

**Strengths:**

The paper's main contribution is highlighting gaps of prior work on FMAD and ZO methods, particularly the lack of comparisons to memory-efficient BP variants like activation checkpointing and incomplete evaluations of trade-offs beyond memory savings. Its unified theoretical framework (Table 2) effectively distills convergence bounds, memory costs, and compute complexities.

The empirical evaluation is broad, spanning large-scale models (Llama 3.1 8B, Qwen 2 VL 7B), tasks (text classification, generation, visual QA), and method variants covering discussions around accuracy, convergence time, memory, and FLOPs.

The variance reduction techniques (Gradient Accumulation, Perturbation Sampling) proposal over vanilla FMAD/ZO demonstrates a significant improvements in accuracy. The failure mode analysis (JVP instabilities, optimizer interactions) offers practical insights into why these alternatives underperform, enhancing the paper's significance for practitioners in low-resource settings.

**Weaknesses:**

While the theoretical analysis is insightful, it relies on assumptions like non-convex but L-smooth functions, which may not fully capture the complexities of modern transformers (attention mechanisms or quantization effects in QLORA); providing sensitivity analyses or relaxations could strengthen claims.

Empirically, the focus on QLORA with low ranks (r=1) limits generalizability—higher ranks or full fine-tuning scenarios might alter trade-offs, especially for FMAD/ZO's scalability in high dimensions.

Some baselines feel incomplete: for instance, comparisons to more advanced ZO variants like Zo-SignSGD or hybrid methods (e.g., combining ZO with BP) are missing, and the vision tasks are limited to one model (Qwen 2 VL), potentially overlooking multimodal-specific challenges.

**Questions:**

In the theoretical bounds (Table 2), how sensitive are the convergence errors for FMAD and ZO to the choice of perturbation budget n in practice? Could you provide ablation results on varying n for a larger model like OPT-13B?

The new variance reduction techniques improve accuracy, but how do they interact with other optimizers beyond AdamW? Did you observe similar JVP spikes or instabilities?

Why focus primarily on QLORA with r=1? Would results hold for higher ranks or adapter-free fine-tuning, where parameter count d increases dramatically?

---

> ### Author Response · Authors · 2025-11-18
>
> Thank you for the thoughtful review, we appreciate your recognition of the importance of comparing FmAD/ZO methods against memory-efficient BP techniques such as activation checkpointing.
> ____
>
> **W1: Theoretical Assumptions.**
>
> Our theoretical analysis indeed assumes $L$-smooth non-convex functions, which is a standard assumption in convergence analyses for first- and zeroth-order methods [1, 2]. The goal was to establish a clear and interpretable baseline bound that isolates how dimension ($d$) and number of perturbations ($n$) jointly influence convergence.
>
> Regarding sensitivity, we note that the bounds in theorems in Appendix I.7 explicitly quantifies this dependence: larger $d$ or smaller $n$ worsen the rate, while smaller $L$ or step size $\eta$ improve stability. Our empirical study in Appendix F.4, which varies model dimension, perturbation count, and learning rate, confirms that the observed behaviors align with the theoretical trends.
>
> As for relaxations, capturing the full dynamics of large-scale transformer optimization is analytically intractable, given the non-linearities of attention and quantization. Our empirical results already demonstrate that the trends predicted by the smooth analysis persist in quantized QLoRA settings, suggesting the theoretical dependence on $(d,n,\eta)$ remains qualitatively valid even under such non-idealities.
>
> [1] Stochastic First- and Zeroth-order Methods for Nonconvex Stochastic Programming (Ghadimi & Lan, 2013)
>
> [2] Randomized gradient-free methods in convex optimization (Nesterov & Spokoiny, 2017)
>
> ____
>
> **W2 and Q3: Choice of QLoRA with Rank 1.**
>
> We focus primarily on QLoRA with rank $r=1$ to study the regime with the lowest parameter count, where memory- and compute-efficient alternatives such as FmAD and ZO are often evaluated on.
>
> Appendix F.4 included experiments with higher LoRA ranks. Table 11 explores accuracy as a function of LoRA rank for OPT-6.7B. The results indicate that:
> BP-Checkpointing degrades gracefully as rank increases, likely due to overfitting, while maintaining stable convergence.
> FmAD becomes unstable beyond $r=1$, often producing NaN outputs, due to the inherent instability of perturbation-based gradient estimates (more on that in Appendix F.5).
> ZO remains stable at higher ranks, but accuracy improvements are limited, achieving only 68.97 % at $r= 32$, 16.57% lower than the 85.54% accuracy from checkpointing.
>
> These observations suggest that increasing the parameter count does not improve (and in fact can worsen) the behavior of FmAD/ZO.
>
> ____
>
> **W3: Comparison with Advanced ZO Variants.**
>
> We have now incorporated the results on SignZO in Appendix F.7, titled ‘Comparison against SignZO’ and colored in blue.
>
> Overall, SignZO improves upon naïve ZO in terms of final accuracy (82.6% vs. 73.6%), but does so by requiring substantially more computation: although its per-iteration FLOPs are nearly identical to ZO, its wall-clock convergence time is 2.7× longer (56.9k s vs. 21.1k s). Compared to BP-Checkpointing, SignZO achieves a markedly smaller memory footprint (5.99 GB vs. 11.66 GB), but only by trading off both efficiency and performance, requiring 3.4× longer time to converge, 3.9× more compute, and yielding 11.2% lower accuracy.
>
> Furthermore, while SignZO converges faster than the variance-reduced ZO-Accumulate and ZO-Multiple baselines, those methods achieve higher accuracies (85.8% and 86.7%), reinforcing the broader trend observed in our paper: stability alone is not sufficient, effective ZO training at scale also requires variance-reduction mechanisms to improve both accuracy and efficiency.
> We also note that SignZO for LLM Finetuning [3] is only published in a workshop, with very limited results: evaluated on just a 2-class classification dataset and two models, and reporting only memory and accuracy without providing compute or wall-clock convergence measurements, and missing backpropagation with checkpointing as a baseline.
>
> We do not include Hybrid ZO–BP methods because they offer limited additional insight on the axes our work studies. These approaches partially reintroduce BP, so their behavior would interpolate between checkpointed BP and pure ZO without revealing new stability dynamics. Their memory footprint also rises toward BP levels, diminishing the very advantage that motivates ZO in the first place. Thus, while such hybrids may yield modest accuracy gains, they would not meaningfully advance our analysis.
>
> [3] Leveraging Coordinate Momentum in SignSGD and Muon: Memory-Optimized Zero-Order LLM Fine-Tuning (Petrov et al., TTODLer-FM @ ICML 2025)

---

> ### Author Response · Authors · 2025-11-18
>
> **W3: More Models for Evaluation.**
>
> We focused on Qwen 2 VL as a representative vision-language model to illustrate scaling behavior and practical feasibility. Based on our current results, we find no intuitive reason why the observed memory–compute–convergence trade-offs would differ substantially with other vision models. We are welcome to suggestions for a specific model if it’s believed that different behavior might occur.
>
> For text-based tasks, we already include a broad range of models: Llama 3.1 (8B), OPT (1.3B, 6.7B, 13B), and medium-sized models Bert (110M, 340M) and RoBERTa (125M, 355M). Across these models, the results are consistent, illustrating that the trends hold broadly.
>
> ___
>
> **Q1: Sensitivity to Perturbation Budget.**
>
> We have included a discussion of the sensitivity to perturbation budget in Appendix F.8 (Sensitivity to Perturbation Budget for OPT13B), with the addition highlighted in blue.
>
> Figure 16 illustrates ZO training of the OPT (13B) model on AGNews with varying perturbation budgets $n$. Increasing $n$ improves gradient estimate quality and final accuracy, but larger budgets applied sequentially due to hardware constraints lead to longer wall-clock convergence times. Small increases in $n$ (e.g., 1 → 10) provide only modest accuracy gains, showing diminishing returns at low budgets. This aligns with our main paper findings on Llama 3.1 (8B), where increasing the number of perturbations improves accuracy but slows convergence, as executing multiple perturbations in parallel is memory-intensive, necessitating sequential application and thus longer training time.
>
> ___
>
> **Q2: Variance Reduction and Choice of Optimizers.**
>
> Thank you for the suggestion to include analyses of the Multiple and Accumulate baselines for both JVP magnitudes and mean gradient values. We have now incorporated these results in Appendix F.5.3, titled ‘Improving Stability via Multiple-Perturbation and Accumulated-Gradient’ and colored in blue.
>
> To summarize, we did not observe the jvp spikes seen in forward only -Vanilla methods when using the variance-reduction baselines. Both -Multiple and -Accumulate variants produce stable, low-variance jvp trajectories under AdamW and SGD, with no divergence-like behavior. This stability arises because averaging (Multiple) or accumulating (Accumulate) gradient signals effectively suppresses the perturbation-induced noise that triggers spikes in the vanilla setting.

---

### Author Response · Authors · 2025-12-03

Dear AC,

The central goal of our work is to critically re-evaluate a growing line of literature [1-8] that explicitly positions Zero-order (ZO) and Forward-mode Auto Differentiation (FmAD) methods as practical alternatives to gradient computation through Backproapgation (BP) for the **standard differentiable LLM training and fine-tuning**. Recent influential works (MeZO [1], MeZO-SVRG [2], Zero-Benchmark [3], SensZOQ [4], FwdLLM [5]) repeatedly claim that ZO and FmAD methods can match the performance of first-order backpropagation while drastically reducing memory usage in standard differentiable models such as LLaMA, OPT, Mistral, and RoBERTa. These claims are made not in gradient-unavailable scenarios, but directly against “backpropagation,” “first-order methods,” or “FO-SGD/FO-Adam,” despite **omitting the most basic memory-efficient baseline**: checkpointed backpropagation (BP-Checkpointing). Our work identifies this omission as a critical gap and provides the missing comparison needed to accurately assess the claimed memory–accuracy trade-offs.

* Reviewers consistently recognize the paper for providing a **comprehensive, rigorous evaluation of gradient-computation alternatives** (FmAD, ZO, and improved variants such as gradient accumulation and adaptive perturbation sampling) (**4Fn3, MFid, nYhR, KQZt**).

* The breadth of the empirical study is highlighted: models from **110M–13B** (BERT, RoBERTa, OPT, LLaMA, Qwen), tasks spanning classification, generation, and VQA, and openly released code (**4Fn3, MFid, nYhR, KQZt**).

* Reviewers commend the **unified theoretical framework** comparing accuracy, memory, compute complexity, and convergence runtime (**4Fn3, MFid, nYhR**).

* Reviewers 4Fn3 and MFid particularly value the inclusion of the **stronger BP-Checkpointing baseline** and the detailed comparison across accuracy, memory, wall-clock convergence, and compute.

* Reviewer 4Fn3 notes that our **failure-mode analysis** (JVP instabilities, optimizer interactions) offers practical insights, a point echoed by nYhR and KQZt.

* Reviewers emphasize the clear empirical conclusion: with the proper baseline, **BP-Checkpointing matches FmAD’s memory while delivering higher accuracy, faster convergence, and lower compute** (**MFid, KQZt, 4Fn3**).

* The paper **corrects misleading impressions** in prior work that compared ZO/FmAD only against non-checkpointed FO-SGD/Adam, showing that reported memory advantages (e.g., 12× or 76%) disappear when benchmarked against memory-efficient BP.
___
[1] Malladi et al., MeZO: Fine-Tuning Language Models with Just Forward Passes, NeurIPS 2023

[2] Gautam et al., Variance-reduced Zeroth-Order Methods for Fine-Tuning Language Models, ICLR 2024

[3] Zhang et al., Revisiting Zeroth-Order Optimization for Memory-Efficient LLM Fine-Tuning: A Benchmark, ICML 2024

[4] Guo et al., Zeroth-Order Fine-Tuning of LLMs with Extreme Sparsity, WANT@ICML 2024

[5] Xu et al., FwdLLM: Efficient Federated Finetuning of Large Language Models with Perturbed Inferences, USENIX ATC 2024

[6] Feng et al., BAFFLE: A Baseline of Backpropagation-Free Federated Learning, ECCV 2024

[7] Cobb et al., Second-Order Forward-Mode Automatic Differentiation for Optimization, arXiv 2024

[8] Panchal et al., Thinking Forward: Memory-Efficient Federated Finetuning of Language Models, NeurIPS 2024

[9] Fournier et al., Can forward gradient match backpropagation?, ICML 2023.

---

> ### Author Response · Authors · 2025-12-03
>
> We further provide a consolidated summary of our responses to all reviewers, aimed at clarifying concerns and resolving misunderstandings.
>
> ___
>
> _Summary of Our Response to Reviewer 4Fn3_
>
> * Clarified that our **smooth non-convex assumptions** follow standard convergence analyses, and referenced sensitivity studies (**Appendix F.4, I.7**) that empirically validate dependence on dimension, perturbation count, and learning rate.
> * Addressed concerns regarding **QLoRA rank-1** by providing results at higher ranks (**Appendix F.4, Table 11**), showing BP-Checkpointing remains stable, FmAD becomes increasingly unstable, and ZO offers limited gains.
> * Expanded comparisons to include a **stronger ZO baseline, SignZO** (**Appendix F.7**), and explained why hybrid ZO–BP approaches yield limited additional insights.
> * Added a **full perturbation-budget sensitivity analysis for OPT-13B** (**Appendix F.8**).
> * Provided **optimizer-specific variance-reduction stability results** (**Appendix F.5.3**), demonstrating that our improved variants avoid JVP-related instabilities.
>
> ___
>
> _Summary of Our Response to Reviewer MFid_
>
> * Addressed the concern about **big-$\mathcal{O}$ notation hiding constants** by pointing to **Appendix H**, which includes explicit FLOP and memory formulas with constants, and by providing empirical wall-clock, memory, and compute measurements across all methods (Sec. 4, Appendix F), confirming that BP-Checkpointing’s higher FLOPs appear in practice.
> * Responded to concerns about **parallelizability of forward-only methods** by referencing our evaluation of **sequential vs. parallel ZO and FmAD variants**, showing that parallelization reduces runtime but introduces increased memory and compute costs, quantified across accuracy, runtime, memory, and compute (Sec. 4.1–4.4).
> * Clarified the questions about **FLOPs/iteration for FmAD vs. BP-Checkpointing** by explaining that FmAD’s per-layer JVP chain introduces a recursive dependency similar to backprop, causing empirical FLOPs to match BP-Checkpointing despite the “forward-only” label.
> * Clarified the **computational complexity of BP-Checkpointing** in **Appendix H.3**, including the recomputation overhead term and its logarithmic scaling with depth, resolving the reviewer’s question about Table 2’s complexity expression.
>
> ___
>
> _Summary of Our Response to Reviewer nYhR_
>
> * Addressed the concern that the paper “reiterates numerical results” by highlighting the **substantive conceptual analyses** provided throughout the work.
>
> * Clarified why **parallel vs. sequential compute costs appear identical**: parallelization reduces wall-clock time but **does not reduce total operations** (Appendix H).
>
> * Addressed fairness concerns by highlighting our comprehensive evaluation of FmAD and ZO variants (-accumulate, -multiple, -SVRG, -Adaptive [an adaptive variant that selects perturbations using prior gradients, which is similar to what is emphasized by the reviewer] and -Sparse), showing that even adaptive strategies yield only limited or inconsistent gains and that the core efficiency–accuracy trade-offs remain unchanged.
>
> * Evaluated multiple **direction-sampling strategies** (Appendix F.6), confirming core limitations persist across variants.
>
> * Responded to reviewer’s request for alternative direction-selection strategies by adding **Appendix F.6 (“Effect of Perturbation Distributions and Normalization Strategies”)**:
>
>   * Compared Normal vs. Uniform sampling and normalized vs. unnormalized directions.
>   * Found that **normalization consistently reduces performance** by discarding informative scale variation.
>   * Showed distribution changes have only mild effects and do **not** mitigate inherent high-variance issues.
>   * Noted that **full basis directions (n = D) are infeasible** for billion-parameter LLMs, making random sampling the only practical strategy.
>
>
> ___
>
> _Summary of Our Response to Reviewer KQZt_
>
>
>
> * We emphasized that our work is a **course-corrective empirical study**, not a negative verdict: if ZO/FmAD inferiority were “obvious,” prior works would have included checkpointing, but none did.
>
> * We **explicitly scoped conclusions to differentiable LLM fine-tuning**, while acknowledging valid use cases of ZO/FmAD in non-differentiable, black-box, or privacy-restricted settings.
>
> * Overall,  **we test the claims made by ZO/FmAD papers, add the missing BP-Checkpointing baseline, and quantify memory, accuracy, runtime, and compute; showing that ZO/FmAD offer no practical advantage in differentiable LLM fine-tuning while remaining useful in other domains.**

---

### Meta-Review · Area_Chair_ywMg · 2026-01-04

**Summary:**

The paper presents an extensive empirical study comparing forward-mode automatic differentiation, zero-order optimization, and checkpointed backpropagation when the gradients of the parameters are available. The authors claim that the community systematically disregards checkpointed backpropagation as a baseline, leading to the incorrect conclusion that the aforementioned alternative optimization methods can compete with backpropagation. It seems that the reviews are coming from two camps: experts who consider the proposed results trivial and unsurprising given their extensive knowledge of the field, and researchers who find the results interesting but are not very confident in their evaluation and may misinterpret the novelty of the results. In particular, the concerns of the experts are:
- The theoretical analysis presented in the paper is not novel, which the authors confirm.
- While the main statement of the paper is that currently the field lacks proper comparison with checkpointing BP, reviewers nYhR and KQZt claim that the proposed results are not surprising at all. The comparison proposed in the paper is common knowledge in the community.
- The paper does not have any additional insights but saying that checkpointing BP has to be included as a baseline.
The concerns raised by the researchers include comparisons with additional baselines, which the authors address primarily.

Overall, the lack of a reviewer willing to champion the paper's acceptance informs my decision to recommend rejection.

**Reviewer Concerns:**

The authors did a great job during the rebuttal, addressing the practical concerns regarding the comparison and clarifying the results and claims. However, the fundamental concerns about the contribution to the field, the novelty of the analysis and the results were not addressed.

**Reviewer Scores:**

The concerns raised by the reviewers with positive scores were relatively minor from the beginning, which might suggest that their careful evaluation of the paper was driven by more fundamental concerns about the novelty of the results. Unfortunately, the discussion of these major concerns regarding novelty does not seem fruitful. Thus, I conclude that no significant changes in scores would take place during the discussion.

---

### Decision · Program_Chairs · 2026-01-26

Reject